# Telomere-to-mitochondria signalling by ZBP1 mediates replicative crisis

Joe Nassour[1], Lucia Gutierrez Aguiar[1], Adriana Correia[1,2], Tobias T. Schmidt[1], Laura Mainz[1], Sara Przetocka[1], Candy Haggblom[1], Nimesha Tadepalle[1], April Williams[1], Maxim N. Shokhirev[1], Semih C. Akincilar[3,4], Vinay Tergaonkar[3,4,5,6], Gerald S. Shadel[1✉] & Jan Karlseder[1✉]

Cancers arise through the accumulation of genetic and epigenetic alterations that enable cells to evade telomere-based proliferative barriers and achieve immortality. One such barrier is replicative crisis—an autophagy-dependent program that eliminates checkpoint-deficient cells with unstable telomeres and other cancer-relevant chromosomal aberrations[1,2]. However, little is known about the molecular events that regulate the onset of this important tumour-suppressive barrier. Here we identified the innate immune sensor Z-DNA binding protein 1 (ZBP1) as a regulator of the crisis program. A crisis-associated isoform of ZBP1 is induced by the cGAS–STING DNA-sensing pathway, but reaches full activation only when associated with telomeric-repeat-containing RNA (TERRA) transcripts that are synthesized from dysfunctional telomeres. TERRA-bound ZBP1 oligomerizes into filaments on the outer mitochondrial membrane of a subset of mitochondria, where it activates the innate immune adapter protein mitochondrial antiviral-signalling protein (MAVS). We propose that these oligomerization properties of ZBP1 serve as a signal amplification mechanism, where few TERRA–ZBP1 interactions are sufficient to launch a detrimental MAVS-dependent interferon response. Our study reveals a mechanism for telomere-mediated tumour suppression, whereby dysfunctional telomeres activate innate immune responses through mitochondrial TERRA–ZBP1 complexes to eliminate cells destined for neoplastic transformation.

Replicative senescence and crisis constitute two anti-proliferative barriers that human cells must evade to gain immortality[3]. Senescence is p53 and RB dependent and occurs when shortened telomeres elicit a DNA-damage response[4,5]. Loss of cell cycle checkpoints renders cells incapable of senescence activation, resulting in continued proliferation and telomere shortening. Such cells eventually succumb to crisis, characterized by extensive cell death and genome instability[1,2,6,7]. Crisis provides a redundant tumour-suppressor mechanism for replicative senescence, whereby CGAS–STING signalling triggers a non-canonical form of autophagy capable of executing cell death rather than sustaining cell survival[1]. Although this discovery established an essential role for autophagy during cell death in crisis, it remained unclear how dysfunctional telomeres engage nucleic-acid-sensing machineries and activate innate immune signalling pathways that are required for cell death.

### ZBP1 mediates an innate immune response in crisis

To study crisis, human papillomavirus E6 and E7 or SV40 large T antigen (SV40LT) were introduced into primary human IMR90 (IMR90$^{E6E7}$) and WI38 (WI38$^{SV40LT}$) lung fibroblasts[1,2], respectively, thereby silencing or disrupting the p53 and RB tumour-suppressor pathways[8–11]. Checkpoint-deficient cells bypassed senescence and reached crisis at population doubling 105–110 (PD105–110) for IMR90$^{E6E7}$ and PD85–90 for WI38$^{SV40LT}$ (refs. [1,2]). RNA sequencing (RNA-seq) analysis revealed profound transcriptional changes during crisis with an overlap of upregulated genes (Extended Data Fig. 1a,b). Crisis-associated processes were predominantly linked to innate immunity and inflammation (Extended Data Fig. 1c). Interferon (IFN)-stimulated genes (ISGs) (such as *CD74*, *DDIT4*, *GBP2*, *ISG20* and *RTP4*) were induced, consistent with an inflammatory status associated with replicative crisis (Extended Data Fig. 2a,b).

The inherent stringency of replicative crisis offered a powerful system in which to conduct a positive selection CRISPR–Cas9 knockout screen (Fig. 1a (left)). Survivors are expected to have lost pathways that are required for crisis, such as the ones linking dysfunctional telomeres to innate immune activation. Enrichment analysis revealed first that Gene Ontology (GO) terms associated with inflammation and innate immunity were predominant (Extended Data Fig. 2c). Second, the innate immune sensor ZBP1 (also known as DAI and DLM-1) emerged as a top hit, along with the previously characterized CGAS–STING–IFN pathway[1] (Fig. 1a (right)). ZBP1, an ISG product[12], was described as cytosolic nucleic acid sensor that induces type I IFNs and regulates innate immunity and cell death[13]. CRISPR-mediated deletion of *ZBP1* demonstrated its essential role in crisis. Control cells (sgLUC, sgGFP)

[1]The Salk Institute for Biological Studies, La Jolla, CA, USA. [2]Departamento de Biologia Vegetal, Faculdade de Ciências da Universidade de Lisboa (FCUL), Lisbon, Portugal. [3]A*STAR Division of Cancer Genetics, Institute of Molecular and Cell Biology (IMCB), Singapore, Singapore. [4]Therapeutics Laboratory of NFκB Signaling, Institute of Molecular and Cell Biology (IMCB), Singapore, Singapore. [5]Department of Pathology, Yong Loo Lin School of Medicine, National University of Singapore (NUS), Singapore, Singapore. [6]Department of Biochemistry, Yong Loo Lin School of Medicine, National University of Singapore (NUS), Singapore, Singapore. ✉e-mail: gshadel@salk.edu; karlseder@salk.edu

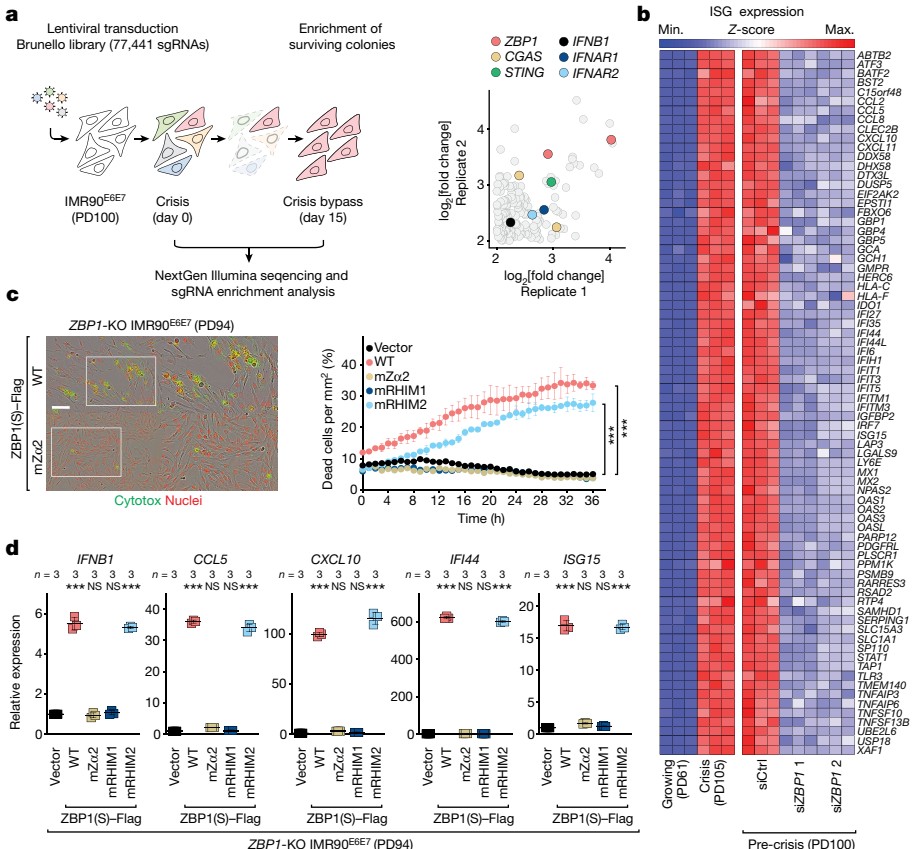

**Fig. 1 | ZBP1 mediates an innate immune response during crisis. a**, Pre-crisis IMR90[E6E7] fibroblasts were transduced with the Brunello library (four sgRNAs per gene), genomic DNA was collected at days 0 and 15, and gRNA changes were measured (left). Right, enrichments. Two technical replicates. Data represent the log₂-transformed fold change in read counts before and after enrichment (PinAPL-Py). gRNAs with a log₂-transformed fold change of >2 are shown (Supplementary Information). Two independent experiments were performed. **b**, RNA-seq analysis of ISGs of growing (PD61), crisis (PD105) and pre-crisis (PD100) IMR90[E6E7] fibroblasts that were transfected with two individual short interfering RNAs (siRNAs) against *ZBP1* (si*ZBP1*) or control (siCtrl). Significantly upregulated (crisis versus growing) and downregulated ISGs (si*ZBP1* versus siCtrl) with a fold change of >1.5 and a false-discovery rate (FDR)-adjusted *P* < 0.05 are shown (the complete list of ISGs used is from ref. [51]). One experiment was performed. **c**, Representative IncuCyte images of pre-crisis (PD94)

*ZBP1*-knockout IMR90[E6E7] fibroblasts reconstituted with either WT ZBP1(S)–Flag or ZBP1(S)–Flag containing point mutations in Zα2, after 12 h of incubation (left). Dead cells are labelled with Cytotox green and nuclei are labelled with NucLight red. Scale bar, 150 μm. Three independent experiments were performed. Right, IncuCyte analysis of cell death of pre-crisis (PD94) *ZBP1*-knockout IMR90[E6E7] fibroblasts reconstituted with either WT ZBP1(S)–Flag or ZBP1(S)–Flag containing point mutations in the Zα2, RHIM1 or RHIM2 domains, measured in real time by Cytotox green. Data are mean ± s.e.m. from technical replicates. Statistical analysis was performed using one-way analysis of variance (ANOVA); ***P* < 0.001. Three independent experiments were performed. **d**, RT–qPCR analysis of ISGs. Expression levels were normalized to control cells with the empty vector. Data are mean ± s.d. from technical replicates. *n* values indicate the number of technical replicates. Statistical analysis was performed using one-way ANOVA; NS, not significant; ***P* < 0.001. Three independent experiments were performed.

entered crisis around PD90 for WI38[SV40LT] and PD107 for IMR90[E6E7], when cell death was frequent and replicative ability was reduced (Extended Data Fig. 3a–d). However, pooled *ZBP1*-knockout cells continued to proliferate for an additional 7–10 population doublings beyond the crisis plateau with a notable reduction in cell death and maintenance of growth potential (Extended Data Fig. 3a–d). Cells that have bypassed crisis showed reduced expression and secretion of IFNβ, indicative of impaired type I IFN activity (Extended Data Fig. 3e,f). Depletion of ZBP1 abrogated the ISG signature, confirming that ISG induction during crisis is attributable to ZBP1 (Fig. 1b). These results directly linked ZBP1 to innate immune activation and cell death during crisis.

ZBP1 can bind to left-handed Z-nucleic acid in a structure-specific manner[14–17]. Originally described as DNA sensor[18], ZBP1 is now appreciated to also sense RNAs that might exist in the Z conformation[14,19–26]. Alternative splicing of human *ZBP1* generates two main isoforms: full-length ZBP1 (ZBP1(L)) contains two N-terminal nucleic-acid-binding domains (Zα1 and Zα2) followed by RHIM domains (RHIM1 and RHIM2) and a structurally undefined C terminus; the short isoform (ZBP1(S)) lacks Zα1 but retains Zα2[27,28] (Extended Data Fig. 4a). *ZBP1* mRNA levels were induced in crisis, consistent with enhanced IFN signalling

(Extended Data Fig. 4b). ZBP1(S) protein was upregulated, whereas ZBP1(L) was less abundant (Extended Data Fig. 4c,d). Increased ZBP1 expression correlated with the activation of type I IFN signalling, marked by TBK1, IRF3 and STAT1 phosphorylation (Extended Data Fig. 4c). LC3 lipidation and accumulation of LC3-II indicated ongoing autophagy[1] (Extended Data Fig. 4c). Targeting the major innate immune sensors and adapters revealed that ZBP1 induction was dependent on CGAS, STING and IFN, suggesting that CGAS–STING could drive an initial transcriptional induction of ZBP1 and prime cells to respond to aberrant accumulation of additional cytosolic immunostimulatory nucleic acid species (Extended Data Fig. 4e).

## ZBP1 signalling requires Zα2 and RHIM1 domains

To address which domains of ZBP1 are critical for signalling during crisis, we disrupted the nucleic-acid-binding and RHIM functions by point mutations or truncations (Extended Data Fig. 5a). *ZBP1*-knockout cells were reconstituted with wild-type (WT) or mutant forms of ZBP1(S) (Extended Data Fig. 5b,c) and tested for their susceptibility to induce ISGs and undergo cell death. Pre-crisis cells expressing WT ZBP1(S)

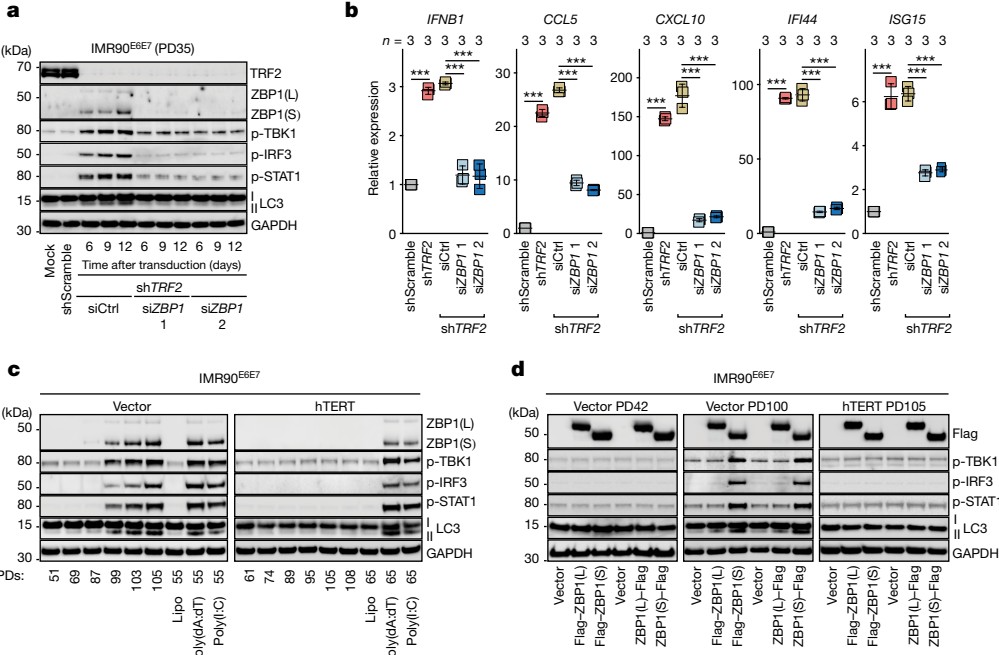

**Fig. 2 | Telomere dysfunction is required for ZBP1-dependent IFN signalling.** **a**, Immunoblot analysis of growing (PD35) IMR90^E6E7 fibroblasts expressing non-targeting control short hairpin RNA (shRNA) or shRNA against *TRF2* (sh*TRF2*). sh*TRF2* cells were transfected with either two individual siRNAs against *ZBP1* (si*ZBP1*) or non-targeting control siRNA at day 4 after shRNA transduction. Protein extracts were collected at days 6, 9 and 12 as shown in the timeline of the experiment in Extended Data Fig. 6i. Mock represents non-transduced cells. GAPDH was the loading control. Two independent experiments were performed. p, phosphorylated. **b**, RT–qPCR analysis of ISGs in growing (PD35) IMR90^E6E7 fibroblasts expressing non-targeting control shRNA or shRNA against *TRF2*. sh*TRF2* cells were transfected with either two individual siRNAs against *ZBP1* or non-targeting control siRNA (siCtrl) at day 4 after shRNA transduction. RNA extracts were collected at day 12. Expression levels were normalized to control cells with non-targeting shRNA. Data are mean ± s.d. of technical replicates.

*n* values indicate the number of technical replicates. Statistical analysis was performed using one-way ANOVA; ***$P$ < 0.001. Three independent experiments were performed. **c**, Immunoblot analysis of IMR90^E6E7 fibroblasts expressing either empty vector or hTERT at the indicated PDs. Growth curves and telomere restriction fragment (TRF) analysis are shown in Extended Data Fig. 7a,b. Cells stimulated with 2 µg ml⁻¹ of poly(deoxyadenylic-deoxythymidylic) (poly(dA:dT)) or poly(inosinic:cytidylic) (poly(I:C)) for 24 h were used as positive controls. GAPDH was used as the loading control. Two independent experiments were performed. **d**, Immunoblot analysis of growing (PD42), pre-crisis (PD100) and telomerase-positive (PD105) IMR90^E6E7 fibroblasts expressing ZBP1(L) or ZBP1(S). Flag tag was added to either the N terminus or the C terminus. GAPDH was the loading control. Two independent experiments were performed. Lipo, lipofectamine.

showed increased expression of ISGs concomitant with elevated cell death (Fig. 1c,d). However, cells reconstituted with ZBP1(S) containing mutations in key conserved residues involved in nucleic acid binding (N141A, Y145A)^29 or RHIM1-based interaction (I206A, Q207A, I208A, G209A)^30,31 did not induce ISGs and cell death (Fig. 1c,d). Mutating RHIM2 (V264A, Q265A, L266A, G267A)^30,31 did not prevent activation of ISGs or cell death (Fig. 1c,d). Comparable patterns were observed with ZBP1(S) truncation mutants (Extended Data Fig. 5d). Examining pathways downstream of ZBP1(S) revealed no role for RIPK1- or RIPK3-dependent PANoptosis, whereas autophagy was induced under these circumstances^1 (Extended Data Fig. 6a–d). Accordingly, suppression of RIPK3/NLRP3/PYCARD/CASP1-mediated pyroptosis, RIPK1/FADD/CASP8-mediated apoptosis or RIPK3/MLKL-mediated necroptosis did not protect cells from ZBP1(S)-induced cell death (Extended Data Fig. 6e). By contrast, elimination of components required for IFN signalling or the autophagy machinery phenocopied the protective effects of ZBP1 depletion (Extended Data Fig. 6e). These findings revealed a mechanism by which ZBP1(S) drives cell death in a manner that is dependent on autophagy and type I IFN activities.

## Innate signalling requires telomere dysfunction

Replicative crisis is a telomere-dependent program. We therefore depleted the telomere protection factor TRF2 in growing IMR90^E6E7 cells, which resulted in telomere deprotection and fusion (Extended Data Fig. 6f–h), accompanied by ZBP1(S) upregulation and a type I

IFN response through TBK1 phosphorylation (Fig. 2a). Depletion of ZBP1 dampened TBK1–IRF3 signalling, reduced ISG expression and attenuated autophagy, without affecting telomere fusions (Fig. 2a,b and Extended Data Fig. 6i,j). Expression of the catalytic subunit of telomerase (hTERT) in growing cells resulted in telomere maintenance and continued growth past crisis (Extended Data Fig. 7a,b). These immortalized cells did not express ZBP1 or activate IFN signalling (Fig. 2c and Extended Data Fig. 7c), whereas control cells entered crisis, upregulated ZBP1(S) and activated the TBK1–IRF3–IFN signalling axis (Fig. 2c and Extended Data Fig. 7c). hTERT expression did not affect the ability of cells to launch an IFN response or stimulate autophagy when treated with exogenous double-stranded RNA or DNA (Fig. 2c). Expression of either ZBP1(L) or ZBP1(S) in growing or telomerase-positive cells did not activate the IFN pathway (Fig. 2d (left and right)). ZBP1(S) potentiated an IFN response only when expressed in pre-crisis cells with short telomeres (Fig. 2d (middle)), suggesting the requirement for an additional immunostimulatory molecule specific for telomere dysfunction. In conclusion, two stimuli are required for launching a ZBP1-dependent IFN response in crisis: (1) upregulation of ZBP1(S) by CGAS–STING, and (2) a signal provided by dysfunctional telomeres.

## ZBP1 detects TERRA-derived RNA

The telomere-driven IFN response requires the nucleic-acid-binding function of ZBP1 (Fig. 1d and Extended Data Fig. 5d). Furthermore, ZBP1 expression in crisis cells was dependent not only on the CGAS–STING

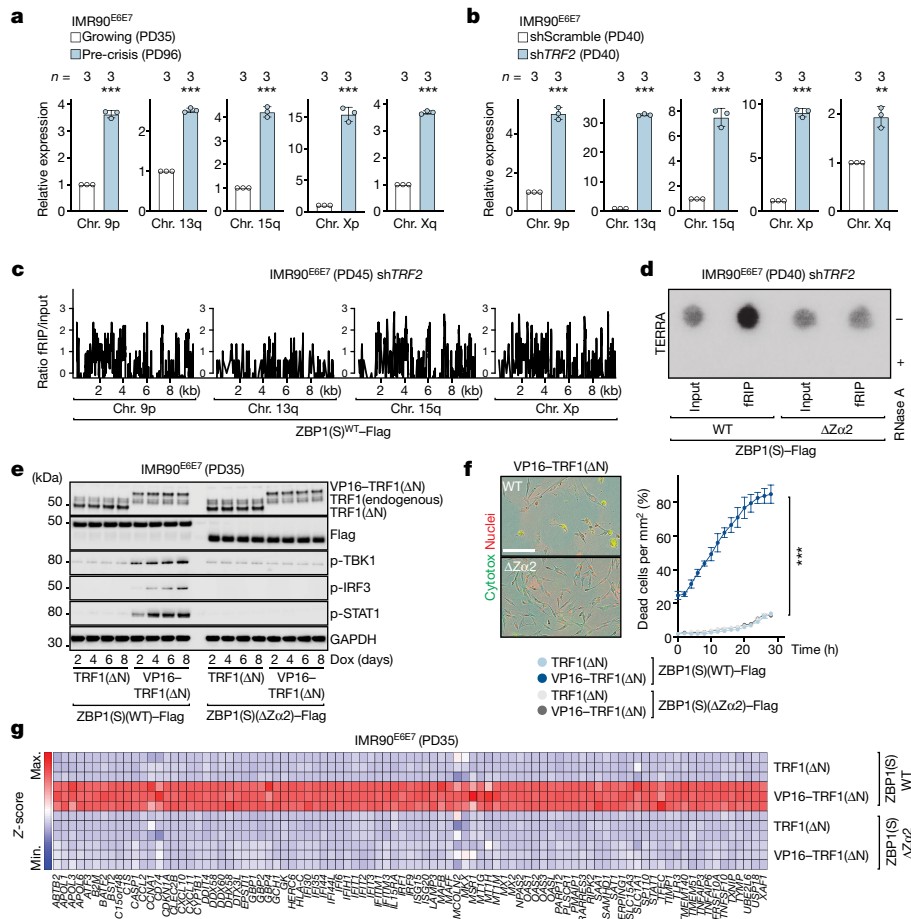

**Fig. 3 | ZBP1 senses TERRA. a,b**, RT–qPCR analysis of TERRA transcripts. RNA from growing and pre-crisis IMR90[E6E7] fibroblasts (**a**) and growing IMR90[E6E7] fibroblasts expressing control shRNA or shRNA against *TRF2* (**b**). RNA was collected at day 4 after transduction, normalized to growing (left) or shScramble (right). Data are mean ± s.d. from technical replicates. *n* values indicate the number of technical replicates. Statistical analysis was performed using two-tailed Student's *t*-test. ***P < 0.001. Three independent experiments. **c**, fRIP–seq enrichment of ZBP1-associated RNA transcripts within 10 kb upstream of the subtelomere–telomere boundary. fRIP analysis was performed on growing (PD45) IMR90[E6E7] fibroblasts expressing WT ZBP1(S)–Flag and transduced with either control shRNA or sh*TRF2*. Immunoprecipitation at day 12 after shRNA transduction. Validation is shown in Extended Data Fig. 8c. fRIP and input samples were normalized to the same sequencing depth and average ratios of individual subtelomeres shown in Extended Data Fig. 8d. Data are log₂-transformed. Two independent experiments were performed. **d**, RNA-dot blot from anti-Flag immunoprecipitates (fRIP) or total lysates (input) using ³²P-dCTP-labelled probes targeting TERRA transcripts. fRIP analysis of growing

(PD40) IMR90[E6E7] fibroblasts expressing either WT ZBP1(S)–Flag or mutant ZBP1(S)–Flag lacking Zα2 with sh*TRF2*. Immunoprecipitation was performed at day 12 after shRNA transduction. Validation is shown in Extended Data Fig. 8e. RNase A treatment was used as a contamination control. Two independent experiments were performed. **e**, Immunoblot analysis of growing (PD35) IMR90[E6E7] fibroblasts (WT ZBP1(S)–Flag or ZBP1(S)–Flag lacking Zα2) with either TRF1(ΔN) or VP16–TRF1(ΔN). Extracts were collected as indicated after doxycycline (Dox) (1 μg ml⁻¹) treatment. Two independent experiments were performed. **f**, IncuCyte images at day 50 (the growth curve is shown in Extended Data Fig. 10a) after 24 h of incubation (left). Cells are labelled as described in Fig. 1c. Scale bar, 200 μm. Three independent experiments were performed. Right, cell death by Cytotox green incorporation. Data are mean ± s.e.m. from technical replicates. Statistical analysis was performed using one-way ANOVA. ***P < 0.001. Three independent experiments were performed. **g**, ISG RNA heat maps. RNA was collected at day 10 after doxycycline treatment. Significantly upregulated ISGs (fold change > 3; FDR-adjusted *P* < 0.05) are shown. One experiment was performed. Chr., chromosome.

pathway, but also on mitochondrial antiviral signalling protein (MAVS) (Extended Data Fig. 4e)—an innate immune adapter that is located at the mitochondrial outer membrane (MOM). MAVS can be activated by the viral RNA sensors retinoic-acid-inducible gene I (RIG-I) and melanoma differentiation-associated protein 5 (MDA5)[32–35], which did not have a role in ZBP1 upregulation during crisis (Extended Data Fig. 4e). Dysfunctional telomeres are actively transcribed into long non-coding RNA species termed TERRA, resulting in subtelomeric RNA sequences followed by a variable number of telomeric UUAGGG repeats[36,37]. RNA-dot blot and quantitative PCR with reverse transcription (RT–qPCR) analysis showed enhanced TERRA transcription from critically short telomeres in crisis or telomeres depleted of TRF2[38] (Fig. 3a,b and Extended Data Fig. 8a,b). Given that long non-coding RNAs interact with cytosolic

sensors to regulate immune responses[39,40] and that TERRA has been linked to inflammation previously[41,42], we reasoned that TERRA could be the immunostimulatory nucleic acid species that is recognized by ZBP1(S). We first depleted TRF2 in growing IMR90[E6E7] fibroblasts expressing WT ZBP1(S) and then performed formaldehyde cross-linking combined with ZBP1-associated RNA immunoprecipitation (Fig. 3c and Extended Data Fig. 8c,d). Formaldehyde cross-linking combined with RNA immunoprecipitation–sequencing (fRIP-seq) reads were mapped, and enrichment profiles were generated by calculating the fold change of the read counts between the fRIP and input samples. The analysis revealed reads for subtelomeres, with sharp peaks appearing at the telomere-proximal region located 5–10 kb upstream of TTAGGG repeats, implying that ZBP1 senses TERRA-containing transcripts

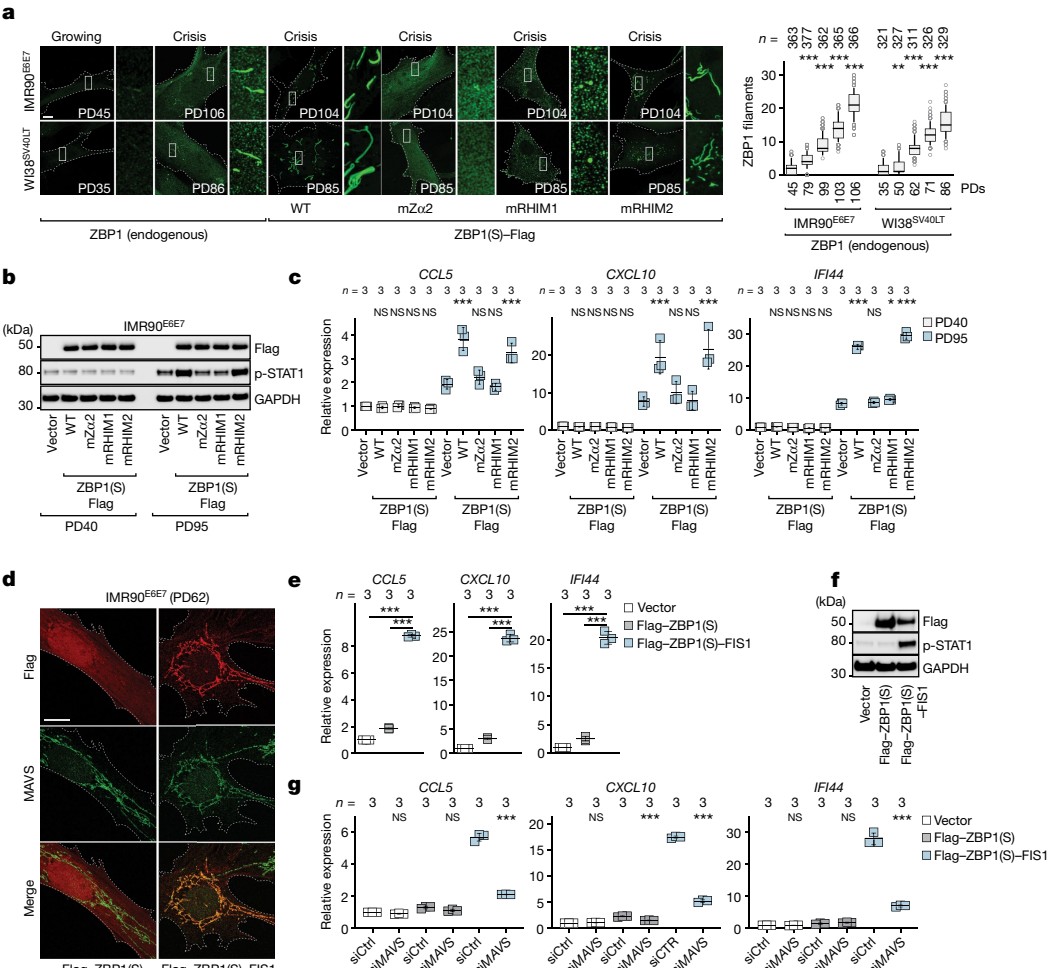

**Fig. 4 | ZBP1-dependent signalling is mediated by MAVS. a**, Growing and crisis IMR90^E6E7 and WI38^SV40LT fibroblasts, and crisis IMR90^E6E7 and WI38^SV40LT fibroblasts expressing WT ZBP1(S)–Flag or ZBP1(S)–Flag containing mutations in Zα2, RHIM1 or RHIM2, immunostained against endogenous ZBP1 or Flag (left). Scale bar, 10 μm. Three independent experiments were performed. Right, box and whisker plots of endogenous ZBP1 filaments. The median (centre line), first and third quartiles (box limits) and 10th and 90th percentiles (whiskers) are shown. *n* values indicate the cell number. Statistical analysis was performed using one-way ANOVA; **P < 0.01; ***P < 0.001. Three independent experiments were performed. The white boxes indicate higher-magnification images. **b**, Immunoblot analysis of growing and pre-crisis IMR90^E6E7 fibroblasts expressing empty vector, WT ZBP1(S)–Flag or ZBP1(S)–Flag containing domain point mutations. GAPDH was used as the loading control. Two independent experiments were performed. **c**, RT–qPCR analysis of ISGs in growing and pre-crisis IMR90^E6E7 fibroblasts. Expression levels were normalized to growing cells expressing the empty vector. Data are mean ± s.d. of technical replicates. *n* values indicate the number of technical replicates. Statistical analysis was performed using one-way ANOVA. NS, not significant; ***P < 0.001. Three

independent experiments were performed. **d**, Growing IMR90^E6E7 fibroblasts expressing Flag–ZBP1(S) or Flag–ZBP1(S)–FIS1 immunostained for Flag and MAVS. Scale bar, 10 μm. Three independent experiments were performed. **e**, RT–qPCR analysis of ISGs. RNA extracts were collected 10 days after transduction. Expression levels were normalized to the empty vector. Data are mean ± s.d. from technical replicates. *n* values indicate the number of technical replicates. Statistical analysis was performed using one-way ANOVA. ***P < 0.001. Three independent experiments were performed. **f**, Immunoblot analysis of growing IMR90^E6E7 fibroblasts as described in **e**. GAPDH was used as the loading control. Two independent experiments were performed. **g**, RT–qPCR analysis of ISGs in growing (PD66) IMR90^E6E7 fibroblasts. Cells were transfected with si*MAVS* or non-targeting control siRNA at day 10 after transduction. RNA extracts were collected at day 14. Expression levels were normalized to empty vector siCtrl cells. Data are mean ± s.d. from technical replicates. *n* values indicate the number of technical replicates. Statistical analysis was performed using one-way ANOVA. NS, not significant; ***P < 0.001. Three independent experiments were performed.

(Fig. 3c and Extended Data Fig. 8c,d). To confirm ZBP1–TERRA interactions, we suppressed TRF2, immunoprecipitated WT or mutant ZBP1 lacking Zα2 (ZBP1(ΔZα2)) and assessed the presence of TERRA using an RNA-dot blot analysis (Fig. 3d and Extended Data Fig. 8e). A strong signal for TERRA was detected in immunoprecipitates from WT ZBP1 cells but not in those expressing mutant ZBP1 (Fig. 3d). Treatment with RNase A confirmed RNA-specific signals (Fig. 3d). These results indicate that telomere dysfunction contributes to the accumulation of TERRA molecules that physically interact with ZBP1 through its Zα2 domain.

To evaluate the role of TERRA–ZBP1 interactions in innate immune activation, we expressed a mutant form of TRF1 lacking the N terminus

(TRF1(ΔN)) alone or fused to the VP16 transcriptional activation domain (VP16–TRF1(ΔN))[42] in growing IMR90^E6E7 fibroblasts. Stable expression of these proteins was confirmed, and upregulation of TERRA was validated (Fig. 3e and Extended Data Fig. 9a–c). WT ZBP1-expressing cells responded to TERRA induction by activating the IFN signalling pathway and ISG expression (Fig. 3e,g). Accordingly, cells entered a crisis-like state associated with cell death and elevated ISG levels (Fig. 3f and Extended Data Fig. 10a–d). No loss of telomere protection, no activation of ATM- and ATR-dependent DNA-damage response and no formation of fused chromosomes were observed (Extended Data Fig. 11a,b). Partial depletion of TERRA in cells expressing WT

ZBP1 delayed STAT1 activation (Extended Data Fig. 10e–g). Cells expressing mutant ZBP1 with disrupted nucleic-acid-binding activity were insensitive to TERRA and did not activate the innate immune response (Fig. 3e–g and Extended Data Fig. 10a–d). These findings demonstrate a mechanism of innate immune signalling triggered by dysfunctional telomeres, in which TERRA functions as the messenger molecule that binds to ZBP1(S) in the cytoplasm to promote an enhanced type I IFN response.

## ZBP1 forms filaments at mitochondrial membranes

To decipher the mechanism of ZBP1 signalling, we performed immunostaining of ZBP1 in crisis cells, which revealed filamentous structures in the cytoplasm (Fig. 4a). ZBP1 lacking functional Zα2 and RHIM1 domains did not form these filaments, whereas no significant effect was observed after disruption of RHIM2 (Fig. 4a). A lack of filaments correlated with impaired type I IFN signalling (Fig. 4b,c), indicating that the assembly of ZBP1 filaments is essential for innate immune signalling and requires both TERRA binding through Zα2 and self-oligomerization through homotypic RHIM1 interactions. Multimerization is used by several sensors for signal propagation, including RIG-I[43] and MDA5[44], which translocate to mitochondria to interact and activate MAVS[32–35]; however, they often form aggregated structures rather than filaments[45], suggesting an alternative oligomerization mechanism for ZBP1. We found co-localization of ZBP1 filaments with a subset of mitochondria based on MOM (TOM20) and matrix (MitoTracker and TFAM) markers (Extended Data Fig. 11c). Inhibition of mtDNA replication with 2′,3′-dideoxycytidine[46] had no effect on the number of ZBP1 filaments even though mtDNA and mtRNA levels were significantly reduced (Extended Data Fig. 11d). Depletion of either SUV3 or PNPase enzymes, components of the mitochondrial degradosome[47], led to significant accumulation of mtRNA without altering ZBP1 filament formation (Extended Data Fig. 11e). We therefore concluded that ZBP1-mediated innate immune response requires formation of filaments on mitochondria, independently of sensing mtDNA or mtRNA.

## MAVS is required for a ZBP1-mediated IFN response

Given that MAVS, a crucial adapter for RNA sensors anchored to the MOM, was enriched in our CRISPR–Cas9 knockout screen (Supplementary Information), and its depletion prevented IFN-dependent ZBP1 induction during crisis (Extended Data Fig. 4e), we reasoned that ZBP1-driven cell death could be mediated through MAVS. Indeed, depletion of MAVS prevented ISG expression in response to ZBP1 and TERRA co-expression independently of RIG-I and MDA5 (Extended Data Fig. 12a–c). Deletion of *MAVS* also reverted ISG expression during crisis, prevented autophagy upregulation and reduced the frequency of cell death (Extended Data Fig. 12d–f), suggesting that ZBP1 filaments at the mitochondria could be the critical event downstream of telomere stress signalling that leads to MAVS activation. To test this, ZBP1(S) was fused to the mitochondrial-targeting sequence of FIS1 to direct it specifically to the MOM in growing cells with functional telomeres (Extended Data Fig. 13a). ZBP1(S)–FIS1 colocalized with MAVS, triggered an IFN response, stimulated autophagy and induced cell death in fibroblasts and epithelial cells (Fig. 4d–f and Extended Data Fig. 13c,d). ISG induction by mitochondria-targeted ZBP1(S)–FIS1 was dependent on MAVS (Fig. 4g and Extended Data Fig. 13b). Finally, as shown by suppression of IFNAR2, cell death by autophagy required a secondary signalling pathway downstream of ZBP1–MAVS, by which secreted IFNs bind to and activate their cognate IFN receptor complexes in an autocrine and paracrine manner (Extended Data Fig. 13c,d). Together, these results suggest that ZBP1 signalling in crisis cells involves binding to immunostimulatory TERRA, followed by conformational changes allowing cytosol-to-MOM translocation and oligomerization to activate MAVS. We propose that these properties of ZBP1 serve as a signal-amplification mechanism

enabling TERRA molecules to trigger a second wave of IFN signalling that, together with autophagy activation, causes cell death during crisis.

Although ZBP1 functions as an RNA sensor with the ability to activate a MAVS and IFN signalling response, it differs from RIG-I and MDA5 in several aspects. First, ZBP1 filament formation occurs at the mitochondrial surface, whereas RIG-I[43,48] and MDA5[44,49,50] undergo oligomerization along double-stranded RNA structures. Second, ZBP1 lacks the caspase recruitment domain (CARD), whereas RIG-I and MDA5 activate MAVS through CARD–CARD interactions[35]. Finally, we propose that TERRA transcripts constitute a structure-specific ligand for ZBP1, with low or no binding affinity to RIG-I and MDA5. These differences may prime ZBP1 to propagate the inflammatory signalling cascade specifically in response to telomere dysfunction, revealing a unique mechanism of MAVS-dependent IFN activation and subsequent cell death by autophagy.

Our study reveals a mechanism for telomere-mediated tumour suppression, whereby dysfunctional telomeres in crisis stimulate two intertwined cytosolic nucleic-acid-sensing pathways and trigger a lethal IFN response (Extended Data Fig. 13e). We propose that the breakage of fused telomeres and the subsequent release of nuclear DNA into the cytoplasm drives initial activation of the CGAS–STING pathway and expression of ISGs, including *ZBP1*. However, crisis-associated cell death requires additional downstream activation of ZBP1(S) by TERRA and formation of ZBP1 filaments on mitochondria. This promotes an inflammatory loop leading to the expression of an ISG profile that, in concert with activation of autophagy through an as yet undetermined pathway, drives cell death specifically in replicative crisis. The simultaneous activation of CGAS–DNA-sensing and ZBP1–RNA-sensing pathways enables the innate immune system to orchestrate an efficient, type-I-IFN-dependent cell death response to eliminate precancerous cells with unstable telomeres. These findings highlight a synergy between critically short telomeres, mitochondria and innate immunity that has evolved to prevent age-associated cancer initiation in humans.

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

## Methods

### Cell culture

IMR90 (CCL-186) and WI38 (AG06814-N) fibroblasts were purchased from ATCC and the Coriell Institute for Medical Research, respectively. IMR90 and WI38 fibroblasts were grown under 7.5% $CO_2$ and 3% $O_2$ in GlutaMax-DMEM (Gibco, 10569-010) supplemented with 0.1 mM non-essential amino acids (Corning, 25-025-Cl) and 15% fetal bovine serum (VWR/Seradigm, 97068-085, 311K18). Human mammary epithelial cells (HMECs; CC-2551) were purchased from Lonza and were grown under 5% $CO_2$ and 3% $O_2$ using the mammary epithelial cell medium complete kit (Lifeline Cell Technology, LL-0061). The number of PDs was calculated at each passage using the following equation: $PD = \log[\text{number collected/seeded number}]/\log_2$. Cells have been tested to be free of mycoplasma.

### Plasmids

Non-coding scramble shRNA was obtained from D. Sabatini through Addgene (plasmid 1864)[52]. *TRF2* shRNA in pLKO.1 was obtained from Open Biosystems. HPV16 E6E7, SV40 LT and H2B-mCherry in pLXSN3 and hTERT in pBabe were obtained from Karlseder laboratory stock[2]. To generate shZBP1 1, oligos (forward, 5′-CCGGCCAAGTCCTCTACCGAATGAACTCGAGTTCATTCGGTAGAGGACTTGGTTTTTG-3′; and reverse, 3′-AATTCAAAAACCAAGTCCTCTACCGAATGAACTCGAGTTCATTCGGTAGAGGACTTGG-5′) were annealed by temperature ramp from 100 °C to 25 °C and cloned in a AgeI- and EcoRI-digested non-coding scramble shRNA vector using T4 DNA ligase. *ZBP1* shRNA in pLKO.1 (shZBP1 2) was obtained from Sigma-Aldrich (TRCN0000123050).

To generate pooled knockout cell populations and knockout clones lentiCRISPRv2 was used. The plasmid was obtained from F. Zhang through Addgene (plasmid 52961)[53] and the guide target sequence was cloned as phosphorylated adapter in the Esp3I-digested lentiCRISPRv2 vector. Individual guide RNAs (gRNAs) for the targeting of corresponding genomic exons were designed using the CRISPOR web tool (http://crispor.org). The guide RNA sequences are as follows: sgZBP1 1, 5′-CACCTGGTGCCATTGAAGGG-(PAM)-3′; sgZBP1 2, 5′-GGACGATTTACCGCCCAGGT-(PAM)-3′; sgZBP1 3, 5′-TGGGACACAGCAATGAGATG-(PAM)-3′; sgGFP, 5′-GAAGTTCGAGGGCGACACCC-(PAM)-3′; sgLUC (sgrenilla_luciferase), 5′-GGTATAATACACCGCGCTAC-(PAM)-3′; sgMAVS, 5′-AAGTTACCCCATGCCTGTCC-(PAM)-3′; and sgSTING, 5′-AATATGACCATGCCAGCCCA-(PAM)-3′. *ZBP1*-knockout clones were generated using sgZBP1 2 5′-GGACGATTTACCGCCCAGGT-(PAM)-3′. After transduction, cells were selected with 1 µg ml⁻¹ puromycin for 1 week and individual clones were isolated using the array dilution method in 96 wells. Clones were tested by western blotting for the absence of ZBP1 protein staining.

All *ZBP1* expression plasmids used here are derivates of pLenti-CMV-GFP-Hygro-(656-4) obtained from E. Campeau and P. Kaufman through Addgene (plasmid 17446)[54]. The sequence of full-length human *ZBP1* transcript variant 1 (GenBank: NM_030776.3) was obtained as gblock from IDT. The empty vector control pLenti-MCS-hygro was cloned in three steps: first, the pLenti-CMV-GFP-Hygro-(656-4) vector was digested with XbaI and SalI and the GFP sequence was replaced with a multiple cloning site (MCS) adapter (MCS1_fw, 5′-CTAGACCTCAGGATCCCGGGACGCGTG-3′; and MCS2_rev, 5′-TCGACACGCGTCCCGGGATCCTGAGGT-3′) to generate pLenti-CMV-MCS-Hygro. Second, the CMV promoter in the pLenti-CMV-MCS-Hygro vector was excised using ClaI and BamHI and replaced with a second adapter (MCS2_fw, 5′-CGATCCTGAGGTCTAGAGAATTCG-3′; and MCS2_rev, 5′-GATCCGAATTCTCTAGACCTCAGGAT-3′). Finally, pLenti-MCS-WPRE-Hygro was digested with SalI and XhoI to remove the WPRE sequence followed by religation to generate pLenti-MCS-Hygro. To clone N-terminally 3×Flag-tagged *ZBP1* expression constructs, pLenti-

CMV-3×Flag-(GS)₅-MCS-Hygro was generated first. For this, the Kozak sequence with a 3×Flag tag and a (GS)₅ linker was amplified from pLenti-FNLS-P2A-GFP-PGK-Puro (obtained from L. Dow through Addgene (plasmid 110869)[55] using primers 5′-CTAGACCTCAGGATCGCCACCATGGACTATAAGGAC-3′ and 5′-ACGCGTCCCGGGATCCAGAGCCGGACCCGCTCCCGGAGCCCTTATCGTCATCGTCTTTGTAATCAATATCATGAT-3′. The fragment was cloned into BamHI-linearized pLenti-CMV-MCS-Hygro using InFusion cloning (Takara Bio, 639650).

*ZBP1* full-length (ZBP1(L)) and short isoform (ZBP1(S), also known as Delta-exon-2, Delta-Z-alpha or isoform 7) were amplified from full-length *ZBP1* gblock DNA using either the primer 5′-GTCCGGCTCTGGATCCATGGCCCAGGCTCCTGCT-3′ or primer 5′-GTCCGGCTCTGGATCCATGGCCCAGGCTCCTGCTGACCCGGGCAGAGAAGCCGAGAGGCCCCAGC-3′ in combination with the primer 5′-ACGCGTCCCGGGATCCCTAAATCCCACCTCCCCACC-3′. *ZBP1* fragments were cloned into BamHI-linearized pLenti-CMV-3×Flag-(GS)₅-MCS-Hygro using InFusion cloning. To clone C-terminally 3×Flag-tagged *ZBP1* expression constructs, pLenti-CMV-MCS-(GS)₅-3×Flag-Hygro was generated first. For this, a (GS)₅ linker with a 3×Flag tag and STOP codon was amplified from pLenti-FNLS-P2A-GFP-PGK-Puro using the primers 5′-CGACTCTAGAGGATCCGGCTCCGGGAGCGGGTCCGGCTCTGACTATAAGGACCACGACGG-3′ and 5′-GAGGTTGATTGTCGACTCACTTATCGTCATCGTCTTTGTAATC-3′. The fragment was cloned into BamHI- and SalI-digested pLenti-CMV-GFP-Hygro-(656-4) using InFusion cloning.

ZBP1(L)-, ZBP1(S)- and ZBP1(S)-deletion mutants were PCR amplified from full-length *ZBP1* gblock DNA and the pLenti-CMV-3×Flag-(GS)₅-ZBP1(S)-Hygro plasmid, respectively, with the primers listed below. Fragments were cloned into BamHI-digested pLenti-CMV-MCS-(GS)₅-3×Flag-Hygro using InFusion cloning.

The primers for the C-terminal tagged *ZBP1* construct cloning were as follows: ZBP1(L) (amino acids 1–429) and ZBP1(S) (amino acids 1–11, 87–429) ZBP1, 5′-CGACTCTAGAGGATCCATGGCCCAGGCTCCTG-3′ and 5′-TCCCGGAGCCGGATCCAATCCCACCTCCCCACC-3′; ZBP1(S)-ΔZα2 (amino acids 167–429), 5′-CGACTCTAGAGGATCCATGCCAGAAGATTCTGGAAGAAGAGCAA-3′ and 5′-TCCCGGAGCCGGATCCAATCCCACCTCCCCACC-3′; ZBP1(S)-ΔZα2-ΔRHIM1-ΔRHIM2 (amino acids 278–429), 5′-CGACTCTAGAGGATCCATGCCGTCCGAGGGCCCT-3′ and 5′-TCCCGGAGCCGGATCCAATCCCACCTCCCCACC-3′; ZBP1(S)-ΔC (amino acids 1–11, 87–319), 5′-CGACTCTAGAGGATCCATGGCCCAGGCTCCTG-3′ and 5′-TCCCGGAGCCGGATCCGGCGGCTTCCCCCT-3′; ZBP1(S)-ΔRHIM1 (amino acids 1–11, 87–194, 220–429) N-terminal fragment, 5′-CGACTCTAGAGGATCCATGGCCCAGGCTCCTG-3′ and 5′-TCCCTGGAGGGTCCATTCTGGCAGATCA-3′ and C-terminal fragment 5′-TGGACCCTCCAGGGAGGACGGT-3′ and 5′-TCCCGGAGCCGGATCCAATCCCACCTCCCCACC-3′; ZBP1(S)-ΔRHIM2 (amino acids 1–11, 87–252, 278–429) N-terminal fragment, 5′-CGACTCTAGAGGATCCATGGCCCAGGCTCCTG-3′ and 5′-TCGGACGGCTGGGGCCCCCAGG-3′ and C-terminal fragment, 5′-GCCCCAGCCGTCCGAGGGCCCTGC-3′ and 5′-TCCCGGAGCCGGATCCAATCCCACCTCCCCACC-3′; and ZBP1(S)-ΔRHIM1-ΔRHIM2 (amino acids 1–11, 87–194, 278–429) N-terminal fragment, 5′-CGACTCTAGAGGATCCATGGCCCAGGCTCCTG-3′ and 5′-TCGGACGGGGGTCCATTCTGGCAGATCA-3′ and C-terminal fragment, 5′-TGGACCCCCGTCCGAGGGCC-3′ and 5′-TCCCGGAGCCGGATCCAATCCCACCTCCCCACC-3′. Zα2, RHIM1 and RHIM2 point mutations were introduced in the pLenti-CMV-ZBP1(S)-(GS)₅-3×Flag-Hygro vector by site-directed mutagenesis using the QuikChange Lightning Site-Directed Mutagenesis Kit (Agilent Technologies, 210518). The primers used here for site-directed mutagenesis were as follows: ZBP1(S)-mZα2 (N141A,Y145A), 5′-GGACAGCAAAAGATGTGGCCCGAGACTTGGCCAGGATGAAGAGCAGGC-3′ and 5′-GCCTGCTCTTCATCCTGGCCAAGTCTCGGGCCACATCTTTTGCTGTCC-3′; ZBP1(S)-mRHIM1

(I206A,Q207A,I208A,G209A), 5′-TGGATTTCCATTGCAAACTCCGA
AGCCGCCGCGGCTGCACACGGGAACATCATTACAAGACAGAC-3′ and
5′-GTCTGTCTTGTAATGATGTTCCCGTGTGCAGCCGCGGCGGCTTCGGA
GTTTGCAATGGAAATCCA-3′; and ZBP1(S)-mRHIM2 (V264A,
Q265A,L266A,G267A), 5′-GCAGTCCATACTGAGACGGGCGGCG
GCGGCACACAGCAATGAGATGAGGC-3′ and 5′-GCCTCATCTCATT
GCTGTGTGCCGCCGCCGCCCGTCTCAGTATGGACTGC-3′.

To express untagged VP16-TRF1ΔN and TRF1ΔN from a
doxycycline-inducible tight TRE promoter, VP16-TRF1ΔN and TRF1ΔN
was subcloned into pLPT-Empty[56]. For this, VP16-TRF1ΔN and TRF1ΔN
were PCR amplified from pINDUCER20-VP16–TRF1ΔN(44–439) and
pINDUCER20-TRF1ΔN(44–439), respectively (both plasmids were kind
gifts from P. M. Lieberman[57]), using either primer 5′-TGGAGAAT
TGGCTAGCATGGCCCCCCCGACCG-3′ or primer 5′-TGGAGAATTGG
CTAGCATGCTTCTCGAGTGCCAGGTGC-3′ in combination with
primer 5′-CCCCAACCCCGGATCCTCAGTCTTCGCTGTCTGAGG
AAATCAG-3′. Fragments were cloned into the NheI and BamHI-
digested pLPT-Empty vector using InFusion cloning.

To localize ZBP1(S) to the mitochondrial outer membrane, the
outer mitochondrial membrane localization (OMM) signal of FIS1 was
fused to the C terminus of 3×Flag-ZBP1(S). The construct was cloned
in three steps: first, pLenti-CMV-MCS-3×Flag-OMM-hygro was generated
by amplifying the OMM of FIS1 with the primers 5′-CGATAAGG
CCATCGTGGGAGGC-3′ and 5′-GAGGTTGATTGTCGACTCAGGA
TTTGGACTTGGACACAG-3′ from MAVS-Mito (obtained from J. Kagan
through Addgene (plasmid 44556)[58] and (GS)5-linker-3×Flag with
the primers 5′-CGACTCTAGAGGATCCGGCTCCGGGAGCGGGT
CCGGCTCTGACTATAAGGACCACGACGG-3′ and 5′-CACGATGGCC
TTATCGTCATCGTCTTTGTAATC-3′ from pLenti-FNLS-P2A-GFP-
PGK-Puro[5]. The two fragments were cloned into BamHI/SalI-
digested pLenti-CMV-GFP-Hygro(656-4) (see methods of ref. [4]) using
InFusion cloning. Next, pLenti-CMV-MCS-(GS)5-3Flag-OMM-hygro
was linearized with BamHI and ZBP1(S) that was amplified with
the primers 5′-CGACTCTAGAGGATCCATGGCCCAGGCTCCTG-3′
and 5′-TCCCGGAGCCGGATCCAATCCCACCTCCCCACC-3′ from
pLenti-CMV-3×Flag-(GS)5-ZBP1(S)-hygro was cloned into the lin-
earized vector using InFusion cloning. Finally, pLenti-CMV-3×Flag-
(GS)5-ZBP1(S)-OMM-hygro was generated by InFusion cloning
using amplified fragments of pLenti-CMV-ZBP1(S)-(GS)5-3Flag-
OMM with the primers 5′-GCCATCGTGGGAGGCATG-3′ and
5′-GGATCCTCTAGAGTCGGTGTCT-3′ and 3×Flag-(GS)5-ZBP1(S)
with the primers 5′-GACTCTAGAGGATCCGCCACCATGGACTAT
AAGGACCACGAC-3′ and 5′-GCCTCCCACGATGGCAATCCCACCTC
CCCACC-3′ from pLenti-CMV-3×Flag-(GS)5-ZBP1(S)-hygro.

T4 ligase (M0202S), T4 PNK (M0201S) and all restriction enzymes
used, except for Esp3I (Thermo Scientific ER0452), were obtained
from NEB. All primers were ordered from Eton Bioscience. PCR
reactions for cloning were performed using KOD Hot Start DNA
Polymerase (Novagen, 71086) according to the manufacturer's
protocol.

### Cell death assays
For real-time assessment of cell death, 2,000 cells per well per con-
dition in triplicate were seeded in 96-well plates (CytoOne, CC7682-
7596) the day before the experiment. The next day, the medium was
changed to medium containing 250 nM IncuCyte Cytotox green dye
(Sartorius, 4633) and IncuCyte Nuclight Rapid red dye (1:100 dilution)
(Sartorius, 4717). Cells were seeded in 96-well plates the day before the
experiment and the medium changed to medium containing 250 nM
IncuCyte Cytotox green dye on the day of the experiment. Cells were
imaged with at least two fields per well every 2 h using the ×10 objective
on the IncuCyte S3 or IncuCyte Zoom live-cell analysis system (Sarto-
rius). The total number of nuclei and the Cytotox-green-dye-positive
nuclei were quantified using the IncuCyte analysis software. The per-
centage of dead cells was calculated by dividing the Cytotox green

dye positive nuclei by the total number of nuclei multiplied by 100.
In Extended Data Figs. 6e and 13d, cells were transfected with siRNA
48 h before cell death measurements. Then, 24 h after transfection,
the medium was changed to medium containing 250 nM IncuCyte
Cytotox green dye (Sartorius, 4633) or IncuCyte Cytotox red dye
(Sartorius, 4632). Then, 48 h after transfection, nuclei were stained
with Hoechst (1 µg ml$^{-1}$), images were taken using the Revolve fluo-
rescence microscope and analysed with CellProfiler v.4.2.1 using a
customized pipeline. Total nuclei and Cytotox-dye-positive nuclei
were segmented using an integrated intensity-based object detection
module. The percentage of dead cells was calculated by dividing the
number of Cytotox-dye-positive nuclei by the total number of nuclei
multiplied by 100.

### Retroviral and lentiviral transduction
Lentiviral and retroviral particles were produced by the laboratory.
Production of lentivirus was performed as described previously[56].
In brief, HEK293T (ATCC, CRL-11268) cells were transfected with 7 µg of
DNA using Lenti-X Packaging Single-Shot system (Clontech, 631276).
Then, 48 h after transfection, viral supernatant was collected, sup-
plemented with serum and used for transduction in the presence of
Lenti-Blast (Oz Biosciences, LB00500). To produce retrovirus, Phoenix
cells were transfected with 20 µg of DNA using 100 µM of chloroquine.
Then, 5 h after transfection, fresh medium was added. The viral superna-
tant was collected 24 h later and used for transduction in the presence
of polybrene 4 µg ml$^{-1}$. Then, 48 h after infection, cells were washed
and selected with 1 µg ml$^{-1}$ puromycin, 600 µg ml$^{-1}$ G418, or 90 µg ml$^{-1}$
hygromycin. IMR90$^{E6E7}$ and WI38$^{SV40LT}$ fibroblasts were subjected to
long-term culturing under antibiotic selection.

### Transfections
DNA transfections were performed using the Lipofectamine 3000
kit (Thermo Fisher Scientific, 1857482) according to the manufac-
turer's instructions. siRNA transfections were performed using the
Lipofectamine RNAiMAX kit (Thermo Fisher Scientific, 13778030)
according to the manufacturer's instructions.

### Western blotting
Western blots were performed as described previously[59]. In brief, cells
were lysed in NuPage LDS sample buffer (Invitrogen, NP0007) at 1 × 10$^4$
cells per µl. Proteins were resolved using NuPage Bis-Tris gel electro-
phoresis (Invitrogen, NP0343, NP0321 or WG1402) and transferred
to nitrocellulose membranes (Amersham, 10600037). Secondary
antibodies were peroxidase-conjugated anti-mouse IgG (Amersham,
NXA931V) or anti-rabbit IgG (Amersham, NA934V). Peroxidase activity
was detected using an ECL kit (Prometheus, 20-302B) and the Syngene
G-Box imager. Primary antibodies are listed below.

### Metaphase spreads for quantification of chromosome fusions
Metaphase spread preparation was performed as described[60]. In brief,
cells were treated with 0.2 µg ml$^{-1}$ of Colcemid (Gibco, 15212-012) for
3 h, collected and incubated with hypotonic solution (75 mM of KCl) for
7 min at 37 °C. The cell suspension was centrifuged, washed in fixative
solution (3:1 methanol:glacial acetic acid), dropped onto superfrost
microscope slides (VWR, 48311-703) and air dried overnight. For tel-
omere staining, cells were fixed in 4% formaldehyde in PBS for 10 min,
dehydrated in a series of ethanol baths (70%, 90% and 100%) for 3 min
each and air-dried for 20 min. Cells were then covered in 0.3 ng µl$^{-1}$ of
the telomeric (PNA Bio, F1004) PNA probes diluted in hybridization
solution (70% deionized formamide, 0.25% blocking reagent (NEN),
10 mM Tris pH 7.5). The samples were heated for 5 min at 80 °C, incu-
bated 2 h at room temperature, washed twice in 70% formamide and
10 mM Tris-HCl pH 7.5 and three times in 50 mM Tris-HCl pH 7.5, 150 mM
NaCl and 0.08% Tween-20. Slides were finally mounted in ProLong
Diamond with DAPI (Invitrogen, P36971).

## Immunofluorescence-FISH analysis of interphase nuclei and metaphase spreads

For interphase nuclei, cells were seeded onto glass coverslips 24 h before the experiment as described previously[60]. Cells were washed in PBS, fixed in 4% formaldehyde in PBS for 10 min and permeabilized in 0.1% Triton X-100 in PBS for 10 min. For metaphase spreads, cells were treated with 20 ng ml$^{-1}$ of Colcemid (Gibco, 15212-012) for 1 h, collected and incubated in hypotonic solution (27 mM KCl, 6.5 mM tri-sodium citrate) for 5 min. The cell suspension was cytocentrifuged, fixed in 4% formaldehyde in PBS for 10 min and permeabilized in KCM buffer (120 mM KCl, 20 mM NaCl, 10 mM Tris pH 7.5, 0.1% Triton X-100) for 10 min. In both settings, the samples were incubated in blocking buffer (20 mM Tris pH 7.5, 2% BSA, 0.2% fish gelatin, 150 mM NaCl, 0.1% Triton X-100, 0.1% sodium azide and 100 µg ml$^{-1}$ RNase A) for 1 h at 37 °C. Cells were incubated with the primary antibody (γH2AX) for 2 h, washed in PBS and incubated with secondary antibody for 1 h at room temperature. The secondary antibodies used were AlexaFluor 568 anti-IgG Mouse (Thermo Fisher Scientific, A-11004) or AlexaFluor 647 anti-IgG Mouse (Thermo Fisher Scientific, A-21235). The samples were finally fixed in 4% formaldehyde in PBS before fluorescence in situ hybridization (FISH) analysis.

## Immunofluorescence

Cells were seeded onto glass coverslips 24 h before the experiment. Cells were fixed in 4% paraformadehyde in PBS for 10 min, washed in PBS and incubated in blocking solution of 5% BSA in PBS at room temperature. Cells were then incubated with primary antibodies for 2 h, washed in PBS and incubated with secondary antibodies for 1 h at room temperature. The secondary antibodies used were AlexaFluor 488 goat anti-rabbit IgG (H+L) (Thermo Fisher Scientific, A-11034) and AlexaFluor 568 goat anti-Mouse IgG (H+L) (Thermo Fisher Scientific, A-11004). The samples were finally washed in PBS and mounted in ProLong Diamond with DAPI (Invitrogen, P36971). For MitoTracker staining, MitoTracker Red CMXRos (Thermo Fisher Scientific, M7512) was added to the culture medium at a final concentration of 200 nM and incubated for 20 min under tissue culture conditions. After staining, cells were washed twice in prewarmed PBS, then fixed in paraformaldehyde 4% PBS for 10 min and washed in PBS. Imaging was performed using the Zeiss LSM 880 with the Airyscan microscope. ZEN (Zeiss) and ImageJ were used for image analysis.

## Crystal violet assay for determining cell viability

Cells before crisis were seeded at low density (476 cells per cm$^2$) and kept at 37 °C for 10 days before fixing with 4% paraformaldehyde in PBS. Cells were then stained with 0.05% crystal violet in distilled water for 20 min.

## fRIP

fRIP–seq was performed as previously described with minor modifications[61]. In brief, IMR90$^{E6E7}$ cells were cross-linked with 0.1% formaldehyde in PBS for 10 min at room temperature and unreacted formaldehyde was then neutralized with 125 mM glycine for 5 min at room temperature. Cells were washed twice with ice-cold PBS, collected by trypsinization and the cell pellets were resuspended in RIPA buffer (50 mM Tris pH 8, 150 mM KCl, 0.1% SDS, 1% Triton X-100, 5 mM EDTA, 0.5% sodium deoxycholate, 0.5 mM DTT (added fresh), protease inhibitor cocktail (Roche, 4693159001), 100 U ml$^{-1}$ RNasin Ribonuclease Inhibitor (Promega, N251B)), and incubated for 20 min at 4 °C under slow rotation. The cell lysates were centrifuged at maximum speed at 4 °C for 10 min, and the supernatants were collected and diluted in equal volumes of freshly made fRIP binding/wash buffer (150 mM KCl, 25 mM Tris pH 7.5, 5 mM EDTA, 0.5% NP-40, 0.5 mM DTT (added fresh), protease inhibitor cocktail (Roche, 4693159001), 100 U ml$^{-1}$ RNasin Ribonuclease Inhibitor (Promega, N251B)). Diluted lysates were first precleared with Dynabeads Protein G (Invitrogen, 10004D)

at a concentration of 25 µl of beads per 5 million cells for 30 min at 4 °C under slow rotation, and then incubated with 10 µg anti-Flag M2 antibodies (Sigma-Aldrich, F3165) previously coupled to 40 µl protein G Dynabeads (per each 5 million cells) for 2 h at 4 °C under slow rotation. The beads were washed twice with fRIP binding/wash buffer and protein−RNA cross-links were reversed by resuspending the beads in 56 µl of RNase-free water + 33 µl of 3× reverse-cross-linking buffer (3× PBS (without Ca$^{2+}$ and Mg$^{2+}$), 6% N-lauroyl sarcosine, 30 mM EDTA, 15 mM DTT (added fresh), 10 µl of 20 mg ml$^{-1}$ proteinase K (Millipore, 70663) and 1 µl of RNasin Ribonuclease Inhibitor. Protein degradation and reverse-cross-linking was performed for 1 h at 42 °C, followed by another hour at 55 °C. RNA was recovered by resuspending the beads and reaction buffer in 1 ml TRIzol Reagent (Invitrogen, 15596018) and purified using the Direct-zol RNA Microprep kit as recommended by manufacturer (Zymo Research, R2061).

## Library generation and sequencing for fRIP−seq

RNA-seq libraries were prepared with immunoprecipitated RNA using the TruSeq stranded total RNA sample preparation kit according to the manufacturer's protocol (Illumina). RNA-seq libraries were multiplexed, normalized and pooled for sequencing. The libraries were sequenced on the MiniSeq system (Illumina) with paired-end 37 bp reads. Image analysis and base calling were performed using Illumina CASAVA v.1.8.2. on the MiniSeq system and sequenced reads were quality-tested using FASTQC. Read pairs were mapped individually to the most complete assembly available of human subtelomeres (http://www.wistar.org/lab/harold-c-riethman-phd/page/subtelomere-assemblies) using STAR v.2.5.3a allowing up to 101 mapping locations[62]. Primary alignments were assigned randomly, and each read is represented by one mapping location. Library sizes were normalized to $1 \times 10^7$ for comparison and the $\log_2$ ratio of enrichment of immunoprecipitates versus input was calculated at each base in the subtelomere using HOMER v.4.10 and the assembly available of human subtelomeres[63]. Visualizations were performed using R v.3.6.1 (R Core Team). Graphical packages (Gviz v.1.28.3, rtracklayer v.1.44.2, gridExtra v.2.3, ggplot2 v.3.3.2) were used to visualize data.

## RT−qPCR

Total RNA was isolated using TRIzol (Invitrogen, 15596018) and purified using the RNAeasy Mini Kit (Qiagen, 74106) according to the manufacturer's instructions. Genomic DNA was eliminated by double digestion with DNase I (RNase-free DNase Set, Qiagen, 79256). For RT−qPCR, 3.5 µg of RNA was reverse-transcribed either with random hexamers for measuring ISGs or with 2 µM of *telC*- and *GAPDH*-specific RT primers[64] using the SuperScript III First-Strand Synthesis System (Thermo Fisher Scientific, 18080-051). qPCR was performed on the CFX384 Touch Real-Time PCR Detection System (BioRad). Reactions were run in triplicates with Power SYBR Green Master Mix (Applied Biosystems, Thermo Fisher Scientific, 4367659) in a total volume of 10 µl with standard cycling conditions. Relative gene expression was normalized using *GAPDH* as a housekeeping gene and calculated using the Comparative CT Method ($\Delta\Delta C_T$ method). The primers are listed below.

## RNA-dot blot

Total RNA (10 µg) was blotted onto a positively charged nitrocellulose membrane (GE, RPNBL/02/10). For fRIP, 200 ng of RNA from IP or input samples were used. For RNase controls, RNA was incubated with RNase A (Invitrogen, 12091-039) at 37 °C for 1 h. RNA was fixed by ultraviolet cross-linking (Stratalinker, 2400) and TERRA was detected by hybridizing overnight at 55 °C with a Church mix containing telomeric repeat probes generated by [CCCTAA]$^3$-primed Klenow labelling of an 800 bp [TTAGGG]$^n$ fragment in the presence of [α$^{32}$P]dCTP. After hybridization, the membrane was washed twice in 2× SSC and 0.1% SDS for 10 min at room temperature and then once for 10 min at 50 °C. The radioactive signal was detected using the Typhoon FLA 9000 imager

(GE Healthcare). After the signal acquisition, membranes were stripped and rehybridized at 50 °C overnight with [32]P-dCTP-labelled probes targeting *GAPDH* transcripts. Signals were measured using ImageJ.

### TRF analysis
TRF analysis was performed as previously described[65]. In brief, genomic DNA was isolated by phenol–chloroform extraction and digested with AluI and MboI overnight. A total of 4 μg of digested gDNA was separated on 0.7% agarose gel at 40 V and transferred to a positively charged Nylon membrane (Amersham, RPN203B). After cross-linking the DNA and prehybridization (5× SSC. 0.1% *N*-lauroylsarcosine sodium salt solution, 0.04% SDS) for 2 h at 65 °C, the membrane was incubated with digoxigenin-labelled TelG probe diluted in hybridization buffer (1.3 nM final concentration) overnight at 65 °C. Digoxigenin-labelled TelG probe was generated as previously described[66]. Then, the membrane was washed three times with wash buffer 1 (2× SSC, 0.1% SDS), one time with wash buffer 2 (2× SSC) for 15 min each and blocked in freshly prepared blocking solution (100 mM maleic acid, 150 mM NaCl, pH 7.5, 1% (w/v) blocking reagent (Roche, 11096176001)) for 30 min. Next, the membrane was incubated for 30 min in anti-digoxigenin-AP antibodies (Roche, 11093274910) diluted in blocking solution, washed twice in wash buffer 3 (100 mM maleic acid, 150 mM NaCl, pH 7.5, 0.3% (v/v) Tween-20) for 15 min each and equilibrated in AP buffer (100 mM Tris, 100 mM NaCl, pH 9.5) for 2 min. Digoxigenin-labelled telomeric DNA was detected using CDP-star ready to use (Roche, 12041677001) solution.

### Genome-wide CRISPR screen
The human Brunello CRISPR knockout pooled library was obtained from D. Root and J. Doench through Addgene (73179-LV)[67]. This ready-to-use lentiviral library has 77,441 gRNAs, targeting 19,114 protein-coding genes, with approximately 4 sgRNAs per gene. For adequate representation of each sgRNA, a total of 100 million pre-crisis (PD100) IMR90[E6E7] fibroblasts were transduced with the lentiviral library at a multiplicity of infection of 0.5. Transductions were performed in six-well plates (3 million cells per well) in medium containing 4 μg ml[−1] polybrene while centrifuging at 1,000 rcf for 1 h at 33 °C (spinfection). The next day, cells were transferred to Cell Factory System (Thermo Fisher Scientific, 140360), and puromycin-containing medium (1 μg ml[−1]) was added for 7 days to eliminate uninfected cells and achieve genome-edited cell pools. After selection, cells were pooled together and divided into two technical replicates of 30 million cells each corresponding to the library baseline control at day 0. Another 30 million cells were replated into one Cell Factory System and positive selection was performed by growing cells for additional 15 days, at which point crisis-associated cell death was frequent. Two technical replicates of 30 million cells each were prepared by collecting cells at day 15, and their genomic DNA was extracted using a modified version of QIAGEN's DNeasy Blood and Tissue Kit provided by the FLI-Seq Library Prep for CRISPR kit (Eclipse Bioinnovations). Genomic DNA was also extracted from the baseline count control sample at day 0. DNA fragments containing the sgRNA sequences were first captured from sheared gDNA, amplified by PCR using the FLI-Seq Library Prep for CRISPR kit (Eclipse Bioinnovations) and processed for next-generation sequencing. CRISPR libraries were multiplexed, normalized and pooled for sequencing. To compensate for low base diversity in CRISPR libraries, high-diversity libraries or PhiX Control v3 Library were spiked in for sequencing and the libraries were sequenced on the HiSeq 2500 system (Illumina) as single reads. Image analysis and base calling were performed using Illumina CASAVA v.1.8.2 on the HiSeq 2500 system and sequenced reads were quality-tested using FASTQC. Fold changes of sgRNA read counts before (day 0) and after (day 15) enrichment were calculated with the PinAPL-Py software and hits were defined as genes with a read count ratio in the survival pool to the baseline control log$_2$[fold change] ≥ 2 in both replicates. GO enrichment analysis was performed using the WEB-based Gene Set

Analysis Toolkit (WebGestalt). The top 20 GO terms with an FDR value of <0.05 were considered to be statistically significant and visualized using the ClueGO and ggplot2 R packages.

### Whole-transcriptome analysis
Total RNA was isolated using TRIzol (Invitrogen, 15596018) and purified using the RNAeasy Mini Kit (Qiagen, 74106) according to the manufacturer's instructions. Genomic DNA was eliminated by double digestion with DNase I (RNase-free DNase Set, Qiagen, 79256). The quality of the isolated total RNA was assessed using the Agilent TapeStation 4200 and RNA-seq libraries were prepared with 500 ng total RNA using the TruSeq stranded mRNA sample preparation kit according to the manufacturer's protocol (Illumina). RNA-seq libraries were multiplexed, normalized and pooled for sequencing. The libraries were sequenced on the HiSeq 4000 system (Illumina) as 50 bp single reads or the NextSeq 500 system (Illumina) as 75 bp single reads. Image analysis and base calling were performed using Illumina CASAVA v.1.8.2 on the HiSeq 4000 system and sequenced reads were quality-tested using FASTQC. Sequenced reads were quality-tested using FASTQC (v.0.11.8)[68] and mapped to the hg19 human genome using the STAR aligner (v.2.5.3a)[62] with the default parameters. Raw or transcripts per kilobase million (TPM) gene expression was quantified across all the exons of RefSeq genes with analyzeRepeats.pl in HOMER (v.4.11.1)[63], which used the top-expressed isoform as proxy for gene expression. Differential gene expression was performed on the raw gene counts with the R package, DESeq2 (v.1.24.0)[69], using replicates to compute within-group dispersion. Differentially expressed genes were defined as having a FDR < 0.05 and a log$_2$[fold change] > 0.585 (-1.5 fold) when comparing two experimental conditions. GO enrichment analysis was performed using the WEB-based Gene Set Analysis Toolkit" (WebGestalt) and the R package clusterProfiler. The top 20 GO terms with an FDR value of <0.05 were considered to be statistically significant and visualized using the ClueGO and ggplot2 R packages. Heat maps and Venn diagrams were generated using the R packages ComplexHeatmap and VennDiagram, respectively.

### ELISA assay
One million cells were seeded in a 10 cm plate for 24 h. Conditioned cell culture medium was collected and centrifuged at 14,000 rpm for 10 min at 4 °C to remove cellular debris. Three technical replicates of 50 μl each were collected for the assay. For preparation of standards, GlutaMax-DMEM (Gibco, 10569-010) supplemented with 0.1 mM non-essential amino acids (Corning, 25-025-Cl) and 15% fetal bovine serum (Thermo Fisher Scientific, SH3007103) was used. IFNβ secretion in cell culture supernatant was quantified using the IFNβ enzyme-linked immunosorbent assay (ELISA) kit (R&D Systems Human IFNβ Quantikine ELISA Kit, DIFNB0) according to the manufacturer's instruction. Supernatants from the treated cells were collected and incubated in IFNβ ELISA kit for 2 h at room temperature and washed three times with wash buffer. The optical density at 450 nm for each sample was measured with a microplate reader and data obtained were plotted against a four-parameter logistic standard curve to determine the concentration of IFNβ.

### Chemical reagents
The reagents were as follows: recombinant human IFN-beta protein: R&D Systems, 8499-IF-010/CF; poly(dA:dT): InvivoGen, tlrl-patn; poly(I:C): InvivoGen, tlrl-pic; 2′,3′-dideoxycytidine: Sigma-Aldrich, D5782; deoxyribonucleic acid sodium salt from herring testes (HT-DNA): Sigma-Aldrich, D6898; MitoTracker: Invitrogen, M22426; IncuCyte Nuclight Rapid red dye: Sartorius, 4717; IncuCyte Cytotox red dye: Sartorius, 4632; IncuCyte Cytotox green dye: Sartorius, 4633; doxycycline: Sigma-Aldrich, D9891; staurosporine: Cell Signaling Technology, 9953; recombinant human TNF-alpha protein: R&D Systems, 210-TA; BV6: Selleckchem, S7597; z-VAD-FMK: Cell Signaling Technology, 60332;

LPS: Cell Signaling Technology, 14011; nigericin: Cayman Chemical, 11437; Hoechst 33342: Thermo Fisher Scientific, 62249.

## Statistical analysis

Statistical analysis was performed using Prism 9. Comparisons between two groups were performed using unpaired two-tailed Student's $t$-tests. Multiple comparisons were performed using one-way ANOVA followed by Tukey's or Dunnett's multiple-comparisons test. For all representative findings, two or three independent experiments were performed, and similar results were obtained. Significance in all figures is denoted as follows: *$P < 0.05$, **$P < 0.01$, ***$P < 0.001$.

Exact $P$ values are as follows.

Figure 1c: 36 h after incubation with Cytotox. Vector versus WT (<0.0001); vector versus mRHIM2 (<0.0001).

Figure 1d: IFNB1: vector versus WT (<0.0001); vector versus mZα2 (0.9775); vector versus mRHIM1 (0.9554); vector versus mRHIM2 (<0.0001). CCL5: vector versus WT (<0.0001); vector versus mZα2 (0.101); vector versus mRHIM1 (0.9799); vector versus mRHIM2 (<0.0001). CXCL10: vector versus WT (<0.0001); vector versus mZα2 (0.8118); vector versus mRHIM1 (0.9958); vector versus mRHIM2 (<0.0001). IFI44: vector versus WT (<0.0001); vector versus mZα2 (0.9426); vector versus mRHIM1 (0.9997); vector versus mRHIM2 (<0.0001). ISG15-vector versus WT (<0.0001); vector versus mZα2 (0.1726); vector versus mRHIM1 (0.8483); vector versus mRHIM2 (<0.0001).

Figure 2b: IFNB1: shScramble versus sh*TRF2* (<0.0001); sh*TRF2*-siCtrl versus sh*TRF2*-si*ZBP1* 1 (<0.0001); sh*TRF2*-siCtrl versus sh*TRF2*-si*ZBP1* 2 (<0.0001). CCL5: shScramble versus sh*TRF2* (<0.0001); sh*TRF2*-siCtrl versus sh*TRF2*-si*ZBP1* 1 (<0.0001); sh*TRF2*-siCtrl versus sh*TRF2*-si*ZBP1* 2 (<0.0001). CXCL10: shScramble versus sh*TRF2* (<0.0001); sh*TRF2*-siCtrl versus sh*TRF2*-si*ZBP1* 1 (<0.0001); sh*TRF2*-siCtrl versus sh*TRF2*-si*ZBP1* 2 (<0.0001). IFI44: shScramble versus sh*TRF2* (<0.0001); sh*TRF2*-siCtrl versus sh*TRF2*-si*ZBP1* 1 (<0.0001); sh*TRF2*-siCtrl versus sh*TRF2*-si*ZBP1* 2 (<0.0001). ISG15-shScramble versus sh*TRF2* (<0.0001); sh*TRF2*-siCtrl versus sh*TRF2*-si*ZBP1* 1 (<0.0001); sh*TRF2*-siCtrl versus sh*TRF2*-si*ZBP1* 2 (<0.0001).

Figure 3a: 9p, growing versus crisis (<0.0001). 13q, growing versus crisis (<0.0001). 15q, growing versus crisis (<0.0001). Xp, growing versus crisis (<0.0001). Xq, growing versus crisis (<0.0001). Figure 3b: 9p, shScramble versus sh*TRF2* (<0.0001). 13q, shScramble versus sh*TRF2* (<0.0001). 15q, shScramble versus sh*TRF2* (<0.0001). Xp, shScramble versus sh*TRF2* (<0.0001). Xq, shScramble versus sh*TRF2* (0.0015).

Figure 3f: 28 h after incubation with Cytotox. ZBP1(S)(WT)–Flag TRF1(ΔN) versus ZBP1(S)(WT)–Flag VP16–TRF1(ΔN) (<0.0001).

Figure 4a: PD45 versus PD79 (<0.0001); PD45 versus PD99 (<0.0001); PD45 versus PD103 (<0.0001); PD45 versus PD106 (<0.0001); PD35 versus PD50 (0.0043); PD35 versus PD62 (<0.0001); PD35 versus PD71 (<0.0001); PD35 versus PD86 (<0.0001).

Figure 4c: CCL5: PD40 vector versus PD40 WT (>0.9999). PD40 vector versus PD40 mZα (>0.9999); PD40 vector versus PD40 mRHIM1 (>0.9999); PD40 vector versus PD40 mRHIM2 (>0.9999); PD95 vector versus PD95 WT (<0.0001); PD95 vector versus PD95 mZα (0.8488); PD95 vector versus PD95 mRHIM1 (0.9998); PD95 vector versus PD95 mRHIM2 (<0.0001). CXCL10: PD40 vector versus PD40 WT (>0.9999); PD40 vector versus PD40 mZα (>0.9999); PD40 vector versus PD40 mRHIM1 (>0.9999); PD40 vector versus PD40 mRHIM2 (>0.9999); PD95 vector versus PD95 WT (0.0005); PD95 vector versus PD95 mZα (0.9766); PD95 vector versus PD95 mRHIM1 (>0.9999); PD95 vector versus PD95 mRHIM2 (<0.0001). IFI44: PD40 vector versus PD40 WT (>0.9999); PD40 vector versus PD40 mZα (>0.9999); PD40 vector versus PD40 mRHIM1 (0.9965); PD40 vector versus PD40 mRHIM2 (>0.9999); PD95 vector versus PD95 WT (<0.0001); PD95 vector versus PD95 mZα (>0.9566); PD95 vector versus PD95 mRHIM1 (0.0338); PD95 vector versus PD95 mRHIM2 (<0.0001).

Figure 4e: CCL5: Flag–ZBP1(S)–FIS1 versus Flag–ZBP1(S) (<0.0001); vector versus Flag–ZBP1(S)–FIS1 (<0.0001). CXCL10: Flag–ZBP1(S)–FIS1 versus Flag–ZBP1(S) (<0.0001); vector versus Flag–ZBP1(S)–FIS1 (<0.0001). IFI44: Flag–ZBP1(S)–FIS1 versus Flag–ZBP1(S) (<0.0001); vector versus Flag–ZBP1(S)–FIS1 (<0.0001).

Figure 4g: CCL5: vector siCtrl versus vector si*MAVS* (0.9002); Flag–ZBP1(S) siCtrl versus Flag–ZBP1(S) si*MAVS* (0.3629); Flag–ZBP1(S)–FIS1 siCtrl versus Flag–ZBP1(S)–FIS1 si*MAVS* (<0.0001). CXCL10: vector siCtrl versus vector si*MAVS* (0.9537). Flag–ZBP1(S) siCtrl versus Flag–ZBP1(S) si*MAVS* (0.0001). Flag–ZBP1(S)–FIS1 siCtrl versus Flag–ZBP1(S)–FIS1 si*MAVS* (<0.0001). IFI44: vector siCtrl versus vector si*MAVS* (>0.9999). Flag–ZBP1(S) siCtrl versus Flag–ZBP1(S) si*MAVS* (0.9966). Flag–ZBP1(S)–FIS1 siCtrl versus Flag–ZBP1(S)–FIS1 si*MAVS* (<0.0001).

Extended Data Fig. 3d: IMR90$^{E6E7}$, 24 h after incubation with Cytotox. sgGFP versus sgZBP1 1 (<0.0001). sgGFP versus sgZBP1 2 (<0.0001); sgGFP versus sgZBP1 3 (<0.0001); sgLUC versus sgZBP1 1 (<0.0001); sgLUC versus sgZBP1 2 (<0.0001); sgLUC versus sgZBP1 3 (<0.0001). WI38$^{SV40LT}$, 24 h after incubation with Cytotox. sgGFP versus sgZBP1 1 (<0.0001); sgGFP versus sgZBP1 2 (<0.0001); sgGFP versus sgZBP1 3 (<0.0001); sgLUC versus sgZBP1 1 (<0.0001); sgLUC versus sgZBP1 2 (<0.0001); sgLUC versus sgZBP1 3 (<0.0001).

Extended Data Fig. 3e: IMR90$^{E6E7}$: sgLUC versus sgGFP (0.0668); sgLUC versus sgZBP1 1 (<0.0001); sgLUC versus sgZBP1 2 (<0.0001); sgLUC versus sgZBP1 3 (<0.0001). WI38$^{SV40LT}$: sgLUC versus sgGFP (0.1293); sgLUC versus sgZBP1 1 (<0.0001); sgLUC versus sgZBP1 2 (<0.0001); sgLUC versus sgZBP1 3 (<0.0001).

Extended Data Fig. 3f: IMR90$^{E6E7}$: sgLUC versus sgGFP (0.9995); sgLUC versus sgZBP1 1 (0.0002); sgLUC versus sgZBP1 2 (<0.0001); sgLUC versus sgZBP1 3 (0.0004). WI38$^{SV40LT}$: sgLUC versus sgGFP (0.7845); sgLUC versus sgZBP1 1 (0.0018); sgLUC versus sgZBP1 2 (<0.0001). sgLUC versus sgZBP1 3 (0.0011).

Extended Data Fig. 4b: IMR90$^{E6E7}$: growing versus crisis (<0.0001). WI38$^{SV40LT}$: growing versus crisis (<0.0001).

Extended Data Fig. 4e: experiment 1: siCtrl versus si*IFNAR1* ($1.39178 \times 10^{-5}$); siCtrl versus si*IFNAR2* ($8.05855 \times 10^{-5}$); siCtrl versus si*CGAS* ($3.65474 \times 10^{-5}$); siCtrl versus si*AIM2* (0.57384863); siCtrl versus si*TLR3* (0.028940902); siCtrl versus si*TLR7* (0.130633803); siCtrl versus si*TLR9* (0.481604543); siCtrl versus si*RIG-I* (0.067963804); siCtrl versus si*MDA5* (0.078645098); siCtrl versus RIG-I + si*MDA5* (0.075426998); siCtrl versus si*STING* (0.000445511); siCtrl versus si*PYCARD* (0.272857261); siCtrl versus si*MyD88* (0.16111072); siCtrl versus si*TRIF* (0.138938302); siCtrl versus si*MAVS* ($3.61752 \times 10^{-5}$). Experiment 2: siCtrl versus si*IFNAR1* ($1.48978 \times 10^{-6}$); siCtrl versus si*IFNAR2* ($1.4765 \times 10^{-5}$); siCtrl versus si*CGAS* ($1.40243 \times 10^{-6}$); siCtrl versus si*AIM2* (0.081532291); siCtrl versus si*TLR3* (0.244271443); siCtrl versus si*TLR7* (0.413559022); siCtrl versus si*TLR9* (0.1970414); siCtrl versus si*RIG-I* (0.015458711); siCtrl versus si*MDA5* (0.027374775); siCtrl versus RIG-I + si*MDA5* (0.857465154); siCtrl versus si*STING* ($9.9925 \times 10^{-5}$); siCtrl versus si*PYCARD* (0.28638452); siCtrl versus si*MyD88* (0.73179998); siCtrl versus si*TRIF* (0.0188581); siCtrl versus si*MAVS* ($7.6601 \times 10^{-6}$). Experiment 3: siCtrl versus si*IFNAR1* ($1.23834 \times 10^{-5}$); siCtrl versus si*IFNAR2* ($6.7646 \times 10^{-5}$); siCtrl versus si*CGAS* ($1.04549 \times 10^{-6}$); siCtrl versus si*AIM2* (0.05401101); siCtrl versus si*TLR3* (0.16170163); siCtrl versus si*TLR7* (0.01702073); siCtrl versus si*TLR9* (0.0189054); siCtrl versus si*RIG-I* (0.00405314); siCtrl versus si*MDA5* (0.373743841); siCtrl versus RIG-I + si*MDA5* (0.024471461); siCtrl versus si*STING* (0.00368923); siCtrl versus si*PYCARD* (0.08524613); siCtrl versus si*MyD88* (0.14064815); siCtrl versus si*TRIF* (0.12424426); siCtrl versus si*MAVS* (0.008386476).

Extended Data Fig. 5d: IFNB1: vector versus WT (<0.0001); vector versus ΔZα2 (0.2671); vector versus ΔZα2ΔRHIM1ΔRHIM2 (0.1795); vector versus ΔC (<0.0001); vector versus ΔRHIM1 (0.6671); vector versus ΔRHIM2 (<0.0001); vector versus ΔRHIM1ΔRHIM2 (0.0151). CCL5: vector versus WT (<0.0001); vector versus ΔZα2 (0.0367); vector

versus ΔZα2ΔRHIM1ΔRHIM2 (0.978); vector versus ΔC (<0.0001); vector versus ΔRHIM1 (0.0675); vector versus ΔRHIM2 (<0.0001); vector versus ΔRHIM1ΔRHIM2 (0.0816). CXCL10: vector versus WT (<0.0001); vector versus ΔZα2 (0.1945); vector versus ΔZα2ΔRHIM1ΔRHIM2 (0.9998); vector versus ΔC (<0.0001); vector versus ΔRHIM1 (0.1365); vector versus ΔRHIM2 (<0.0001); vector versus ΔRHIM1ΔRHIM2 (0.1988). IFI44: vector versus WT (<0.0001); vector versus ΔZα2 (0.0294); vector versus ΔZα2ΔRHIM1ΔRHIM2 (0.9813); vector versus ΔC (<0.0001); vector versus ΔRHIM1 (0.0287); vector versus ΔRHIM2 (<0.0001); vector versus ΔRHIM1ΔRHIM2 (0.2269). ISG15: vector versus WT (<0.0001); vector versus ΔZα2 (0.0003); vector versus ΔZα2ΔRHIM1ΔRHIM2 (0.0242); vector versus ΔC (<0.0001); vector versus ΔRHIM1 (0.0005); vector versus ΔRHIM2 (<0.0001); vector versus ΔRHIM1ΔRHIM2 (0.0001).

Extended Data Fig. 6e: vector versus ZBP1(S)–Flag (0.0082); siCtrl versus si*ZBP1* (<0.0001); siCtrl versus si*RIPK3* (0.9942); siCtrl versus siMLK (>0.9999); siCtrl versus siRIPK1 (>0.9999); siCtrl versus siFADD (0.9425); siCtrl versus siCASP8 (>0.9999); siCtrl versus siNLRP3 (>0.9999); siCtrl versus siPYCARD (>0.9999); siCtrl versus siCASP1 (0.9790); siCtrl versus siIFNAR1 (<0.0001); siCtrl versus siIFNAR2 (<0.0001); siCtrl versus siSTAT1 (<0.0001); siCtrl versus siSTAT2 (<0.0001); siCtrl versus siIRF9 (<0.0001); siCtrl versus siATG5 (<0.0001); siCtrl versus siATG7 (<0.0001); siCtrl versus siATG12 (<0.0001).

Extended Data Fig. 6g: mock versus shScramble day 3 (0.9934); mock versus shScramble day 6 (0.9996); mock versus shScramble day 9 (0.9994); mock versus shScramble day 12 (0.9996); mock versus sh*TRF2* day 3 (<0.0001); mock versus sh*TRF2* day 6 (<0.0001); mock versus sh*TRF2* day 9 (<0.0001); mock versus sh*TRF2* day 12 (<0.0001).

Extended Data Figure 6h: mock versus shScramble day 3 (>0.9999); mock versus shScramble day 6 (>0.9999); mock versus shScramble day 9 (>0.9999); mock versus shScramble day 12 (>0.9999); mock versus sh*TRF2* day 3 (<0.0001); mock versus sh*TRF2* day 6 (<0.0001); mock versus sh*TRF2* day 9 (<0.0001); mock versus sh*TRF2* day 12 (<0.0001).

Extended Data Figure 6j: siCtrl versus si*ZBP1* 1 (0.9038); siCtrl versus si*ZBP1* 2 (0.2923).

Extended Data Fig. 7c: IMR90$^{E6E7}$: PD42 versus PD45 (>0.9999); PD42 versus PD46 (0.9999); PD42 versus PD51 (0.9999); PD42 versus PD55 (0.9997); PD42 versus PD58 (>0.9999); PD42 versus PD62 (0.9998); PD42 versus PD69 (0.9996); PD42 versus PD73 (>0.9999); PD42 versus PD85 (0.9996); PD42 versus PD87 (0.9998); PD42 versus PD90 (0.9998); PD42 versus PD99 (0.9864); PD42 versus PD100 (0.0891); PD42 versus PD101 (0.0003); PD42 versus PD103 (<0.0001); PD42 versus PD105 (<0.0001); PD42 versus PD106 (<0.0001). IMR90$^{E6E7}$, hTERT: PD61 versus PD74 (0.8616); PD61 versus PD89 (0.9996); PD61 versus PD95 (0.9995); PD61 versus PD105 (>0.9999); PD61 versus PD108 (0.9882). WI38$^{SV40LT}$: PD35 versus PD38 (>0.9999); PD35 versus PD41 (>0.9999); PD35 versus PD44 (0.9997); PD35 versus PD47 (>0.9999); PD35 versus PD50 (0.9999); PD35 versus PD53 (>0.9999); PD35 versus PD58 (0.9998); PD35 versus PD62 (0.9998); PD35 versus PD67 (0.9998); PD35 versus PD69 (0.9994); PD35 versus PD71 (0.9997); PD35 versus PD79 (0.9994). PD35 versus PD82 (0.9990); PD42 versus PD86 (<0.0001); PD42 versus PD87 (<0.0001); PD42 versus PD89 (<0.0001); PD42 versus PD90 (<0.0001). WI38$^{SV40LT}$, hTERT: PD50 versus PD63 (0.9576); PD50 versus PD74 (0.9576); PD50 versus PD77 (0.9796); PD50 versus PD83 (>0.9999); PD50 versus PD92 (0.9576).

Extended Data Fig. 8a: growing versus crisis (<0.0001).

Extended Data Fig. 8b: shScramble versus sh*TRF2* (<0.0001).

Extended Data Fig. 9c: 9p: ZBP1(S)(WT)–Flag TRF1(ΔN) versus ZBP1(S)(WT)–Flag VP16–TRF1(ΔN) (<0.0001); ZBP1(S)(ΔZα2)–Flag TRF1(ΔN) versus ZBP1(S)(ΔZα2)–Flag VP16–TRF1(ΔN) (<0.0001). 13q: ZBP1(S)(WT)–Flag TRF1(ΔN) versus ZBP1(S)(WT)–Flag VP16–TRF1(ΔN) (0.0002); ZBP1(S)(ΔZα2)–Flag TRF1(ΔN) versus ZBP1(S)(ΔZα2)–Flag VP16–TRF1(ΔN) (0.0005). 15q: ZBP1(S)(WT)–Flag TRF1(ΔN) versus ZBP1(S)(WT)–Flag VP16–TRF1(ΔN) (<0.0001); ZBP1(S)(ΔZα2)–Flag TRF1(ΔN) versus ZBP1(S)(ΔZα2)–Flag VP16–TRF1(ΔN) (<0.0001). Xp:

ZBP1(S)(WT)–Flag TRF1(ΔN) versus ZBP1(S)(WT)–Flag VP16–TRF1(ΔN) (<0.0001); ZBP1(S)(ΔZα2)–Flag TRF1(ΔN) versus ZBP1(S)(ΔZα2)–Flag VP16–TRF1(ΔN) (<0.0001). Xq: ZBP1(S)(WT)–Flag TRF1(ΔN) versus ZBP1(S)(WT)–Flag VP16–TRF1(ΔN) (0.0023); ZBP1(S)(ΔZα2)–Flag TRF1(ΔN) versus ZBP1(S)(ΔZα2)–Flag VP16–TRF1(ΔN) (0.0009).

Extended Data Figure 10b: CCL5: ZBP1(S)(WT)–Flag TRF1(ΔN) versus ZBP1(S)(WT)–Flag VP16–TRF1(ΔN) (<0.0001); ZBP1(S)(ΔZα2)–Flag TRF1(ΔN) versus ZBP1(S)(ΔZα2)–Flag VP16–TRF1(ΔN) (0.7919). CXCL10–ZBP1(S)(WT)–Flag TRF1(ΔN) versus ZBP1(S)(WT)–Flag VP16–TRF1(ΔN) (<0.0001); ZBP1(S)(ΔZα2)–Flag TRF1(ΔN) versus ZBP1(S)(ΔZα2)–Flag VP16–TRF1(ΔN) (0.9996). IFI44: ZBP1(S)(WT)–Flag TRF1(ΔN) versus ZBP1(S)(WT)–Flag VP16–TRF1(ΔN) (<0.0001); ZBP1(S)(ΔZα2)–Flag TRF1(ΔN) versus ZBP1(S)(ΔZα2)–Flag VP16–TRF1(ΔN) (>0.9999).

Extended Data Figure 10d: CCL5: ZBP1(S)(WT)–Flag TRF1(ΔN) versus ZBP1(S)(WT)–Flag VP16–TRF1(ΔN) (<0.0001); ZBP1(S)(mZα2)–Flag TRF1(ΔN) versus ZBP1(S)(mZα2)–Flag VP16–TRF1(ΔN) (0.8595). CXCL10: ZBP1(S)(WT)–Flag TRF1(ΔN) versus ZBP1(S)(WT)–Flag VP16–TRF1(ΔN) (<0.0001). ZBP1(S)mZα2–Flag TRF1(ΔN) versus ZBP1(S)(mZα2)–Flag VP16–TRF1(ΔN) (>0.9999). IFI44: ZBP1(S)(WT)–Flag TRF1(ΔN) versus ZBP1(S)(WT)–Flag VP16–TRF1(ΔN) (<0.0001). ZBP1(S)(mZα2)–Flag TRF1(ΔN) versus ZBP1(S)(mZα2)–Flag VP16–TRF1(ΔN) (0.9998).

Extended Data Figure 11c: IMR90$^{E6E7}$: TOM20–ZBP1, growing versus crisis (<0.0001); MitoTracker–ZBP1, growing versus crisis (<0.0001); TFAM–ZBP1, growing versus crisis (<0.0001). WI38$^{SV40LT}$: TOM20–ZBP1, growing versus crisis (<0.0001); MitoTracker–ZBP1, growing versus crisis (<0.0001); TFAM–ZBP1, growing versus crisis (<0.0001).

Extended Data Figure 11d: dsDNA intensity: growing versus crisis (<0.0001); dsRNA intensity: growing versus crisis (<0.0001); number of ZBP1 filaments: growing versus crisis (0.0712).

Extended Data Figure 11e: dsRNA intensity: siCtrl versus si*SUV3* (<0.0001); siCtrl versus si*PNPase* (<0.0001); number of ZBP1 filaments: siCtrl versus si*SUV3* (0.6179). siCtrl versus si*PNPase* (0.3240).

Extended Data Figure 12c: CCL5: ZBP1(S)(WT)–Flag TRF1(ΔN): siCtrl versus si*MAVS* (0.6383); siCtrl versus si*MDA5* (0.9499); siCtrl versus si*RIG-I* (>0.9999); siCtrl versus si*MDA5* + *RIG-I* (0.9959). ZBP1(S)(WT)–Flag VP16–TRF1(ΔN): siCtrl versus si*MAVS* (<0.0001), siCtrl versus si*MDA5* (0.2942); siCtrl versus si*RIG-I* (0.9933); siCtrl versus si*MDA5* + *RIG-I* (0.8552). CXCL10: ZBP1(S)(WT)–Flag TRF1(ΔN): siCtrl versus si*MAVS* (0.9997); siCtrl versus si*MDA5* (0.9980); siCtrl versus si*RIG-I* (0.9973); siCtrl versus si*MDA5* + *RIG-I* (>0.9999). ZBP1(S)(WT)–Flag VP16–TRF1(ΔN): siCtrl versus si*MAVS* (<0.0001); siCtrl versus si*MDA5* (0.0645); siCtrl versus si*RIG-I* (0.8284); siCtrl versus si*MDA5* + *RIG-I* (0.1366). IFI44: ZBP1(S)(WT)–Flag TRF1(ΔN): siCtrl versus si*MAVS* (0.9999); siCtrl versus si*MDA5* (>0.9999); siCtrl versus si*RIG-I* (>0.9999); siCtrl versus si*MDA5* + *RIG-I* (>0.9999). ZBP1(S)(WT)–Flag VP16–TRF1(ΔN): siCtrl versus si*MAVS* (<0.0001); siCtrl versus si*MDA5* (<0.0001); siCtrl versus si*RIG-I* (<0.0001); siCtrl versus si*MDA5* + *RIG-I* (<0.0001).

Extended Data Figure 13d: IMR90$^{E6E7}$: vector siCtrl versus si*MAVS* (>0.9999); vector siCtrl versus si*IFNAR2* (>0.9999); Flag–ZBP1(S)–FIS1 siCtrl versus si*MAVS* (<0.0001); Flag–ZBP1(S)–FIS1 siCtrl versus si*IFNAR2* (<0.0001). HMECs: vector siCtrl versus si*MAVS* (>0.9999); vector siCtrl versus si*IFNAR2* (>0.9999); Flag–ZBP1(S)–FIS1 siCtrl versus si*MAVS* (<0.0001); Flag–ZBP1(S)–FIS1 siCtrl versus si*IFNAR2* (<0.0001).

## shRNAs

shRNAs were as follows: Scramble, CCTAAGGTTAAGTCGCCCTCGCTC GAGCGAGGGCGACTTAACCTTAGG, pLKO.1, Addgene, 1864; *TRF2*, ACA GAAGCAGTGGTCGAATC, pLKO.1, Open Biosystems, TRCN0000018358; sh*ZBP1* 1, CCAAGTCCTCTACCGAATGAA, pLKO.1; sh*ZBP1* 2, GCACAATC CAATCAACATGAT, pLKO.1, Sigma-Aldrich, TRCN0000123050.

## siRNAs

siRNAs were as follows: non-targeting pool (Dharmacon, D-001810-10-20): UGGUUUACAUGUCGACUAA, UGGUUUACAUGUUGUGUGA,

UGGUUUACAUGUUUUCUGA, UGGUUUACAUGUUUUCCUA); non-targeting (Dharmacon, D-001810-01-05): UGGUUUACAUGUCGACUAA; *ZBP1* 1 (Dharmacon, J-014650-07-0005): GGACACGGGAACAUCAUUA; *ZBP1* 2 (Dharmacon, J-014650-08-0005): CAAAAGAUGUGAACC GAGA; *ZBP1* 3 (Dharmacon, J-014650-06-0005): GGAUUUCCAUU GCAAACUC; *ZBP1* 4 (Dharmacon, J-014650-05-0005): CAAAGU CAGCCUCAAUUAU; *MB21D1* (Dharmacon, L-015607-02-0005): GAAGAAACAUGGCGGCUAU, AGGAAGCAACUACGACUAA, AGAAC UAGAGUCACCCUAA, CCAAGAAGGCCUGCGCAUU; *TMEM173* (Dharmacon, L-024333-02-0005): UCAUAAACUUUGGAUGCUA, CGAACUCUCUCAAUGGUAU, AGCUGGGACUGCUGUUAAA, GCA GAUGACAGCAGCUUCU; *MAVS* 1 (Dharmacon, J-024237-07-0005): GCAAUGUGGAUGUUGUAGA; *MAVS* 2 (Dharmacon, J-024237-05-0005): AAGUAUAUCUGCCGCAAUU; *MAVS* 3 (Dharmacon, J-024237-06-0005): CAUCCAAAGUGCCUACUAG; *MAVS* 4 (Dharmacon, J-024237-08-0005): CAUCCAAAUUGCCCAUCAA; *PYCARD* (Dharmacon, L-004378-00-0005): GGAAGGUCCUGACGGAUGA, UCACAAACGUU GAGUGGCU, GGCCUGCACUUUAUAGACC, CCACCAACCCAAGCAA GAU; *MYD88* (Dharmacon, L-004769-00-0005): CGACUGAAGUUGU GUGUGU, GCUAGUGAGCUCAUCGAAA, GCAUAUGCCUGAGCGUUUC, GCACCUGUGUCUGGUCUAU; *TLR7* (Dharmacon, L-004714-00-0005): CAACAACCGGCUUGAUUUA, GGAAAUUGCCCUCGUUGUU, GAAUCU AUCACAAGCAUUU, GGAAUUACUCAUAUGCUAA; *TLR3* (Dharmacon, L-007745-00-0005): GAACUAAAGAUCAUCGAUU, CAGCAUCUGUCUU UAAUAA, AGACCAAUCUCUCAAAUUU, UCACGCAAUUGGAAGAUUA; *TLR9* (Dharmacon, L-004066-00-0005): CAGACUGGGUGUACAACGA, GCAAUGCACUGGGCCAUAU, CGGCAACUGUUAUUACAAG, ACA AUAAGCUGGACCUCUA; *TLR8* (Dharmacon, L-4715-00-0005): CAACGGAAAUCCCGGUAUA, CAGAAUAGCAGGCGUAACA, GUG CAGCAAUCGUCGACUA, CUUCCAAACUUAUCGACUA; *DDX58* (Dharmacon, L-012511-00-0005): GCACAGAAGUGUAUAUUGG, CCACAACACUAGUAAACAA, CGGAUUAGCGACAAAUUUA, UCGA UGAGAUUGAGCAAGA; *IFIH1* (Dharmacon, L-013041-00-0005): GAAUAACCCAUCACUAAUA, GCACGAGGAAUAAUCUUUA, UGA CACAAUUCGAAUGAUA, CAAUGAGGCCCUACAAAUU; *TICAM1* (Dharmacon, L-012833-00-0005): GGAGCCACAUGUCAUUUGG, CCAUAGACCACUCAGCUUU, GGACGAACACUCCCAGAUC, CCACUG GCCUCCCUGAUAC; *IFNAR1* (Dharmacon, L-020209-00-0005): GCGAAAGUCUUCUUGAGAU, UGAAACCACUGACUGUAUA, GAAAAU UGGUGUCUAUAGU, GAAGAUAAGGCAAUAGUGA; *IFNAR2* (Dharmacon, L-015411-00-0005): CAGAGGGAAUUUUAAGAA, GAGUAAACCAGAA GAUUUG, CACCAGAGUUUGAGAUUGU, UCACCUAUAUCAUUGACAA; *AIM2* (Dharmacon, L-011951-00-0005): GCACAGUGGUUUCUUUA GAG, UCAGACGAGUUUAAUAUUG, GAAAGUUGAUAAGCAAUAC, GUUCAUAGCACCAUAAAGG; *SUPV3L1* (Dharmacon, L-017841-01-0005): UGGCUAAGCUACCGAUUUA, GUAAGGAUGAUCUACGUAA, CGGUGCAGCUCAUGCGGAU, GGAAAGACUUAUCACGCAA; *PNPT1* (Dharmacon, L-020112-00-0005): GACAGAAGUAGUAUUGUAA, ACAGAAAGAUUAUUGGCUA, GAAUGUAAGUUGUGAGGUA, AAUCA GAGAUACUGGUGUA; *RIPK3* (Dharmacon, L-003534-00-0005): CCA CAGGGUUGGUAUAAUC, AACCAGCACUCUCGUAAUG, GCUACGAU GUGGCGGUCAA, GACCGCUCGUUAACAUAUA; *ATG7* (Dharmacon, L-020112-00-0005): CCAACACACUCGAGUCUUU, GAUCUAAAUCU CAAACUGA, GCCCACAGAUGGAGUAGCA, GCCAGAGGAUUCAACAUGA; TERRA 1 (Dharmacon, CTM-536949): AGGGUUAGGGUUAGGGUUAUU; TERRA 2 (Dharmacon, CTM-536950): GGGUUAGGGUUAGGGUUAGUU; *MLKL* (Dharmacon, L-005326-00-0005): GAGCAACGCAUGCCU GUUU, CAAACUUCCUGGUAACUCA, GAAGGAGCUCUCGCUGUUA, GGAUUUGCAUUGAUGAAAC; *RIPK1* (Dharmacon, L-004445-00-0005): CCACUAGUCUGACGGAUAA, UGAAUGACGUCAACGCAAA, GCACAAAUACGAACUUCAA, GAUGAAAUCCAGUGACUUC); *FADD* (Dharmacon, L-003800-00-0005): CAUUUAACGUCAUAUGUGA, GGAGAAGGCUGGCUCGUCA, UGACAGAGCGUGUGCGGGA, GCAUCUACCUCCGAAGCGU); *CASP8* (Dharmacon, L-003466-00-0005): GGACAAAGUUUACCAAAUG, GCCCAAACUUCACAGCAUU,

GAUAAUCAACGACUAUGAA, GUCAUGCUCUAUCAGAUUU); *NLRP3* (Dharmacon, L-017367-00-0005): GGAUCAAACUACUCUGUGA, UGCAAGAUCUCUCAGCAAA, GAAGUGGGGUUCAGAUAAU, GCAA GACCAAGACGUGUGA); *CASP1* (Dharmacon, L-004401-00-0005): GGAAGACUCAUUGAAACAUA, GAUGGUAGAGCGCAGAUGC, CCGCAA GGUUCGAUUUUCA, GAGUGACUUUGACAAGAUG); *STAT1* (Dharmacon, L-003543-00-0005): GCACGAUGGGCUCAGCUUU, CUACGAACAUGAC CCUAUC, GAACCUGACUUCCAUGCGG, AGAAAGAGCUUGACAGUAA); *STAT2* (Dharmacon, L-012064-00-0005): GGACUGAGUUGCCUG GUUA, GGACUGAGGAUCCAUUAUU, GAGCCCUCCCUGGCAAGUUA, GAUUUGCCCUGUGAUCUGA); *IRF9* (Dharmacon, L-020858-00-0005): GCAGAGACUUGGUCAGGUA, CCACCGAAGUUCCAGGUAA, GCGUGGAGCUCUUCAGAAC, GAAAGUACCAUCAAAGCGA); *ATG5* (Dharmacon, L-004374-00-0005): GGCAUUAUCCAAUUGGUUU, GCAGAACCAUACUAUUUGC, UGACAGAUUUGACCAGUUU, ACAAA GAUGUGCUUCGAGA); *ATG12* (Dharmacon, L-010212-00-0005): GAACACCAAGUUUCACUGU, GCAGUAGAGCGAACACGAA, GGGAA GGACUUACGGAUGU, GGGAUGAACCACAAAGAAA).

## Primers for RT–qPCR

The primers were as follows: 9p-F, GAGATTCTCCCAAGGCAAGG; 9p-R, ACATGAGGAATGTGGGTGTTAT; 13q-F, CTGCCTGCCTTTGGGATAA; 13q-R, AAAACCGTTCTAACTGGTCTCTG; 15q-2-F, CAGCGAGATTCTCC CAAGCTAAG; 15q-2-R, AACCCTAACCACATGAGCAACG; Xq-F, AGCAA GCGGGTCCTGTAGTG; Xq-R, GGTGGAACTTCAGTAATCCGAAA; Xp-F, AAGAACGAAGCTTCCACAGTAT; Xp-R, GGTGGGAGCAGATTAGAGA ATAAA; GAPDH-F, AGCCACATCGCTCAGACAC; GAPDH-R, GCCCAATAC GACCAAATCC; GAPDH RT, GCCCAATACGACCAAATCC; TERRA RT, CCCTAACCCTAACCCTAACCCTAACCCTAA; ZBP1-F, AACATGCAGC TACAATTCCAGA; ZBP1-R, AGTCTCGGTTCACATCTTTTGC; IFNB1-F, ACGCCGCATTGACCATCTAT; IFNB1-R, GTCTCATTCCAGCCAGTGCT; ISG15-F, CGCAGATCACCCAGAAGATCG; ISG15-R, TTCGTCGCATTTGTC CACCA; IFI44-F, AGCCGTCAGGGATGTACTATAAC; IFI44-R, AGGGAAT CATTTGGCTCTGTAGA; CCL5-F, CCAGCAGTCGTCTTTGTCAC; CCL5-R, CTCTGGGTTGGCACACACTT; CXCL10-F, GTGGCATTCAAGGAGTAC CTC; CXCL10-R, TGATGGCCTTCGATTCTGGATT.

## Antibodies

Antibodies used were as follows: TRF2 (Karlseder laboratory), ZBP1 (Novus Biologicals, NBP1-76854 and Cell Signaling Technology, 60968), LC3 (Cell Signaling Technology, 2775 and 3868), STING (Cell Signaling Technology, 13647), CGAS (Cell Signaling Technology, D1D3G), γH2AX (Millipore, 05-636-I), Flag (Sigma-Aldrich, F1804), GAPDH (Abnova, PAB17013), TBK1 (Cell Signaling Technology, 3504S), phosphorylated TBK1 (Cell Signaling Technology, 5483S), STAT1 (Cell Signaling Technology, 9172S), phosphorylated STAT1 (Cell Signaling Technology, 9167L), IRF3 (Cell Signaling Technology, 11904S), phosphorylated IRF3 (Cell Signaling Technology, D4947S), MAVS (Cell Signaling Technology, 24930S), TOMM20 (Abcam, ab56783), TFAM (Abcam, ab119684), dsDNA (Abcam, ab27156), dsRNA (Millipore, MABE1134), phosphorylated ATR (Abcam, ab223258), phosphorylated ATM (Cell Signaling Technology, 5883), the apoptosis antibody sampler kit (Cell Signaling Technology, 9915), the necroptosis antibody sampler kit (Cell Signaling Technology, 98110) and the pyroptosis antibody sampler kit (Cell Signaling Technology, 43811).

## Reporting summary

Further information on research design is available in the Nature Portfolio Reporting Summary linked to this article.

## Data availability

All data are archived at the Salk Institute. RNA-seq data are available at the Gene Expression Omnibus (GEO) repository under accession code GSE218396. A list of enriched gRNAs (log$_2$ fold change > 2) of the

CRISPR–Cas9 screen is available in the Supplementary Information. Raw, uncropped images of western blots, southern blots of terminal restriction fragments and RNA dot-blots are provided in the Supplementary Information. Source data are provided with this paper.

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

**Acknowledgements** We thank N. Arnoult for advice; B. Silva and C. M. Azzalin for discussions; P. Lieberman for reagents; R. Evans for the use of their IncuCyte S3 system; S. P. Moreno for sharing protocols for necroptosis activation; and N. Luhtala for sharing THP-1 cell line for pyroptosis activation. J.N. is supported by the European Molecular Biology Organization (ALTF 213-216) and the Hewitt Foundation and the National Cancer Institute (K99CA252447). A.C., L.M. and N.T. have received support from the Glenn Foundation for Biology of Aging Research. T.T.S. has received support from the European Molecular Biology organization (ALTF 668-2019). S.P. has received support from Swiss National Science Foundation Early Postdoc Mobility Fellowship (P2ZHP3_195173). M.N.S and A.W. are supported by the NCI (P30-014194) and NIA (AG073084, AG064049 and AG068635). J.K. receives support from The Salk Institute Cancer Centre Core Grant (P30CA014195), the National Institutes of Health (RO1CA227934, RO1CA234047, RO1CA228211 and RO1AG077324), the Donald and Darlene Shiley Chair, the Samuel Waxman Cancer Research Foundation, the Highland Street Foundation and the American Heart Association (19PABHI34610000). G.S.S. receives support from the National Institutes of Health (RO1AG077324 and R01AR069876), the American Heart Association-Allen Institute (19PABHI34610000) and the Audrey Geisel Chair in Biomedical Science.

**Author contributions** Experiments were designed by J.N., G.S.S. and J.K. Experiments were performed by J.N. (all experiments except for the ones outlined below), L.G.A. (those shown in Fig. 3a,b and Extended Data Figs. 7c, 8a,b and 10b,d,f), A.C. (those shown in Figs. 1d, 2b and 3c and Extended Data Figs. 4d and 12d–f), T.T.S. (those shown in Figs. 1c and 3f and Extended Data Fig. 3d, and design of sgZBP1 and ZBP1 expression vectors), L.M. (those shown in Extended Data Fig. 7b), S.P. (those shown in Extended Data Figs. 6e, 12b and 13b), S.C.A. (those shown in Extended Data Fig. 3g), and M.N.S. and A.W. (bioinformatics analysis of fRIP–seq and RNA-seq). N.T. provided experimental advice for Extended Data Fig. 11d. V.T. provided advice for Extended Data Fig. 3g. C.H. coordinated all supplies. J.N., G.S.S. and J.K. wrote the manuscript.

**Competing interests** The authors declare no competing interests.

**Additional information**
**Correspondence and requests for materials** should be addressed to Gerald S. Shadel or Jan Karlseder.

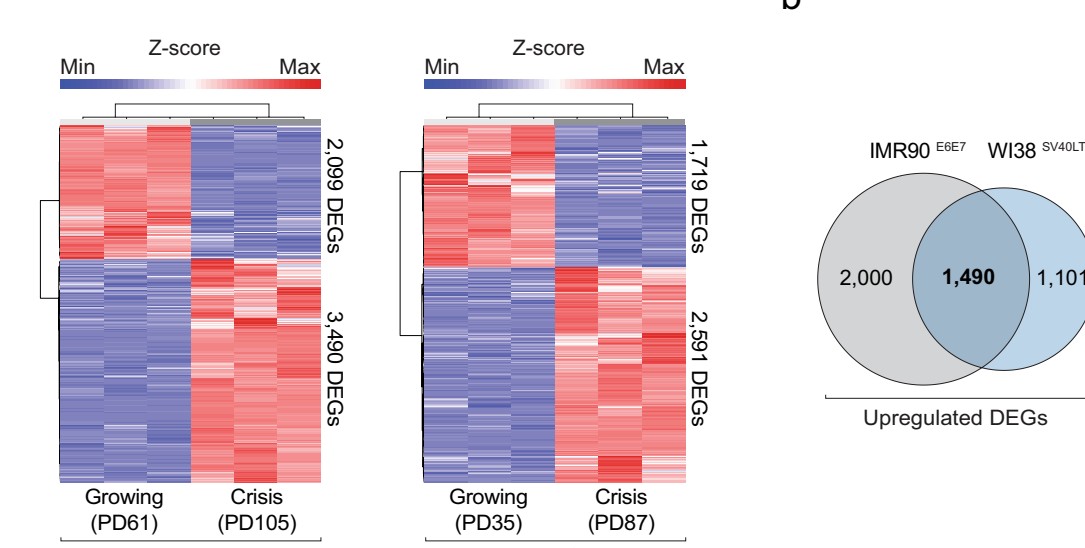

a

IMR90 E6E7

WI38 SV40LT

b

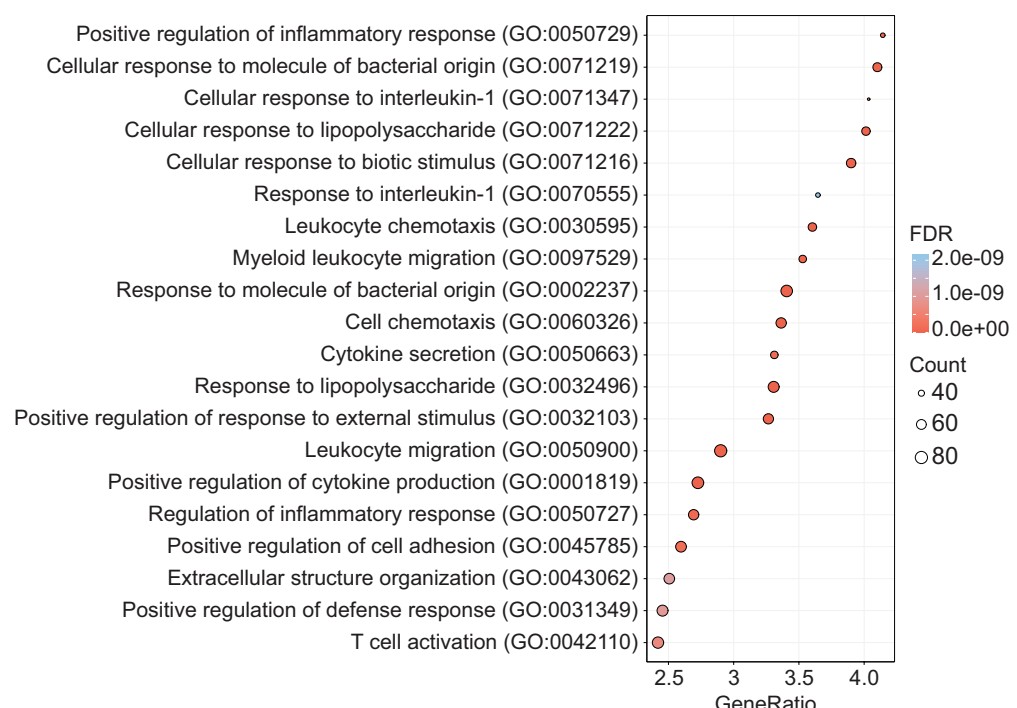

c

GO biological process (top 20 terms; n=1,490 genes)

**Extended Data Fig. 1 | Crisis cells exhibit an inflammatory gene signature.**
**a**, Heat maps representing RNA-sequencing data for DEGs between growing and crisis IMR90[E6E7] and WI38[SV40LT] at the indicated PDs. DEGs with a fold change >1.5 and false-discovery rate <0.05 are shown. One experiment was performed. **b**, Venn diagram of commonly upregulated DEGs (n = 1,490) in crisis of IMR90[E6E7] and WI38[SV40LT] as compared with growing cells. **c**, Graphical representation of the enrichment of gene ontology terms for commonly upregulated DEGs (n = 1,490) in crisis. The top 20 biological processes are shown. Abbreviations: DEGs: differentially expressed genes; GO: gene ontology; PDs: population doublings.

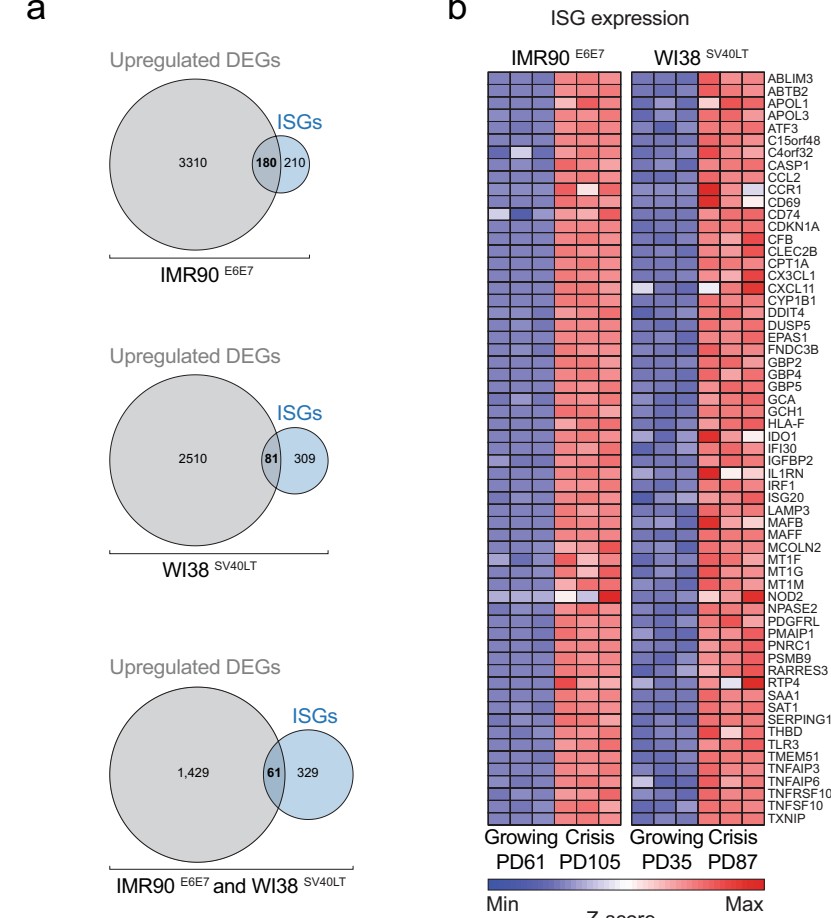

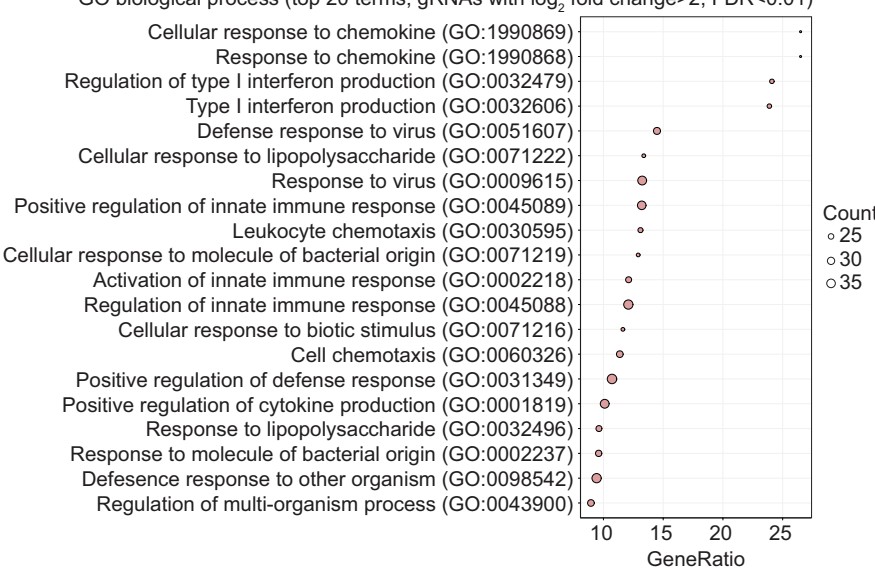

**Extended Data Fig. 2 | ISGs are induced during crisis. a**, Venn diagram of RNA-sequencing data showing significantly upregulated ISGs in crisis of IMR90[E6E7], WI38[SV40LT], or both types of fibroblasts as compared with growing cells (PDs are indicated in Extended Data Fig. 1b). Upregulated ISGs with a fold change > 1.5 and false-discovery rate < 0.05 are shown. The complete list of ISGs includes 390 genes and is based on the RNAseq analysis of 5 cell types transfected with 20 viruses from 9 virus classes[51]. One experiment was performed. **b**, Heat maps representing RNA-sequencing data of upregulated ISGs (n = 61) in crisis of both types of fibroblasts as in **a** (lower panel). **c**, Graphical representation of the enrichment of gene ontology terms for the genome-wide CRISPR/Cas9 knock-out screen. sgRNAs with log$_2$ fold change > 2 are used. The top 20 biological processes are shown. Two independent experiments were performed. Abbreviations: DEGs: differentially expressed genes; sgRNA: single guide RNA; GO: gene ontology; ISGs: interferon stimulated genes, PD: population doubling.

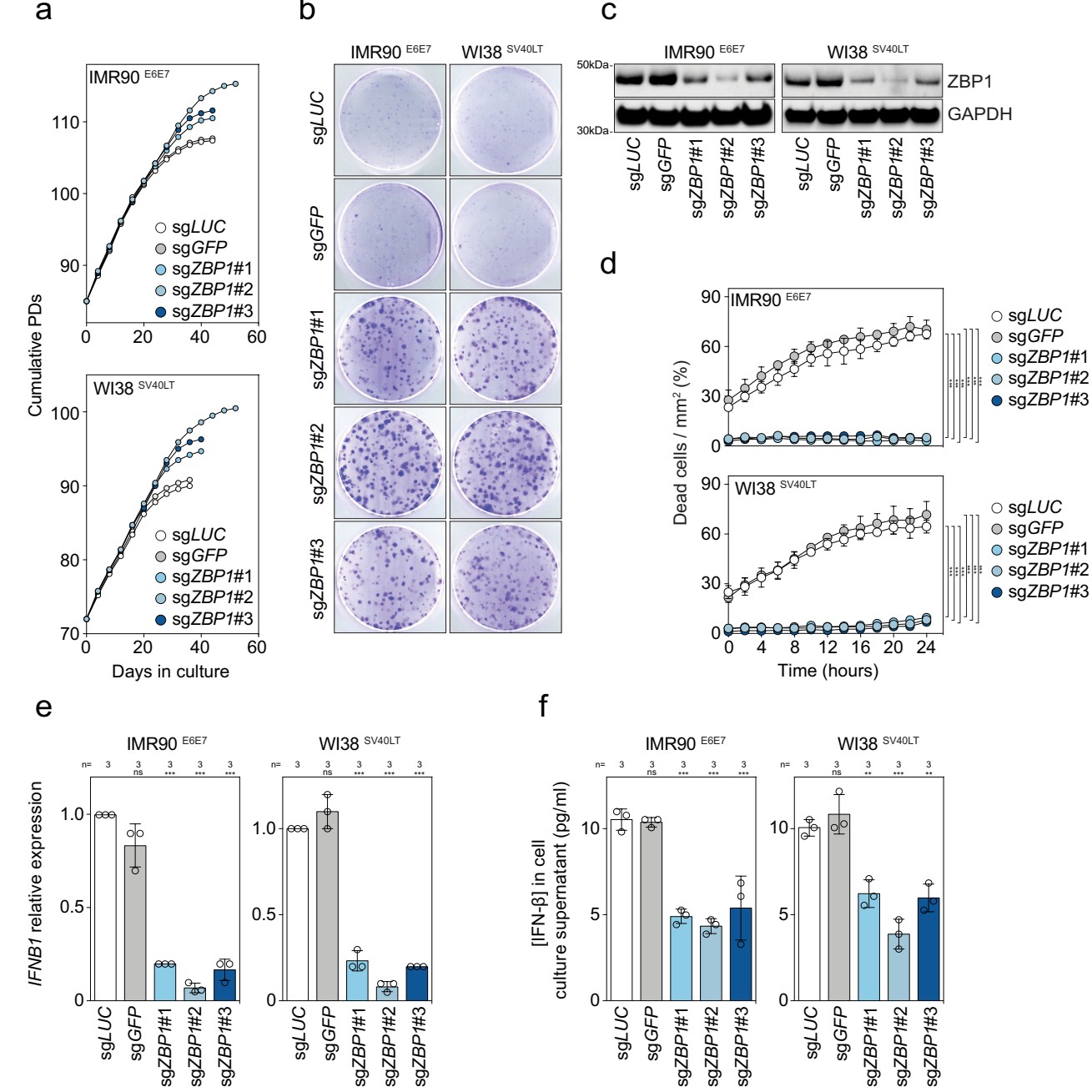

**Extended Data Fig. 3 | Loss of ZBP1 attenuates IFN response and promotes crisis bypass. a**, Line chart showing growth curves analysis of IMR90[E6E7] or WI38[SV40LT] that have bypassed senescence expressing control sgRNAs (sg*LUC*, sg*GFP*) or three individual sgRNAs targeting *ZBP1* (sg*ZBP1*#1, sg*ZBP1*#2, sg*ZBP1*#3). Three independent experiments were performed. **b**, Crystal violet viability assay of IMR90[E6E7] or WI38[SV40LT] as in **a**. Cells were seeded at day 30 and fixed at day 40. One experiment was performed. **c**, Immunoblotting of IMR90[E6E7] or WI38[SV40LT] as in **a**. Protein extracts were collected at day 30 when control cells (sg*LUC*, sg*GFP*) entered crisis and upregulated *ZBP1*. GAPDH loading control. Two independent experiments were performed. **d**, Line chart showing IncuCyte analysis of cell death of IMR90[E6E7] or WI38[SV40LT] as in **a**, measured in real time by Cytotox green incorporation at day 30 of the experiment. Bars represent mean ± s.e.m. from technical replicate. One-way

ANOVA, *** p < 0.001. Two independent experiments were performed. **e**, Scatter plots with bars showing RT-qPCR analysis of *IFNB1* in IMR90[E6E7] or WI38[SV40LT] as in **a**. RNA extracts were collected at day 30 of the experiment. Expression levels were normalized to control cells with sg*LUC*. Bars represent mean ± s.d. of technical replicates. n: number of technical replicates. One-way ANOVA, ns: not significant, *** p < 0.001. Two independent experiments were performed. **f**, Scatter plots with bars showing ELISA assays of IFN-β levels secreted by IMR90[E6E7] or WI38[SV40LT] as in **a**. Culture supernatant were collected at day 30 of the experiment. Bars represent mean ± s.d. of technical replicates. n: number of technical replicates. One-way ANOVA, ns: not significant, *** p < 0.001. One experiment was performed. Abbreviations: GFP: Green fluorescent protein; LUC: luciferase; PDs: population doublings.

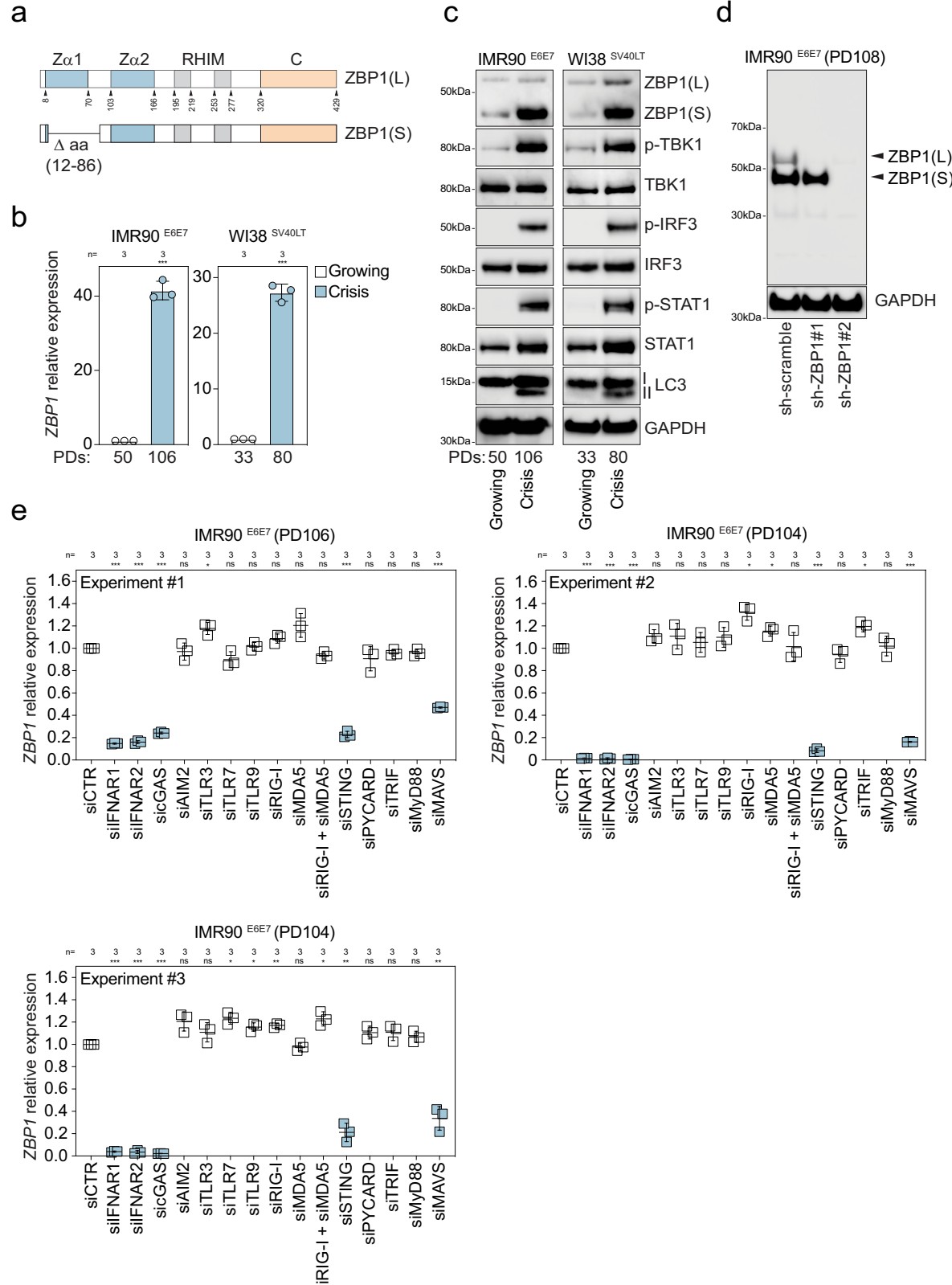

**Extended Data Fig. 4 |** See next page for caption.

**Extended Data Fig. 4 | ZBP1(S) levels increase during crisis dependent on cGAS, STING and MAVS. a**, Graphical representation of human ZBP1 protein showing the positions of Zα1, Zα2 and RHIM domains with reference to the two main ZBP1 isoforms, ZBP1(L) and ZBP1(S). **b**, Scatter plots with bars showing RT-qPCR analysis of ZBP1 in IMR90$^{E6E7}$ and WI38$^{SV40LT}$ at the indicated PDs. Expression levels were normalized to growing cells. Bars represent mean ± s.d. of technical replicates. n: number of technical replicates. Two-tailed Student's *t*-test, *** p < 0.001. Three independent experiments were performed. **c**, Immunoblotting of IMR90$^{E6E7}$ and WI38$^{SV40LT}$ at the indicated PDs. GAPDH loading control. Three independent experiments were performed. **d**, Immunoblotting of crisis (PD108) IMR90$^{E6E7}$ transduced with two individual lentiviral shRNA vectors targeting either ZBP1(L) by binding to a sequence within the Zα1 or a sequence present in both ZBP1 isoforms. Protein extracts were collected at day 6 post-shRNA transfection. GAPDH loading control. Two independent experiments were performed. **e**, Scatter plots showing RT-qPCR analysis of ZBP1 in crisis (PD106) IMR90$^{E6E7}$ transfected with siRNA targeting IFN receptors and main nucleic-acid sensors and adapters. RNA extracts were collected at day 4 post-siRNA transfection. Expression levels were normalized to siCTR cells. Bars represent mean ± s.d. from technical replicates. n: number of technical replicates. One-way ANOVA, ns: not significant, * p < 0.05, ** p < 0.01, *** p < 0.001. Three independent experiments were performed. Abbreviations: ZBP1(L): long isoform; ZBP1(S): short isoform; CTR: control; PDs: population doublings.

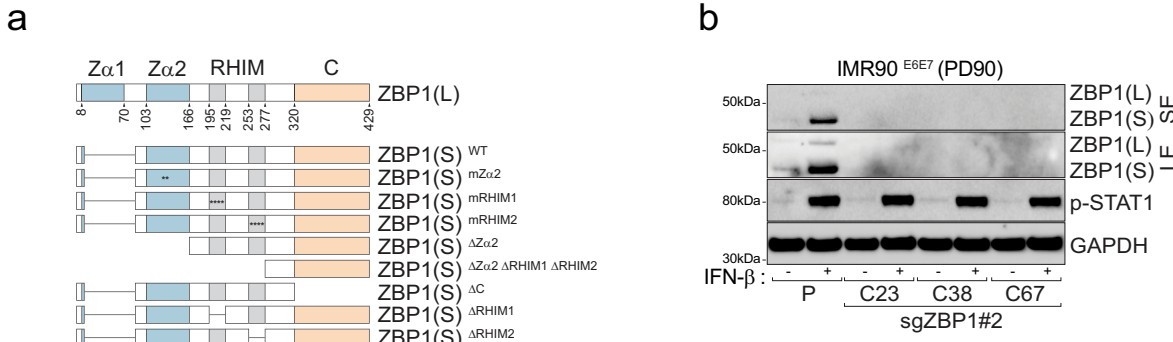

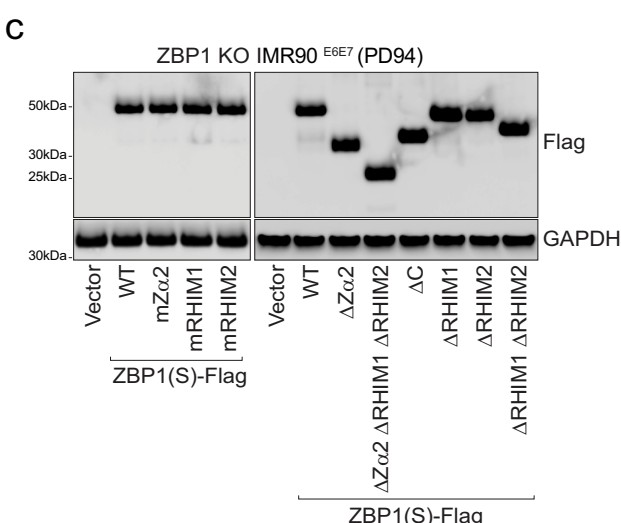

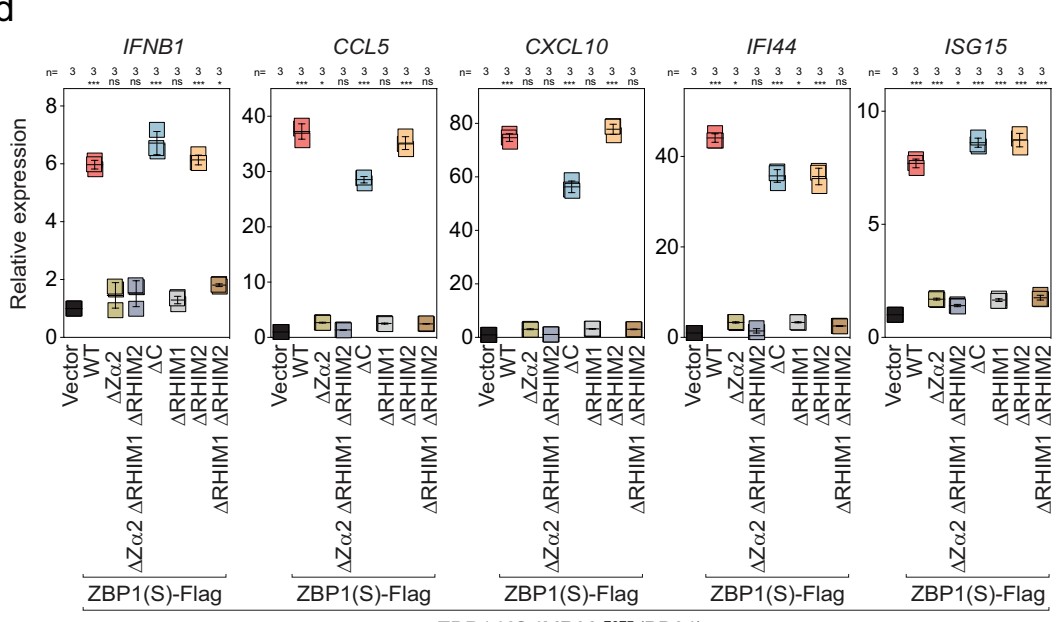

**Extended Data Fig. 5** | See next page for caption.

**Extended Data Fig. 5 | Induction of ISGs requires both Zα2 and RHIM1 domains of ZBP1. a**, Schematic representation of the human ZBP1 mutants. Asterisk represents amino-acid mutation localized in Zα2, RHIM1, or RHIM2. Numbers indicate amino-acid positions. Names of the mutants are listed on the right. **b**, Immunoblotting of parental and ZBP1 CRISPR KO clones of IMR90[E6E7]. Protein extracts were collected 72 h after treatment with 500 U ml[−1] of IFN-β. GAPDH loading control. Two independent experiments were performed. **c**, Immunoblotting of ZBP1 KO MR90[E6E7] (clone 38) reconstituted with either WT or mutant forms of ZBP1(S)-Flag listed in **a**. Protein extracts were collected 14 days after transduction. GAPDH loading control. Two independent experiments were performed. **d**, Scatter plots showing RT-qPCR analysis of ISGs in ZBP1 KO IMR90[E6E7] (clone 38) reconstituted with either WT ZBP1(S)-Flag or mutants ZBP1(S)-Flag lacking Zα2, Zα2-RHIM1-RHIM2, C-terminal region, RHIM1, RHIM2, or RHIM1-RHIM2. Expression levels were normalized to control cells with an empty vector. Bars represent mean ± s.d. of technical replicates n: number of technical replicates. One-way ANOVA, ns: not significant, * p < 0.05, *** p < 0.001. Three independent experiments were performed. Abbreviations: C: clone; P: parental; KO: knock-out; ZBP1(L): long isoform; ZBP1(S): short isoform; SE: short exposure; LE: long exposure; WT: wild-type; PD: population doubling.

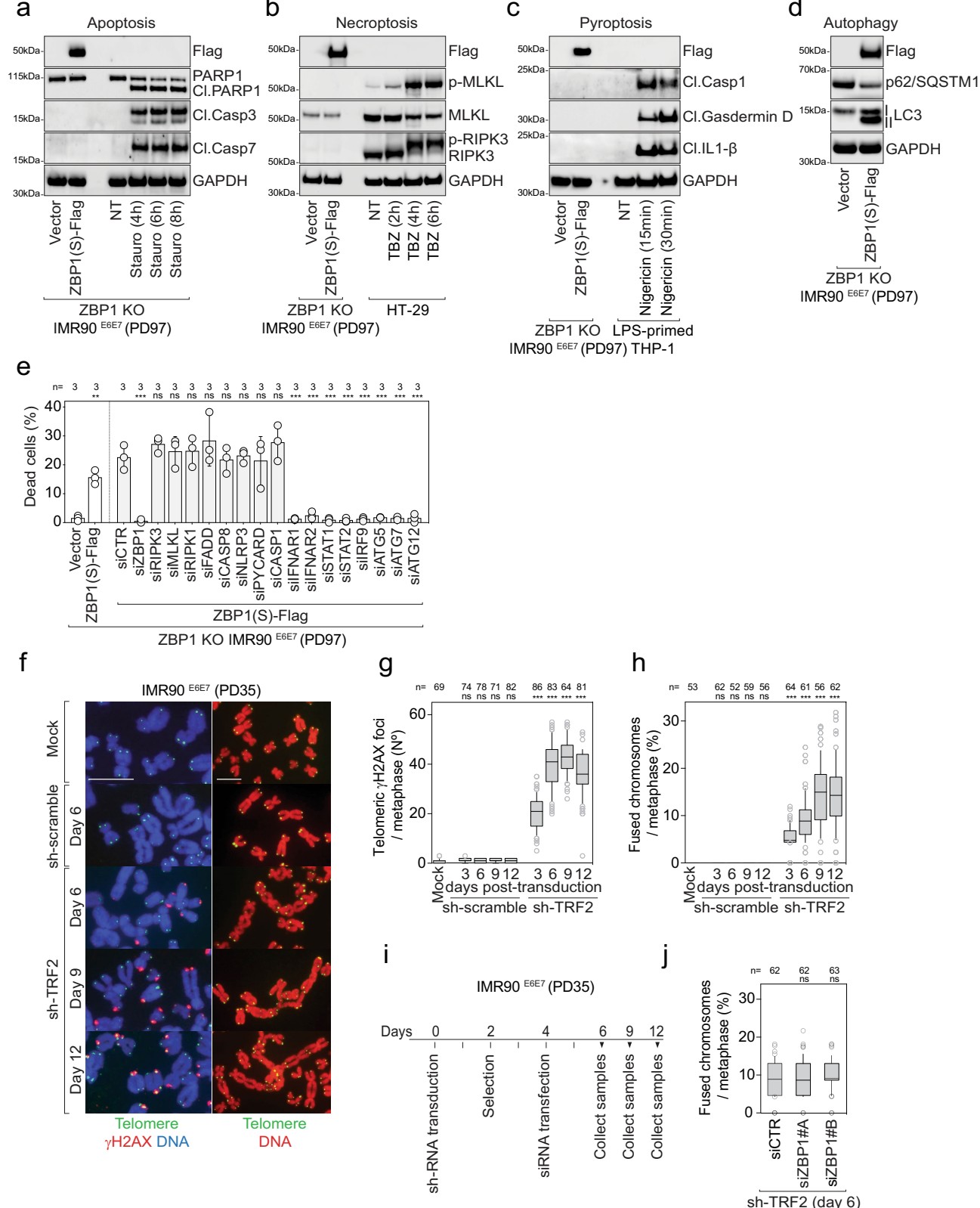

**Extended Data Fig. 6** | See next page for caption.

**Extended Data Fig. 6 | ZBP1(S)-mediated cell death does not involve PANoptosis. a**, Immunoblotting of pre-crisis (PD97) IMR90$^{E6E7}$ expressing empty vector or ZBP1(S)-Flag at day 10 post-transduction. Pre-crisis IMR90$^{E6E7}$ treated with staurosporine (1 μM) serve as positive controls. GAPDH loading control. Two independent experiments were performed. **b**, Immunoblotting of pre-crisis (PD97) IMR90$^{E6E7}$ expressing empty vector or ZBP1(S)-Flag at day 10 post-transduction. HT-29 cells treated with TBZ (TNF (1000 U ml$^{-1}$) + BV6 (1 μM) + z-VAD-FMK (20 μM)) serve as positive controls. GAPDH loading control. Two independent experiments were performed. **c**, Immunoblotting of pre-crisis (PD97) IMR90$^{E6E7}$ expressing empty vector or ZBP1(S)-Flag at day 10 post-transduction. THP-1 cells primed with LPS (1 μg ml$^{-1}$) for 4 h prior to treatment with nigericin (10 μM) for 15 and 30 min serve as positive controls. GAPDH loading control. Two independent experiments were performed. **d**, Immunoblotting of pre-crisis (PD97) IMR90$^{E6E7}$ expressing empty vector or ZBP1(S)-Flag at day 10 post-transduction. GAPDH loading control. Two independent experiments were performed. **e**, Scatter plot with bars showing the percentage of dead pre-crisis (PD97) IMR90$^{E6E7}$ expressing either empty vector or ZBP1(S)-Flag at day 10 post-transduction. ZBP1(S)-Flag expressing cells were transfected with siRNAs at day 8 post-transduction and incubated with Cytotox green for 24 h before imaging. Dead cells are labelled with Cytotox green and nuclei with Hoechst. Bars represent mean ± s.d. from biological replicates. n: number of biological replicates. One-way ANOVA, ns: not significant, p** < 0.01, *** p < 0.001. Three independent experiments were performed. **f**, Left: Representative fluorescence microscopy images of Metaphase-TIF assays stained with DAPI (blue), telomere FISH (green) and γH2AX immunofluorescence (red) performed on growing (PD35) IMR90$^{E6E7}$ expressing non-targeting control shRNA or shRNA against TRF2. Mock represents non-transduced cells. Experiments were performed at the indicated days post-shRNA transduction. Scale bar 10 μm. Three independent experiments were performed. Right: Representative fluorescence microscopy images of cytogenetic preparations stained with DAPI (red) and telomere FISH (green) performed on growing (PD35) IMR90$^{E6E7}$ expressing non-targeting control shRNA or shRNA against TRF2. Mock represents non-transduced cells. Experiments were performed at the indicated days post-shRNA transduction. Scale bar 10 μm. Three independent experiments were performed. **g**, Box and whisker plots showing the number of telomeric γH2AX foci / metaphase (left) and number of fused chromosomes / metaphase (right) at the indicated days post-shRNA transduction. Centre line: median; box limits: 1st and 3rd quartiles; whiskers: 10th and 90th percentiles. n: number of metaphases analysed. One-way ANOVA, ns: not significant, *** p < 0.001. Three independent experiments were performed. **i**, Experimental timeline of Fig. 2a. Growing (PD35) IMR90$^{6E7}$ were first transduced with non-targeting control shRNA or shRNA against TRF2, selected, and transfected with either of two individuals siZBP1 or non-targeting control siRNA at day 4 post-shRNA transduction. Protein extracts and metaphase spreads were prepared at days 6,9, and 12 post-shRNA transduction. **j**, Box and whisker plots showing the number of fused chromosomes / metaphase at day 6 post-shRNA transduction. Centre line: median; box limits: 1st and 3rd quartiles; whiskers: 10th and 90th percentiles. n: number of metaphases analysed. One-way ANOVA, ns: not significant. Three independent experiments were performed. Abbreviations: KO: knock-out; stauro: staurosporine; TBZ: TNF-BV6-z-VAD-FMK; LPS: lipopolysaccharide; Cl.: cleaved; Casp: caspase; NT: non-treated; TIF: telomere-dysfunction-induced foci; FISH: fluorescence in situ hybridization; CTR: control; PDs: population doublings.

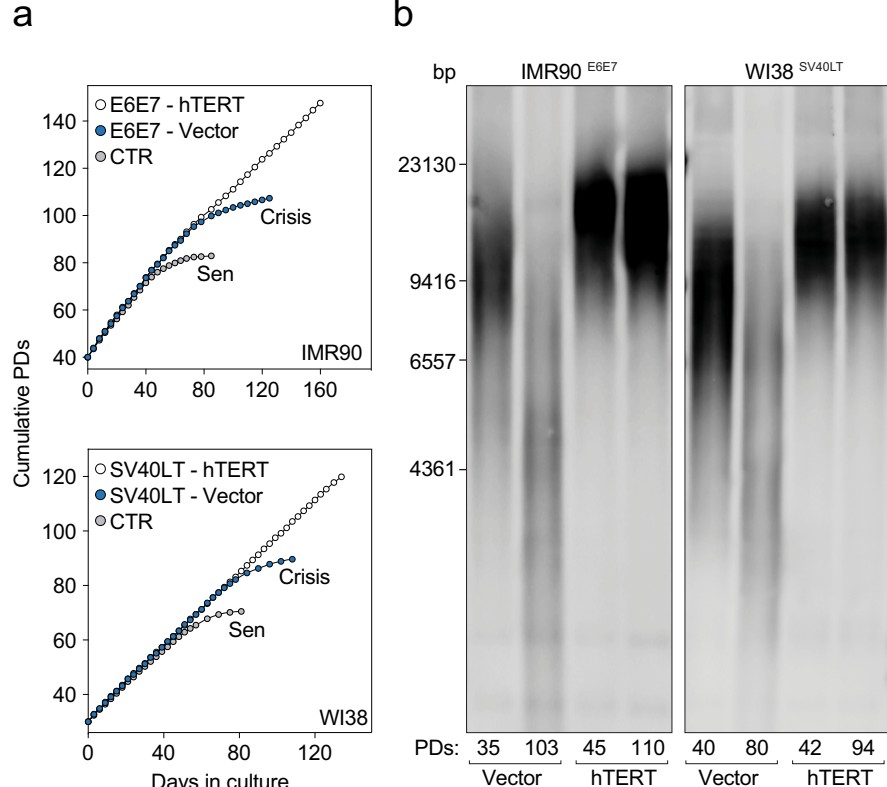

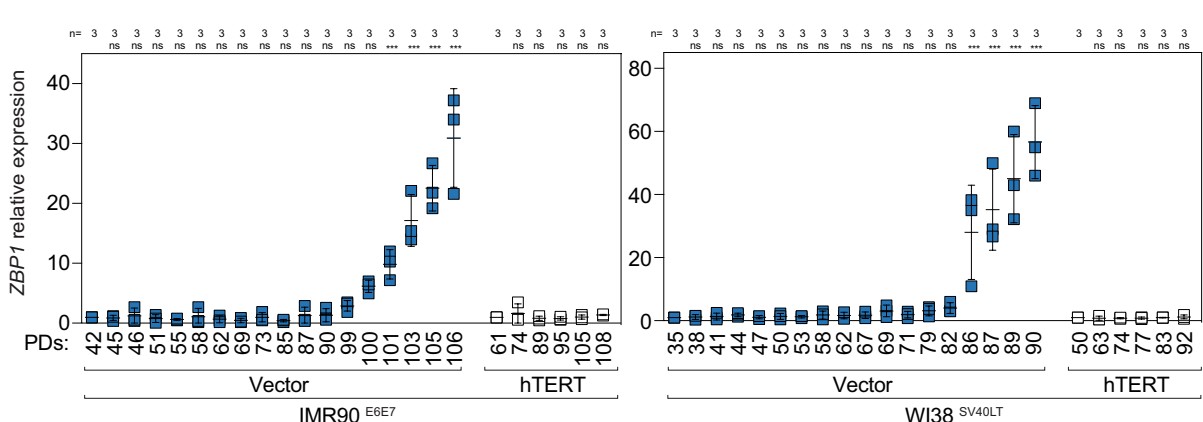

**Extended Data Fig. 7 | Telomere dysfunction is required for ZBP1-dependent IFN signalling. a**, Line chart showing growth curves analysis of IMR90 or WI38 expressing either empty vector (pLXSN3), HPV-E6E7 or SV40-LT in combination with an empty vector (pBABE), or hTERT. Cells with empty vector (pLXSN3) are indicated as CTR. One experiment was performed. **b**, TRF assays performed on IMR90 and WI38 expressing either HPV-E6E7 or SV40-LT in combination with an empty vector (pBABE) or hTERT. Two experiments were performed. DNA extracts were collected from cells at the indicated PDs. **c**, Scatter plots showing RT-qPCR analysis of ZBP1 levels in cells as in **a**. PDs are indicated. Expression levels were normalized to control cells with the earliest PDs. Bars represent mean ± s.d. from biological replicates. n: number of biological replicates. One-way ANOVA, ns: not significant, *** p < 0.001. Three independent experiments were performed. Abbreviations: hTERT: human telomerase reverse transcriptase; CTR: control; PDs: population doublings.

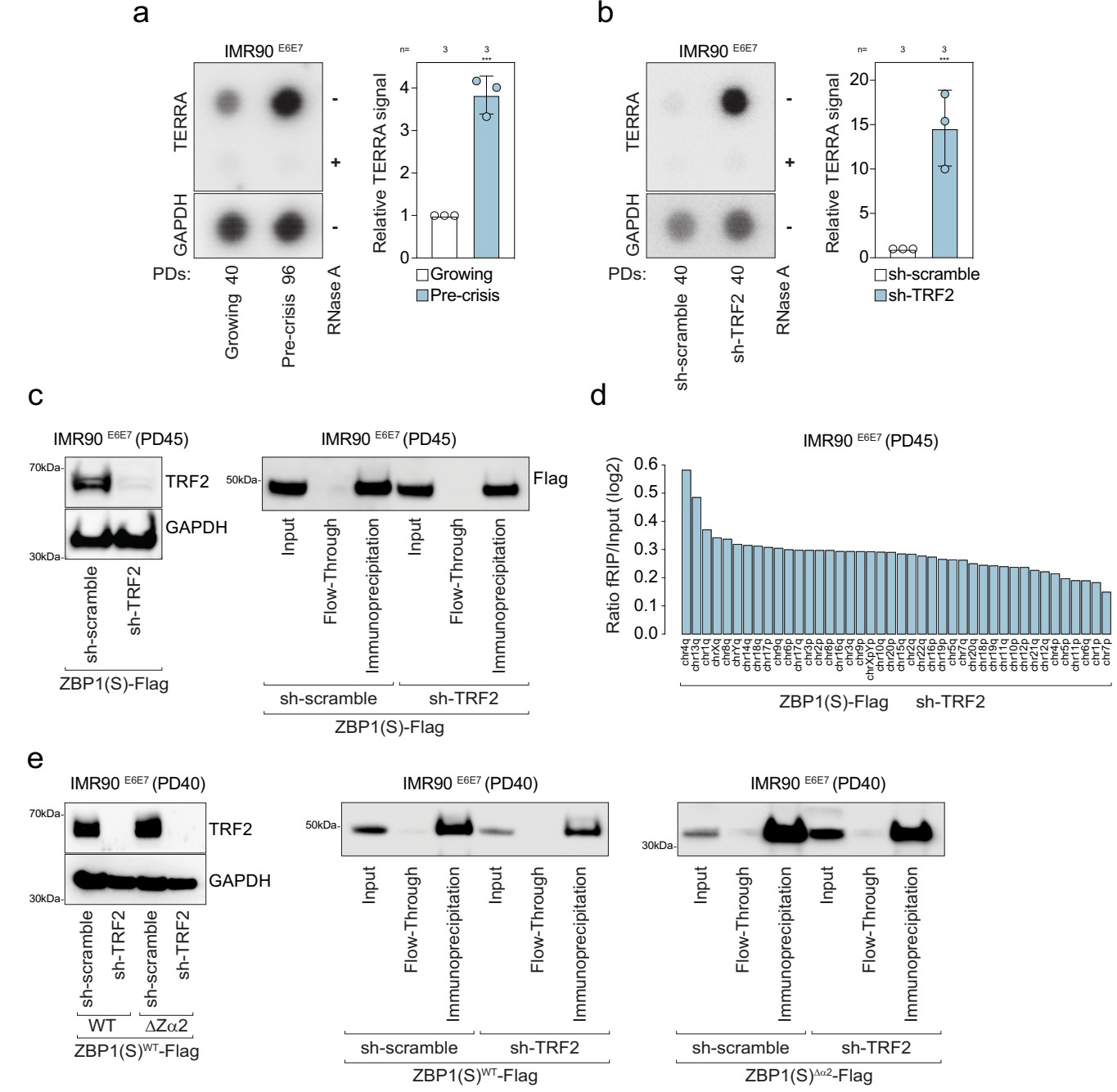

**Extended Data Fig. 8** | See next page for caption.

**Extended Data Fig. 8 | TERRA levels increase upon telomere deprotection.**
**a**, Left: RNA-dot blot performed on RNA isolated from total cell lysates of growing (PD40) and pre-crisis (PD96) IMR90$^{E6E7}$ using $^{32}$P-dCTP-labelled probes targeting either TERRA or GAPDH transcripts. RNase A treatment was used to assess possible DNA contamination. GAPDH loading control. Three independent experiments were performed. Right: Scatter plots with bars showing relative TERRA signal normalized to GAPDH and growing cells. Bars represent mean ± s.d. of biological replicates. n: number of biological replicates. Two-tailed Student's $t$-test, *** p < 0.001. Three independent experiments were performed. **b**, Left: RNA-dot blot performed on RNA isolated from total cell lysates of growing (PD40) IMR90$^{E6E7}$ expressing non-targeting control shRNA or shRNA against TRF2 using $^{32}$P-dCTP-labelled probes targeting either TERRA or GAPDH transcripts. GAPDH loading control. RNA extracts were collected at day 6 post-shRNA transduction. RNase A treatment was used to assess possible DNA contamination. GAPDH loading control. Three independent experiments were performed. Right: Scatter plots with bars showing relative TERRA signals normalized to GAPDH and sh-scramble cells. Bars represent mean ± s.d. of biological replicates. n: number of biological replicates. Two-tailed Student's $t$-test, *** p < 0.001. Three independent experiments were performed. **c**, Left: Immunoblotting of growing (PD45) IMR90$^{E6E7}$ expressing WT ZBP1(S)-Flag and transduced with non-targeting control shRNA or shRNA against TRF2. Total protein extracts were collected at day 12 post-shRNA transduction. GAPDH loading control. Two independent experiments were performed. Right: Immunoblotting growing (PD45) IMR90$^{E6E7}$ as in left. Protein extracts were immunoprecipitated with anti-Flag magnetic beads at day 12 post-shRNA transduction. The Input represents total protein extracts, and the Flow-Through represents unbound fractions. **d**, Column bar graph showing the average ratio fRIP / input (log$_2$) at individual subtelomeres. fRIP and input samples were normalized into the same sequencing depth. Bars represent mean. Two independent experiments were performed. **e**, Left: Immunoblotting of growing (PD40) IMR90$^{E6E7}$ expressing either WT ZBP1(S)-Flag or mutant ZBP1(S)-Flag lacking Zα2 transduced with non-targeting control shRNA or shRNA against TRF2. Total protein extracts were collected at day 12 post-shRNA transduction. GAPDH loading control. Three independent experiments were performed. Right: Immunoblotting of growing (PD40) IMR90$^{E6E7}$ as in left. Protein extracts were immunoprecipitated with anti-Flag magnetic beads at day 12 post-shRNA transduction. The Input represents total protein extracts, and the Flow-Through represents unbound fractions. Three independent experiments were performed. Abbreviations: ZBP1(S): short isoform; Chr: chromosome; WT: wild-type; PDs: population doublings.

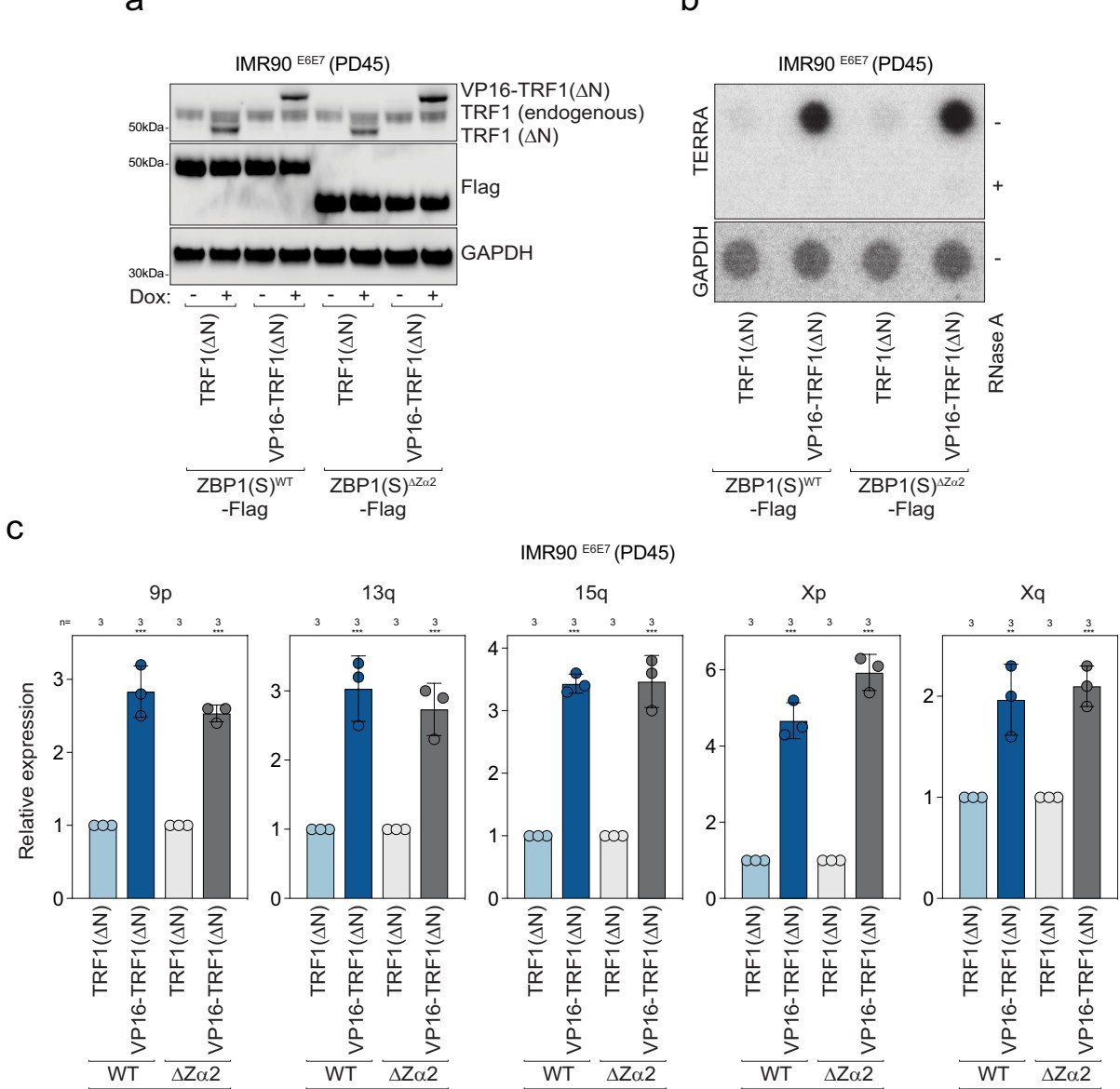

**Extended Data Fig. 9 | Sensing of TERRA by ZBP1 triggers an IFN response.**
**a**, Immunoblotting of growing (PD45) IMR90[E6E7] expressing either WT
ZBP1(S)-Flag or mutant ZBP1(S)-Flag lacking Zα2 with either TRF1(ΔN) or
VP16-TRF1(ΔN). Protein extracts were collected at day 4 post-doxycycline
treatment (1 μg/ml). GAPDH loading control. Two independent experiments
were performed. **b**, RNA-dot blot performed on RNA isolated from cells as in
**a** using [32]P-dCTP-labelled probes targeting either TERRA or GAPDH transcripts.
RNA extracts were collected at day 4 post-doxycycline treatment (1 μg ml⁻¹).
GAPDH loading control. RNase A treatment was used to assess possible DNA

contamination. Two independent experiments were performed. **c**, Scatter
plot with bars showing RT-qPCR analysis with 5 pairs of subtelomere-specific
primers designed to measure TERRA transcribed from individual chromosome
ends. Expression levels were normalized to control cells expressing TRF1(ΔN).
RNA extracts were collected from cells as in **a**. Bars represent mean ± s.d.
from technical replicates. n: number of technical replicates. One-way ANOVA,
** $p < 0.01$, *** $p < 0.001$. Two independent experiment were performed.
Abbreviations: ZBP1(S): short isoform; Dox: doxycycline; WT: wild-type; fRIP:
formaldehyde RNA immunoprecipitation; PDs: population doublings.

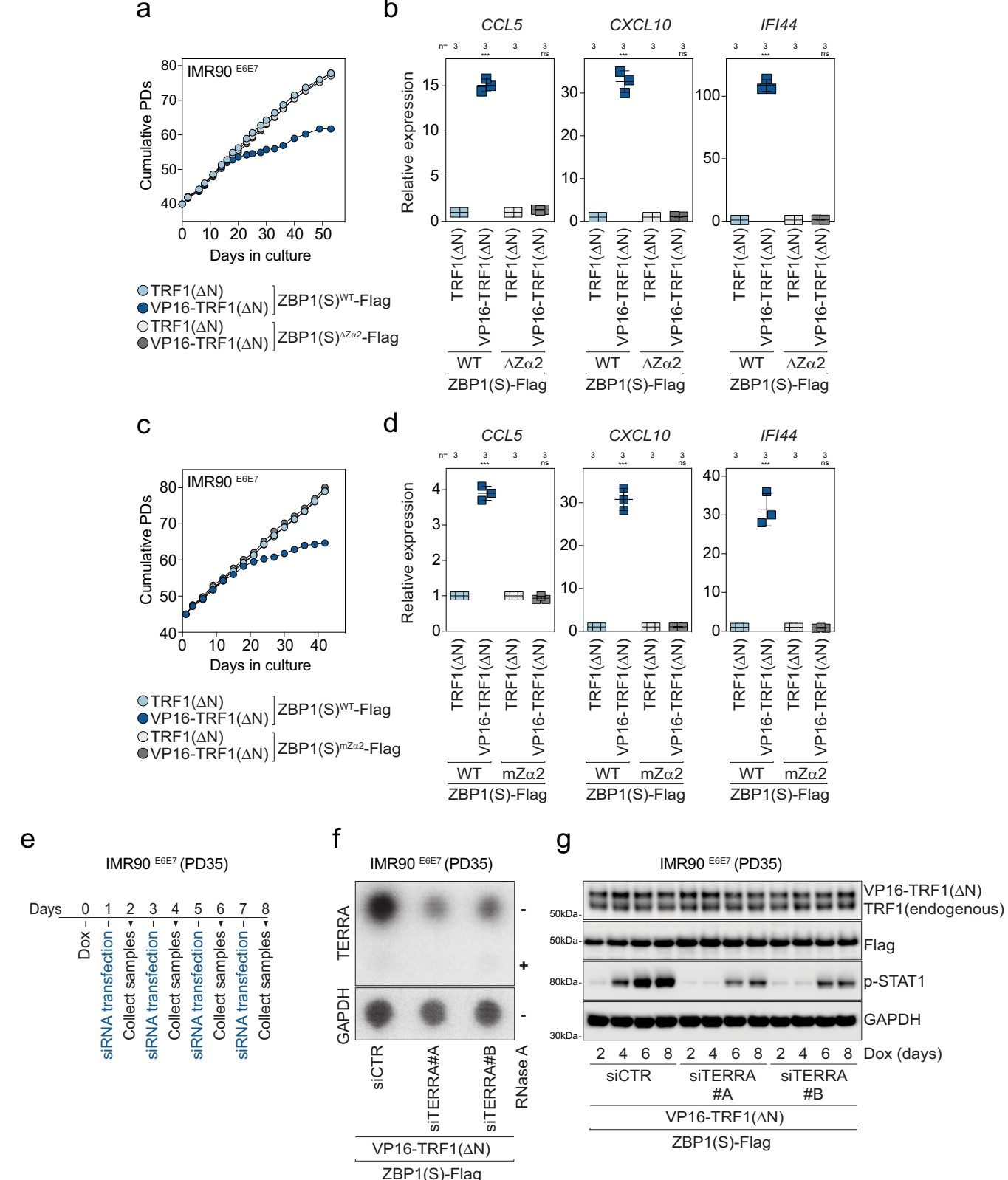

**Extended Data Fig. 10** | See next page for caption.

**Extended Data Fig. 10 | Sensing of TERRA by ZBP1 triggers an IFN response.**
**a**, Line chart showing growth curve analysis of IMR90[E6E7] expressing either
WT ZBP1(S)-Flag or ZBP1(S)-Flag lacking Zα2 with either TRF1(ΔN) or VP16-
TRF1(ΔN). Fresh media with doxycycline (1 µg ml⁻¹) was added every 2 days. Two
independent experiments were performed. **b**, Scatter plots showing RT-qPCR
analysis of ISG levels in cells as in **a**. Expression levels were normalized to
control cells expressing either WT ZBP1(S)-Flag or ZBP1(S)-Flag lacking Zα2
with TRF1(ΔN). RNA extracts were collected at day 20 of the experiment. Bars
represent mean ± s.d. from technical replicates. n: number of technical
replicates. One-way ANOVA, ns: not significant, *** p < 0.001. Two independent
experiments were performed. **c**, Line chart showing growth curve analysis of
IMR90[E6E7] expressing either WT ZBP1(S)-Flag or ZBP1(S)-Flag containing point
mutations in Zα2, with either TRF1(ΔN) or VP16-TRF1(ΔN). Fresh media
including 1 µg ml⁻¹ of doxycycline was added every 2 days. Two independent
experiments were performed. **d**, Scatter plots showing RT-qPCR analysis of ISG
levels in cells as in **c**. Expression levels were normalized to control cells
expressing either WT ZBP1(S)-Flag or ZBP1(S)-Flag containing point mutations
in Zα2 with TRF1(ΔN). RNA extracts were collected at day 30 of the experiment.
Bars represent mean ± s.d. from technical replicates. n: number of technical
replicates. One-way ANOVA, ns: not significant, *** p < 0.001. Two independent
experiments were performed. **e**, Experimental timeline. Growing (PD35)
IMR90[E6E7] expressing WT ZBP1(S)-Flag were treated with doxycycline (1 µg ml⁻¹)
prior to four sequential siRNA transfections. Two individual siRNAs targeting
TERRA were used. Samples were collected at days 2,4,6, and 8 post-
doxycycline. **f**, RNA-dot blot performed on RNA isolated from total cell lysates
at day 8 post-doxycycline as shown in the timeline of the experiment, using
³²P-dCTP-labelled probes targeting either TERRA or GAPDH transcripts. RNase
A treatment was used to assess possible DNA contamination. GAPDH loading
control. Three independent experiments were performed. **g**, Western blotting
performed on protein extracts collected at days 2,4,6, and 8 post-doxycycline
treatment as shown in **e**. GAPDH loading control. Two independent experiments
were performed. Abbreviations: ZBP1(S): short isoform; WT: wild-type; CTR:
control; PDs: population doublings.

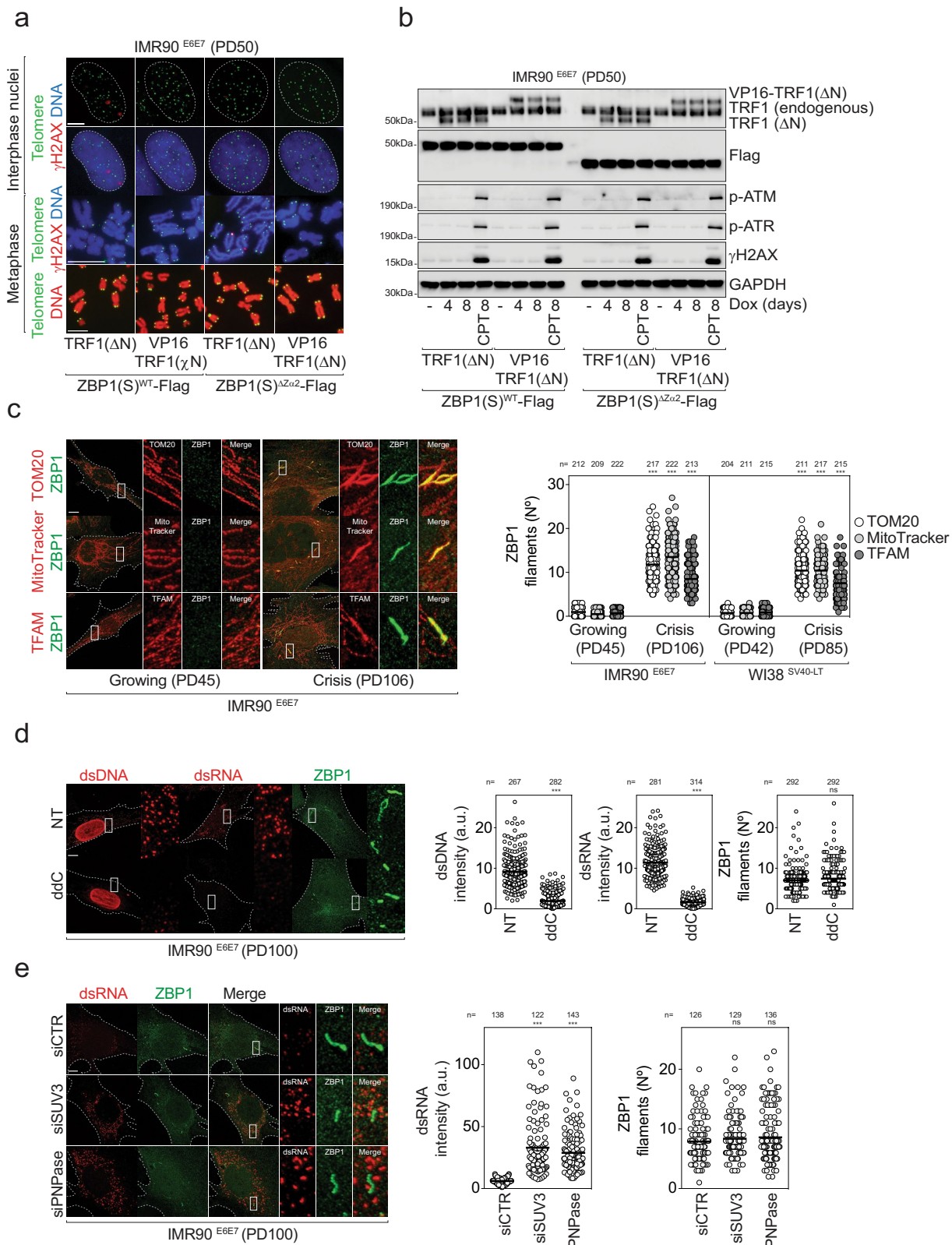

**Extended Data Fig. 11** | See next page for caption.

**Extended Data Fig. 11 | Assembly of ZBP1 filaments at mitochondria does not require sensing of mtDNA or mtRNA. a**, Upper panel: Representative confocal images of growing (PD50) IMR90^E6E7 expressing either WT ZBP1(S)-Flag or ZBP1(S)-Flag lacking Zα2 with either TRF1(ΔN) or VP16-TRF1(ΔN). Interphase nuclei were stained with DAPI (blue), telomere FISH (green), and γH2AX immunofluorescence (red). Scale bar 5 μm. Middle panel: Representative fluorescence microscopy images of Metaphase-TIF assays stained with DAPI (blue), telomere FISH (green) and γH2AX immunofluorescence (red). Scale bar 10 μm. Lower panel: Representative fluorescence microscopy images of cytogenetic preparations stained with DAPI (red) and telomere FISH (green). Scale bar 10 μm. Three independent experiments were performed at day 8 post-doxycycline treatment (1 μg ml⁻¹). **b**, Immunoblotting of cells as in **a** either untreated or at days 4 and 8 post-doxycycline treatment (1 μg ml⁻¹). Cells treated with 1 μM camptothecin (CPT) for 6 h serve as a positive control. GAPDH loading control. Two independent experiments were performed. **c**, Left: Representative confocal microscopy images of growing and crisis IMR90^E6E7 co-immunostained for endogenous ZBP1 and mitochondrial markers TOM20, TFAM, and MitoTracker at the indicated PDs. Scale bar 10 μm. Right: Scatter plots showing the number of ZBP1 filaments colocalizing with either TOM20, TFAM, or MitoTracker in growing and crisis IMR90^E6E7 and WI38^SV40LT at the indicated PDs. Centre line: mean. n: number of cells analysed. One-way ANOVA, *** p < 0.001. Three independent experiments were performed. **d**, Left: Representative confocal microscopy images of crisis (PD100) IMR90^E6E7 immunostained with antibodies against endogenous ZBP1, dsDNA, and dsRNA. Cells were incubated with 5 μM of 2',3' dideoxycytidine (ddC) for 14 days. Non-treated cells serve as controls. Scale bar 10 μm. Three independent experiments were performed. Right: Scatter plots showing the signal intensity of dsDNA (left), the signal intensity of dsRNA (middle), and the number of ZBP1 filaments (right). Centre line: mean. n: number of cells analysed. Three independent experiments were performed. Two-tailed Student's t-test, ns: not significant, *** p < 0.001. Three independent experiments were performed. **e**, Left: Representative confocal microscopy images of growing (PD100) IMR90^E6E7 transfected with either non-targeting control siRNA (siCTR), siSUV3, or PNPase, and co-immunostained for endogenous ZBP1 and double-stranded RNA (dsRNA). Experiment was performed at day 7 post-siRNA transfection. Scale bar 10 μm. Three independent experiments were performed. Right: Scatter plots showing the signal intensity of dsRNA (left) and the number of ZBP1 filaments (right). Centre line: mean. n: number of cells analysed. Two-tailed Student's t-test, ns: not significant, *** p < 0.001. Three independent experiments were performed. Abbreviations: ddC: dideoxycytidine; dsDNA: double-stranded DNA; dsRNA: double-stranded RNA; Mito: mitochondria; WT: wild-type; CTR: control; TIF: telomere-dysfunction-induced foci; FISH: fluorescence in situ hybridization; CPT: camptothecin; WT: wild-type; PDs: population doublings.

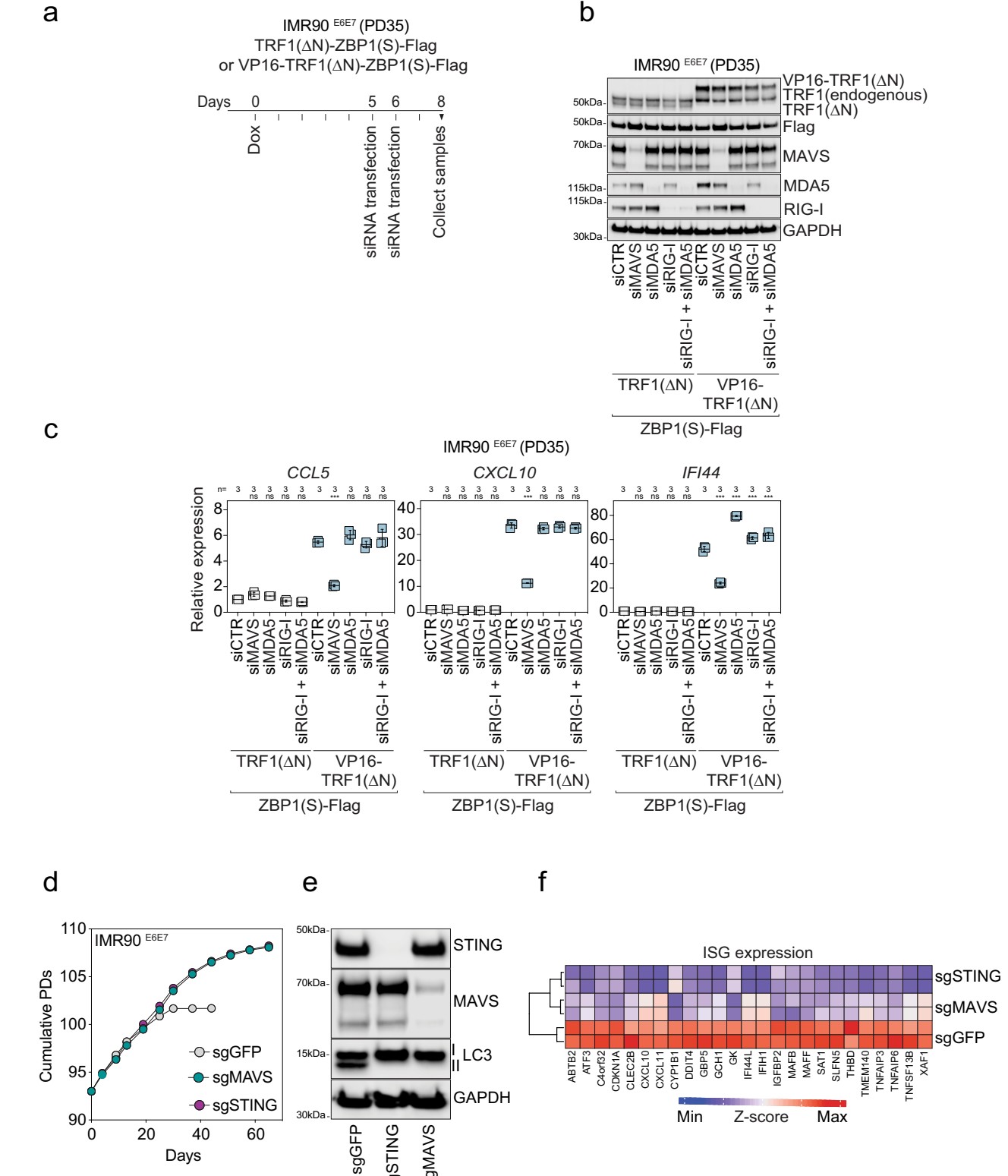

**Extended Data Fig. 12** | See next page for caption.

**Extended Data Fig. 12 | TERRA triggers a MAVS-dependent IFN response. a**, Experimental timeline. Growing (PD35) IMR90$^{E6E7}$ expressing WT ZBP1(S)-Flag with either TRF1(ΔN) or VP16-TRF1(ΔN) were treated with doxycycline (1 µg ml$^{-1}$) prior to two sequential siRNA transfections. Samples were collected at day 8 post-doxycycline. **b**, Immunoblotting of cells as in **a** at day 8 post-doxycycline. GAPDH loading control. Three independent experiments were performed. **c**, Scatter plots showing RT-qPCR analysis of ISGs in cells as in **b**. Expression levels were normalized to control cells TRF1(ΔN) transfected with non-targeting control (siCTR) siRNA. Bars represent mean ± s.d. from technical replicates. n: number of technical replicates. One-way ANOVA, ns: not significant, *** p < 0.001. Three independent experiments were performed. **d**, Line chart showing growth curves analysis of IMR90$^{E6E7}$ that have bypassed senescence expressing control sgRNAs (sg*GFP*) or sgRNAs targeting *MAVS or STING*. Two independent experiments were performed. **e**, Immunoblotting of IMR90$^{E6E7}$ as in **a**. Protein extracts were collected at day 30 when control cells entered crisis. GAPDH loading control. Two independent experiments were performed. **f**, Heatmap representing RNA-sequencing data for ISGs in IMR90$^{E6E7}$ as in **a**. RNA extracts were collected at day 30 when control cells entered crisis. ISGs that are significantly downregulated (sgMAVS versus sgGFP) with a fold change > 1.5 and false-discovery rate < 0.05 are shown. The complete list of ISGs includes 390 genes and is based on the RNAseq analysis of 5 cell types transfected with 20 viruses from 9 virus classes[51]. One experiment was performed. Abbreviations: ZBP1(S): short isoform; sgRNA: single guide RNA; CTR: control; WT: wild-type; PDs: population doublings.

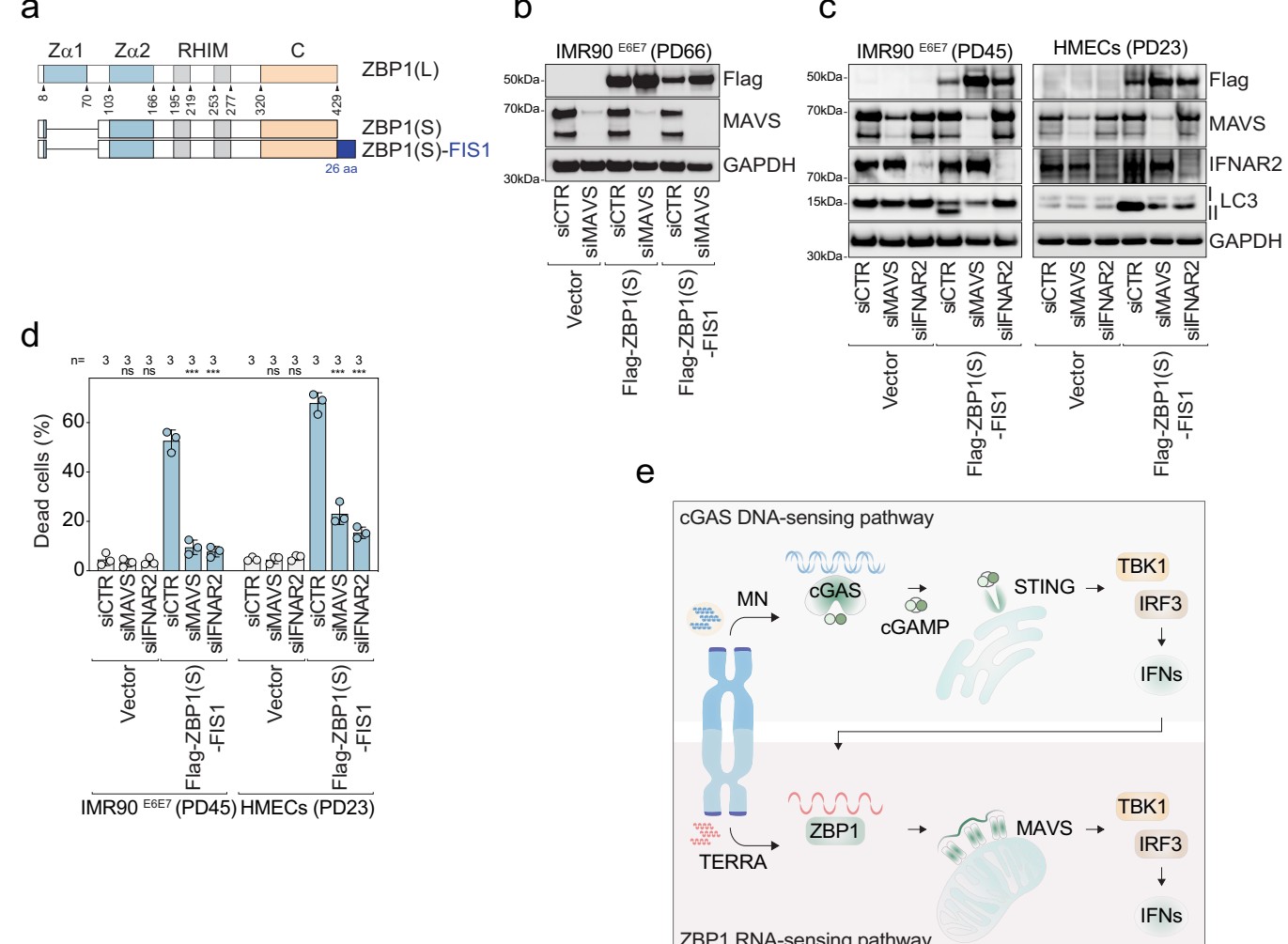

**Extended Data Fig. 13 | ZBP1-MAVS axis promotes IFN-driven cell death. a**, Graphical representation of ZBP1(S) harbouring the mitochondrial targeting sequence of FIS1 with reference to the two main ZBP1 isoforms, ZBP1(L) and ZBP1(S). **b**, Immunoblotting of growing (PD66) IMR90$^{E6E7}$ expressing either empty vector, Flag-ZBP1(S), or Flag-ZBP1(S)-FIS1. Cells were transfected with either siMAVS or non-targeting control siRNA at day 10 post-transduction and protein extracts were collected at day 14. GAPDH loading control. Two independent experiments were performed. **c**, Immunoblotting of growing (PD45) IMR90$^{E6E7}$ (left) and post-senescence (PD23) HMECs expressing either empty vector or Flag-ZBP1(S)-FIS1. Cells were transfected with either siMAVS, siIFNAR2, or non-targeting control siRNA at day 10 post-transduction and protein extracts were collected at day 14. GAPDH loading control. Two independent experiments were performed. **d**, Scatter plot with bars showing the percentage of dead cells as in **c** after 24 h of incubation with Cytotox. Cells were transfected with siRNAs at day 8 post-transduction and incubated with

Cytotox for 24 h before imaging. Dead cells are labelled with Cytotox red and nuclei with Hoechst. Bars represent mean ± s.d. from biological replicates. One-way ANOVA, ns: not significant, *** p < 0.001. Three independent experiments were performed. **e**, Crisis-associated cell death requires the simultaneous activation of two distinct, but functionally interconnected innate immune sensing pathways. cGAS recognizes nuclear DNA species released to the cytosol as byproducts breakage-fusion-bridge (BFB) cycles and primes the expression of several ISGs. ZBP1 becomes induced and further enhances the innate immune response by detecting TERRA molecules stemming from deprotected telomeres. This triggers ZBP1 self-oligomerizing into filaments at mitochondria capable of activating MAVS. Activation of DNA- and RNA-sensing pathways results in a chronic production of type I IFNs and other inflammatory cytokines responsible for the elimination of aberrant cells with unstable telomeres. Abbreviations: ZBP1(S): short isoform; CTR: control; WT: wild-type; MN: micronuclei; IFNs: interferons; PDs: population doublings.

# Reporting Summary

## Statistics

For all statistical analyses, confirm that the following items are present in the figure legend, table legend, main text, or Methods section.

| n/a | Confirmed | |
|---|---|---|
| ☐ | ☒ | The exact sample size (*n*) for each experimental group/condition, given as a discrete number and unit of measurement |
| ☒ | ☐ | A statement on whether measurements were taken from distinct samples or whether the same sample was measured repeatedly |
| ☐ | ☒ | The statistical test(s) used AND whether they are one- or two-sided *Only common tests should be described solely by name; describe more complex techniques in the Methods section.* |
| ☒ | ☐ | A description of all covariates tested |
| ☒ | ☐ | A description of any assumptions or corrections, such as tests of normality and adjustment for multiple comparisons |
| ☐ | ☒ | A full description of the statistical parameters including central tendency (e.g. means) or other basic estimates (e.g. regression coefficient) AND variation (e.g. standard deviation) or associated estimates of uncertainty (e.g. confidence intervals) |
| ☒ | ☐ | For null hypothesis testing, the test statistic (e.g. *F*, *t*, *r*) with confidence intervals, effect sizes, degrees of freedom and *P* value noted *Give P values as exact values whenever suitable.* |
| ☒ | ☐ | For Bayesian analysis, information on the choice of priors and Markov chain Monte Carlo settings |
| ☒ | ☐ | For hierarchical and complex designs, identification of the appropriate level for tests and full reporting of outcomes |
| ☐ | ☒ | Estimates of effect sizes (e.g. Cohen's *d*, Pearson's *r*), indicating how they were calculated |

*Our web collection on statistics for biologists contains articles on many of the points above.*

## Software and code

Policy information about availability of computer code

| Data collection | n/a |
|---|---|
| Data analysis | n/a |

For manuscripts utilizing custom algorithms or software that are central to the research but not yet described in published literature, software must be made available to editors and reviewers. We strongly encourage code deposition in a community repository (e.g. GitHub). See the Nature Portfolio guidelines for submitting code & software for further information.

## Data

Policy information about availability of data

All manuscripts must include a data availability statement. This statement should provide the following information, where applicable:
- Accession codes, unique identifiers, or web links for publicly available datasets
- A description of any restrictions on data availability
- For clinical datasets or third party data, please ensure that the statement adheres to our policy

RNA-sequencing data are available through the Gene Expression Omnibus (GEO) repository under accession codes xxx and xxx.
Raw, uncropped images of western blots, southern blots of terminal restriction fragments, and RNA dot-blots are provided in Supplementary Figures.
Source data for Figures 1-4 and Extended Data Figures 1-12 are provided.

## Human research participants

Policy information about studies involving human research participants and Sex and Gender in Research.

| | |
|---|---|
| Reporting on sex and gender | n/a |
| Population characteristics | n/a |
| Recruitment | n/a |
| Ethics oversight | n/a |

Note that full information on the approval of the study protocol must also be provided in the manuscript.

# Field-specific reporting

Please select the one below that is the best fit for your research. If you are not sure, read the appropriate sections before making your selection.

☒ Life sciences ☐ Behavioural & social sciences ☐ Ecological, evolutionary & environmental sciences

For a reference copy of the document with all sections, see nature.com/documents/nr-reporting-summary-flat.pdf

# Life sciences study design

All studies must disclose on these points even when the disclosure is negative.

| | |
|---|---|
| Sample size | Sample size was determined by significance. |
| Data exclusions | No data were excluded. |
| Replication | Data were reproduced and number of repeats are indicated. Repetitions confirmed the initial results. |
| Randomization | Randomization was not possible, since the experiments were completed by the same individuals. |
| Blinding | Blinding was not possible, since the experiments were completed by the same individuals. |

# Reporting for specific materials, systems and methods

We require information from authors about some types of materials, experimental systems and methods used in many studies. Here, indicate whether each material, system or method listed is relevant to your study. If you are not sure if a list item applies to your research, read the appropriate section before selecting a response.

## Materials & experimental systems

| n/a | Involved in the study |
|---|---|
| ☐ | ☒ Antibodies |
| ☐ | ☒ Eukaryotic cell lines |
| ☒ | ☐ Palaeontology and archaeology |
| ☒ | ☐ Animals and other organisms |
| ☒ | ☐ Clinical data |
| ☒ | ☐ Dual use research of concern |

## Methods

| n/a | Involved in the study |
|---|---|
| ☒ | ☐ ChIP-seq |
| ☒ | ☐ Flow cytometry |
| ☒ | ☐ MRI-based neuroimaging |

## Antibodies

| | |
|---|---|
| Antibodies used | Antibodies are described in the methods section. |
| Validation | Antibodies were validated by band size, and by si/sh suppression of targets. |

# Eukaryotic cell lines

Policy information about cell lines and Sex and Gender in Research

Cell line source(s)

Cell lines were purchased from Coriell cell repository.

Authentication

Commercial c ell lines were authenticated by the provider (Coriell)

Mycoplasma contamination

Our cells are tested for mycoplasma monthly. All cells in use were found negative consistently.

Commonly misidentified lines
(See ICLAC register)

n/a

