## [Peer Review File · Nature]

Manuscript Title: Telomere-to-mitochondria signaling by ZBP1 mediates replicative crisis

Reviewer Comments & Author Rebuttals

Reviewer Reports on the Initial Version:

Referees' comments:

Referee #1 (Remarks to the Author):

This manuscript reports that replicative crisis is (1) characterised by the induction of a type I interferon (IFN) response and (2) that this IFN response and cell death following crisis require ZBP1. These are important and timely findings. The data supporting these conclusions are clear-cut and strong.

The authors then go on to investigate the underlying mechanism. An interesting but largely unexplored observation is that ZBP1 has two isoforms and that only the short isoform ZBP1(S) is required for crisis (see points 2-6 below). Furthermore, dysfunctional telomere-derived lncRNAs associate with ZBP(S) and induction of such RNAs induces ZBP1-dependent IFN and cell death responses (see point 7 below). Finally, the authors suggest that ZBP1 forms mitochondria-associated filaments in crisis cells, that these filaments contain MAVS, and that ZBP1-MAVS signalling explains IFN induction (see points 8-12 below).

These mechanistic conclusions are somewhat provocative; in particular, it is well established that MAVS signals downstream of RIG-I and MDA5. Unfortunately, the experimental support appears insufficient as detailed below (see points 8-12) and more work is required. As the data stand, it is entirely possible that MAVS simply induces a degree of tonic IFN in pre-crisis cells, which results in low levels of cGAS-STING expression. cGAS then detects DNA damage in crisis cells, resulting in ZBP1 upregulation, which drives cell death independently of MAVS, with DNA released from dead cells further amplifying the IFN response via cGAS and not MAVS.

These criticisms should not distract from the fact that unravelling the molecular machinery driving crisis is an important subject and that the discovery of ZBP1-dependency in itself (ie without mechanism) is a transformative finding. In addition, the manuscript is extremely well written and was a pleasure to read.

Major points

1. What is the type of (ZBP1(S)-dependent) cell death in crisis cells? ZBP1 is well known to induce necroptosis, and can also trigger pyroptosis and apoptosis. This should be tested, for example by analysing biochemical markers such as cleaved casp-3 (apoptosis), cleaved GSDMD (pyroptosis) and MLKL phosphorylation (necroptosis).
2. What are the predicted molecular weights of ZBP1(L) and ZBP1(S)? Please show at least one example of a full western blot probed with anti-ZBP1 antibody, including size markers, to confirm that ZBP1(S) runs at the predicted molecular weight. This could be included in Fig S4.

3. What is the effect of ZBP1(L) in the experiments in Fig 1c,d?
4. Does ZBP1(L) show a different cellular localisation pattern than ZBP1(S) during crisis?
5. Would it be possible to devise an RNAi strategy that selectively depletes ZBP1(L) or ZBP1(S)? The former might be achievable with an siRNA in the Za domain sequence and the latter with an siRNA spanning the splice junction unique to ZBP1(S).
6. How is ZBP1 alternative splicing regulated?
7. To proof causality of TERRA RNAs in triggering ZBP1, can such RNAs be targeted by RNAi?
8. The specificities of the ZBP1 and MAVS antibodies used in immunofluorescence in Fig 4 urgently require validation in corresponding KO cells.
9. Since co-localisation does not always translate to direct interaction, can IP experiments be performed to show that ZBP1(S) interacts with MAVS?
10. It is surprising that the ZBP1-MAVS signalling axis proposed here has been missed thus far by the many other groups working on ZBP1. Is it ZBP1(S) specifically that activates MAVS, and not ZBP1(L)? Is this why this has been overlooked?
11. More work in MAVS-deficient cells is needed to test whether MAVS drives IFN induction (e.g. measured by RT qPCRs) and cell death during crisis.
12. Fig S4d/lines 206-207. To exclude RIG-I/MDA5 redundancy, the authors should analyse ZBP1 induction in cells depleted simultaneously of both sensors.

Minor points

13. Line 35. Delete 'formidable'
14. Line 80. Delete 'entirely'
15. Line 89. Delete 'striking'
16. Line 133 please rephrase "barely detectable" – from the blots shown, especially in Extended Fig 4C, the ZBP1(L) protein band is clearly visible
17. Since the manuscript will also be of interest to those outside the field of telomere research, it would be helpful to briefly elaborate on some key aspects of the experimental models. For example, in lines 84-86, why HPV E6E7 and SV40-LT were introduced into the cell lines, and lines 245-246, the rationale or hypothesis regarding the use of mutant TRF1 and V16-TRF1.
18. Fig 1a. Please also show STING, encoded by the TMEM173 gene (nb 29 in Table S1)
19. Lines 273-4 (Cellular imaging of such molecules, however, often reveals aggregated structures rather than long filaments...). Please reference this statement.
20. Rename 'Zbeta' to 'Zalpha2' to avoid confusion with the non-functional Zbeta of Adar1-p150.

Referee #2 (Remarks to the Author):

In this paper, Karlseder and colleagues describe fundamentally new insights into how short telomeres induce cell death during cell crisis. It is demonstrated that during cell crisis, telomeres increase expression of the long noncoding RNA TERRA. TERRA associates with the RNA sensor ZBP1 (Z-DNA binding protein 1) whose expression is also induced by the cGAS-STING DNA sensing machinery. TERRA-activated ZBP1 associates with MAVS at mitochondrial membranes forming filaments, inducing innate immune signaling resulting in cell death by autophagy. Classically, it was assumed that chromosome end-to-end fusions and chromosome rupture and mis-segregation

events would cause cell death during crisis. This paper, however discovers a signaling pathway which is elicited at critically short telomeres by TERRA, which via ZBP1 induces programmed cell death by autophagy.

The described discoveries are paradigm-shifting providing unprecedented insights into how cell crisis suppresses tumorigenesis. The paper is of utmost importance for the understanding of cancer development. The experiments are very convincing and beautifully executed. The writing is also excellent. I am thrilled by this masterpiece.

Minor comments:

- On line 45, when MAVS is first mentioned, it might be helpful for the general readership to briefly provide some background information on MAVS function.
- Figure 1a: I did not understand why there are 3 dots for IFNAR, 4 dots for ZBP1 and only 1 dot for IFNB1. The experiment was done twice. Perhaps information could be provided in the Figure legend.
- Figures 3a and 3b show technical replicates for which statistical analysis is done. Perhaps the biological replicates could also be shown as extended data, if not in the main Figures.
- Figure 3f: The dot sizes could be reduced in order to distinguish the 4 conditions in the Figure.
- Extended data Figure 5b: please define or spell out LE and SE.
- Line 180: delete "they".
- Line 217: Refer here to Figures 3a and b
- Line 218: As no telomere marker is included, I would write: "presumably reflecting TERRA association with telomeres".
- Extended data Figure 8c: I'm not too convinced by the FISH analysis of TERRA. In the protocol, the cells were pre-extracted and cytoplasmic proteins may have been removed. Indeed, the outline of cells is not visible and therefore the localization in the cytoplasm seems speculative. Ref. 39 also reports on extracellular but not cytoplasmic TERRA, if I understand correctly. Thus, I wonder if the claim that TERRA associates with ZBP1 in the cytoplasm is not premature.
- Extended data Figure 14: spell out MN (or define in the legend).

Referee #3 (Remarks to the Author):

Extensive telomere shortening in cells with defective cell cycle checkpoint leads to cell death, a process known as replicative crisis. Nassour et al performed a CRISPR screen to identify genes required for cell death during replicative crisis. They identified ZBP1 as well as components of the cGAS-STING pathway known to activate an innate immune response to DNA. Focusing on ZBP1, they found that a short form of ZBP1, ZBP1(S) was induced during replicative crisis in a manner dependent on cGAS and type-I interferon signaling. Deletion of ZBP1 inhibits cell death as well as induction of interferon-stimulated genes during replicative crisis. Data was presented to suggest that ZBP1 binds to telomere repeat RNA (TERRA), and that deletion of a Z-RNA binding domain (Z-beta) impaired ZBP1's function, suggesting that binding of ZBP1 to TERRA mediates downstream signaling. ZBP1 forms filaments when cells enter crisis and some of these filaments appeared to co-localize with MAVS filaments, which are on the mitochondrial membrane and known to activate the type-I interferon pathway. The authors proposed that during replicative senescence, cytosolic DNA (perhaps in micronuclei) activates cGAS, which induces ZBP1. ZBP1 then associates with TERRA and

form filaments on the mitochondrial surface in a process that depends on MAVS. The ZBP1 filaments activate MAVS, leading to IRF3 activation and induction of interferons, which somehow mediate cell death.

While the finding that ZBP1 is important for replicative crisis is interesting, the mechanism proposed in this paper is not convincing. The authors proposed that ZBP1 activates MAVS, which then mediates cell death through activation of IRF3 and type-I interferons (Extended Data Figure 14). However, MAVS activation of IRF3 and IFNs does not necessarily lead to cell death. If IRF3 and IFNs cause cell death, why is activation of cGAS-STING not sufficient to lead to cell death (Extended Data Figure 14)? There is good evidence that RIPK3 is important for ZBP1-induced cell death; the authors need to look into the simpler model that ZBP1 recruits RIPK3 to cause cell death.

Other points:

- 1) Does transfection of TERRA into ZBP1-expressing cells lead to MAVS-dependent, and RIG-I/MDA5-independent induction of IFNs?
- 2) If ZBP1 is upstream of MAVS, why is ZBP1 filament formation dependent on MAVS? Does ZBP1 filament formation precede MAVS filaments or is this the other way around? If the latter, what causes MAVS filament formation? The temporal order of ZBP1 and MAVS filament formation needs to be dissected in order to evaluate the proposed mechanism.
- 3) What features of TERRA make it capable of activating MAVS in a manner dependent on ZBP1 but not RIG-I or MDA5?
- 4) Is there a way to deplete TERRA in cells and show that it is essential for ZBP1 activation?
- 5) Is there direct biochemical evidence that TERRA causes ZBP1 filament formation?
- 6) ZBP1 knockout mice are available. Do these mice exhibit a phenotype consistent with defective replicative crisis? The mouse telomeres are different from those of humans, but efforts to address the role of ZBP1 in an in vivo model would help to support (or refute) the authors' model.

Referee #4 (Remarks to the Author):

Nassour et al., propose that ZBP1 acts as a sensor for telomere damage-dependent replicative crisis in human lung fibroblasts that have been immortalized with either papillomavirus or polyomavirus oncogenes that overcome cellular senescence. This crisis is known to depend on cGAS-STING (a cytosolic DNA sensor) but here dependence is extended to telomeric repeat-containing RNA (TERRA), where "TERRA-ZBP1 interactions are sufficient to launch a detrimental MAVS-dependent innate immune signaling response". The work represents observational novelty, particularly because it places focus on the short form of ZBP1. As presented, the work is very much underdeveloped and does not demonstrate awareness of existing understanding of ZBP1-dependent death pathways or its recognized partners, particularly RIPK3 and RIPK1. By focusing on oligomerization of ZBP1 "into long filaments that interact and colocalize with MAVS on the outer mitochondrial membrane of a subset of mitochondria" without considering more direct consequences, the reader is left with the distinct impression that not all the important features have been investigated or disclosed. The potential activation of ZBP1 by cGAS-STING signal transduction (the latter already suspected of

contributing to an autophagy-dependent cell death program) leaves unresolved the impact of ZBP1 signal transduction via RIPK3, in particular. Basically, the manuscript follows from possibility to possibility without disclosing concrete and compelling mechanisms that fit with what is already known.

ZBP1 is best known for triggering RIPK3-MLKL-dependent necroptosis and RIPK3-RIPK1-FADD-CASP8-dependent apoptosis (as well as nondeath consequences of component scaffold interactions that leads to activation of cytokines and potentially pyroptosis). Of course, MAVS is a component of the RIG-I-like signaling pathway. Thus, the reader faces data supporting several known innate immune signaling pathways all apparently collaborating in the execution of cells entering crisis. The work overlooks the question of whether this pattern represents known collaboration of NF-kappaB and IRF3, but show that IFN itself does contribute; however, this manuscript fails to unambiguously address its stated goal of demonstrating how “dysfunctional telomeres engage nucleic acid (NA) sensing machineries and activate immune signaling pathways required for crisis-associated cell death”.

Since being studied for senescence, human WI38 fibroblasts (and IMR90 fibroblasts) have been work horses of this field. When immortalized by HPV E6/E7 or SV40 LT antigen (or when immortalized by overexpression of telomerase itself), these cells proliferate beyond the so-called “Hayflick limit” but nevertheless reach a point of replicative crisis that impacts most cells in a culture. This crisis is influenced by cGAS-STING signaling, has characteristics of a type I interferon (IFN) response and results in autophagy-type cell death. Here, IMR90 fibroblasts immortalized with E6/E7 and WI38 cells immortalized with SV40 LT are found to show an increased IFN signature pattern as they reach a passage level just prior to crisis, suggesting that an inflammatory profile driven by the events preceding crisis contribute to crisis. This correlation drives the experimental design, but remains confounding throughout.

ZBP1 has shown the ability to rescue defects in some of the most unexpected settings in mice as well as in human cells and organoids, potentially informing the question at hand with replicative senescence. Starting with observations on the combined role of TNF together with type I IFN in the death of RIPK1-deficient cells and mice, shown to be rescued by the combined elimination of RIPK3 and caspase-8 (Kaiser et al., PNAS, 2014), continuing with observations in RelA-deficient cells and mice (Thapa et al., Mol Cell Biol, 2011, and Xu et al., J Immunol, 2018), and the clear demonstration that ZBP1 is an important sensor that drives the activation of RIPK3 in the absence of RIPK1 or RIPK1 RHIM function (Newton et al., Nature, 2016; Lin et al., Nature 2016; Ingram et al., J Immunol 2019) the likely contribution of interferons in stressful settings seems predictable, particularly in combination with other inflammatory cytokines such as TNF. Observations first by Mandal et al. (Immunity, 2018) on endotoxic shock (a model of sepsis), but followed by Yang et al. (Cell Mol Immunol, 2020) and Karki et al. (Cell, 2021) with systemic inflammatory response syndrome (SIRS) revealed the combined impact of TNF and either type I or type II IFN signaling in driving lethal insult. Other allied work, such as Frank et al. (Cell Death Diff, 2019) showed that cell cycle disrupting cancer drugs known to induce genotoxic stress in cancer cells act in concert with type I IFN, triggering ZBP1-RIPK3-MLKL necroptosis. Also, genome instability-dependent death due to disruption of Setdb1 in mice or human organoids have been rescued by elimination of RIPK3, MLKL or ZBP1 (as well as by pharmaceutical inhibition of RIPK3 kinase activity) as shown Wang et al. (Nature 2020). The

contribution of type I IFN seemed key to activation of ZBP1 such that authors commented, “the induction of cell death in organoids requires not only the upregulation of ZBP1 but also an unknown factor or factors to initiate ZBP1-mediated necroptosis”. In that work they go on to show derepressed endogenous RNA from “large endogenous retroviruses (ERVs)” in both the mouse and human settings as “the other factor”, observations relevant here. Recently published observations of Z-RNA as the ZBP1 ligand in both the influenza (Zhang et al., Cell 2021 – Ref 13) and vaccinia virus (Koehler et al Cell Host Microbe 2021) infection examples raises further concerns as to whether the experimental design here has overlooked some obvious contributors and outcomes from ZBP1-dependent pathways that play out independent of MAVS or other mitochondrial cell death signaling components.

Although data presented in Figure 1 and Extended Data 1 provides evidence consistent with the authors hypothesis, the data are mostly correlative. The connection to actual crisis (where a majority of cell die) needs more complete single cell documentation. From the studies that have made use of human fibroblasts to evaluate ZBP1 contribution to triggering cell death over the past decade or so, two features stand out: (i) as a Z-nucleic acid sensor, ZBP1 is both induced by type I IFN and induces type I IFN and (ii) ZBP1 levels, together with levels of its partner RIPK3, vary dramatically with passage and from cell strain to cell strain in culture, as well as upon immortalization. To repeat a feature already mentioned, once activated, ZBP1 recruitment of RIPK3 has been implicated most consistently in the death of cells from all insults evaluated. Here, ZBP1 is purported to play a sensor role leading to crisis, but the absence of ZBP1 is shown to have a dramatic impact on the basal level of IFN signaling in these cells and that may be as important in the observations as its contribution as a sensor.

Authors find ZBP1-KO cells enjoy a modest number of additional passages without performing the obvious flip side experiment. The reader is not convinced that ZBP1 represents a crucial player with only this level of observation, given “reduced expression and secretion of IFN- β ”, a lower tonic level of IFN signaling seems likely in the absence of ZBP1, meaning any of multiple ISGs could be limiting. If the authors’ viewpoint is correct, conditional expression of ZBP1 expression should be sufficient to trigger crisis at an earlier passage (an experiment that will go much further to demonstrate its critical role as a driver of crisis). If this reader’s interpretation is correct, that overexpression of ZBP1 may provide a basal level of protection. The cells that are entering crisis must be subjected to single cell evaluation to establish whether elevated ZBP1-GFP levels precede death in time-dependent assays within a given cell. The authors contention that “deletion of ZBP1 demonstrated its essential role in crisis” is premature. Indirect impacts on IFN induction as well as IFN signaling are likely confounders here that make it appear ZBP1 contributes when the underlying tonic inflammatory state may be the culprit. Specificity of any of the signaling pathways being implicated here remain all important.

(In 133) As the authors proceed to evaluate type I IFN activation with attention first towards TBK1 and later towards MAVS, contributions of ZBP1 and cGAS-STING must be better distinguished in an unambiguous manner as these data are presented. The number of sensors that can contribute to innate immune activation of type I IFN includes both as well as many more that are not included in this study. In particular, genotoxic stress has long been a topic of study as mentioned above. The conclusion that “cGAS/STING pathway is responsible for an initial transcriptional induction of ZBP1” requires more complete demonstration. Current data does not allow the reader to comprehend the

contributions of these respective pathways or the independence of the signaling outcomes.

(In 142) Authors report that “individual suppression of cGAS, STING, or IFN receptors (IFNAR1 and IFNAR2) resulted in a significant reduction of ZBP1” but fail to evaluate the expected contributions of RHIM signal transduction via RIPK1 or RIPK3.

(In 157) The important ZBP1 Zalpha2 (called Zbeta here) domain here seems reminiscent of the sensing of RNA in MCMV, HSV, VACV and influenza infections, where the activating ligand is now known to be Z-RNA in both influenza and VACV (Zhang et al and Koehler et al.). Also, the most proximal RHIM has been implicated in these systems (starting with Upton et al., Cell Host Microbe 2012, where an MCMV-encoded suppressor of RHIM signaling was reported). In every system that has been studied since, the role of this RHIM has recruitment of RIPK3 and/or RIPK1. Here the ongoing IFN response together with autophagy contribute, but in what order and with what signaling consequences? Authors should make a more sincere effort to describe how hTERT immortalized fibroblasts proceed to crisis, either here or elsewhere because this is a common means of human fibroblast immortalization. Authors must investigate whether a necroptotic (RIPK3-MLKL) outcome or multiple death (RIPK3-MLKL and RIPK3-RIPK1-FADD-CASP8) pathways play out in the presence of viral oncogenes or hTERT to make this complete. Overlooking this enormous body of research while implicating ZBP1 as a sensor of impending crisis has likely left key early steps leading to crisis completely undisclosed. Something similar may be playing out in crisis if, and only if ZBP1 is actually acting as a sensor and not as a regulator of basal interferon levels. Finally, for this data series to be compelling, levels of all mutant forms of ZBP1 must be evaluated and shown on western blots.

The authors' focus on the short form of ZBP1 is based on its abundance in IFN-activated cells, which leaves much to be desired. Little has been done with this form in the past and so one reference is not sufficient to bring this forward as a candidate. Comparison of the two forms must be pursued. Although the short isoform is predicted to lack Zalpha1, nothing concrete has been reported. Thus, authors must be more complete in their evaluation of this aspect of ZBP1 here or leave it for a subsequent follow-up study.

(In 174) The fact that “depletion of ZBP1 dampened the TBK1-IRF3 signaling axis, reduced ISG expression, and attenuated autophagy” may again be an indirect consequence of the impact ZBP1 has on basal/tonic levels rather than a direct role in the final steps of crisis. The observation that hTERT “resulted in telomere maintenance and continued growth past the crisis barrier” although these cells “did not express ZBP1, nor they did they activate IFN signaling” comes again to the chicken and egg question raised repeatedly above. For this section to be more compelling, conditional expression should be employed to bring on ZBP1 at various times in order to determine any direct contribution at different passage level as cells approach crisis. Control should include the Zalpha2 mutant reconstituted cells.

(In 199) This section of the manuscript dealing with TERRA as a ligand for ZBP1 represents potentially the most interesting topic in the manuscript, particularly with the previous implications as a ligand for RNA sensors. This section must also be complete show compelling evidence that TERRA interaction with ZBP1 triggers activation through additional complementary methods. Such a

strategy has already been accomplished in other settings recently published (Zhang et al., and Koehler et al.) that more completely dissect the direct role of Z-RNA interactions with ZBP1. The experiments in the submitted manuscript seem to show that the RIG-I-like receptors are not involved with MAVS activation, which runs contrary to a large body of work, so must also be presented in a more compelling manner. The work must distinguish from nonspecific impacts such as those described by the Lieberman lab with their exosome work. CD data showing this RNA is Z-form will be needed above and beyond dot blots. Authors must more completely address interactions natural or synthetic TERRA binds recombinant Zalpha beyond the immunoprecipitations shown in the submitted manuscript for this all to reach an objective conclusion.

(In 270) The authors' suggestion that TERRA-activated ZBP1 SHORT drives "homotypic RHIM1 domain interactions" forming "aggregated structures rather than long filaments" that "then translocate to mitochondria to interact and activate MAVS" must include more compelling data. To begin, this process must be shown to be independent of other RHIM-containing adaptors (RIPK1, RIPK3 and TRIF), and to precede their activation when they are present. The physical nature of the oligomer must be dissected through biophysical assessment. The manuscript requires considerable cell free work with ZBP1, ZBP1 mutants and TERRA RNA to get anywhere close to the conclusions currently being posited in this manuscript. At this point it is unclear if the oligomer is a consequence rather than a cause of crisis. The reader is left wondering "Is ZBP1 promote MAVS oligomerization" or "Is MAVS promoting ZBP1 fibril formation? (Or is this all a consequence of ZBP1 recruitment of RIPK3?).

Finally, it is completely unclear whether the observations that are presented here for immortal WI38 and IMR90 are broadly relevant and generalizable. Many questions come to mind but are left unanswered.... Do normal (pre-senescent) fibroblasts have any of these characteristics as they reach senescence? Do fully transformed fibroblasts completely avoid this phenomenon? Are there natural settings where crisis occurs in a cancer cell progression that relate to these stages with diploid fibroblasts? This reader's suspicion remains that this series of observations describes a particular correlative phenomenon in the two cell lines chosen. Nothing more.

Most importantly, and also related to impact, autophagy remains a highly controversial mode of death regardless of the number of reports that implicate this pathway as a manner in which cells die. Given ZBP1 kills by far more established and direct programmed modes, the data must explore those conventional apoptotic and necroptotic mechanisms driven off of the same RHIM of ZBP1 as implicated here. Timing of events are likely key. Given that crisis occurs several passages beyond the point where experiments can be carried out, time course evaluation of single cells appears to be key to objective experimentation along with the need to recognize that ZBP1 is already known to activate RIPK3, MLKL, RIPK1, FADD and CASP8 signaling.

Author Rebuttals to Initial Comments:

We thank the reviewers for their time and critical feedback on our manuscript. We have responded in earnest to their comments with new data to support our main conclusions, which significantly improved the manuscript. Below, is our point-by-point response to each reviewer. New data have been included in the main figures and extended data figures (as indicated). In addition, due to space limitations, some data are only included in this response letter for the referees to consider.

Referee#1 (Remarks to the Author):

This manuscript reports that replicative crisis is (1) characterised by the induction of a type I interferon (IFN) response and (2) that this IFN response and cell death following crisis require ZBP1. These are important and timely findings. The data supporting these conclusions are clear-cut and strong. The authors then go on to investigate the underlying mechanism. An interesting but largely unexplored observation is that ZBP1 has two isoforms and that only the short isoform ZBP1(S) is required for crisis (see points 2-6 below).

Furthermore, dysfunctional telomere-derived lncRNAs associate with ZBP(S) and induction of such RNAs induces ZBP1-dependent IFN and cell death responses (see point 7 below). Finally, the authors suggest that ZBP1 forms mitochondria-associated filaments in crisis cells, that these filaments contain MAVS, and that ZBP1-MAVS signalling explains IFN induction (see points 8-12 below). These mechanistic conclusions are somewhat provocative; in particular, it is well established that MAVS signals downstream of RIG-I and MDA5. Unfortunately, the experimental support appears insufficient as detailed below (see points 8-12) and more work is required. As the data stand, it is entirely possible that MAVS simply induces a degree of tonic IFN in pre-crisis cells, which results in low levels of cGAS-STING expression. cGAS then detects DNA damage in crisis cells, resulting in ZBP1 upregulation, which drives cell death independently of MAVS, with DNA released from dead cells further amplifying the IFN response via cGAS and not MAVS.

These criticisms should not distract from the fact that unravelling the molecular machinery driving crisis is an important subject and that the discovery of ZBP1-dependency in itself (ie without mechanism) is a transformative finding. In addition, the manuscript is extremely well written and was a pleasure to read.

Response: We thank the referee for acknowledging the novelty and impact of our findings, and for the constructive criticism, our responses to which provided additional mechanistic insights into the ZBP1-driven cell death pathway during replicative crisis.

Major points

1. What is the type of (ZBP1(S)-dependent) cell death in crisis cells? ZBP1 is well known to induce necroptosis and can also trigger pyroptosis and apoptosis. This should be tested, for example by analysing biochemical markers such as cleaved casp-3 (apoptosis), cleaved GSDMD (pyroptosis) and MLKL phosphorylation (necroptosis).

Response: We agree with the referee and apologize for the omission of additional data to this point. ZBP1 is a RHIM domain-containing protein, which interacts with RIPK3 and mediates various forms of cell death in response to viral infections (IAV, HSV, MCMV, and VV). Upon sensing viral RNAs or viral ribonucleoprotein (vRNP) complexes, ZBP1 (full-length) associates with RIPK3 to drive parallel cell death pathways of apoptosis, pyroptosis, and necroptosis (PANoptosis). Indeed, the recruitment and activation of RIPK3 by ZBP1 (full-length) is essential for MLKL-driven necroptosis, FADD-mediated apoptosis, and NLRP3 inflammasome activation

and Gasdermin D-driven pyroptosis¹⁻⁹. It is important to note however, that these studies, were performed primarily using rodent cell lines or models, which all appear to highly express both RIPK3 and ZBP1 (full-length), but entirely lack ZBP1(S).

To address the concern, we performed a series of experiments to characterize the type of ZBP1(S)-dependent cell death in human fibroblasts and found no significant activation of the pyroptotic, apoptotic and necroptotic pathways upon expression of ZBP1(S) (Extended Data Figure 6a-c) or in crisis cells (Extended Data Figure 6e-g). Indeed, markers of apoptosis (cleaved caspase 3, cleaved caspase 7, and cleaved PARP1), markers of necroptosis (phosphorylation of RIPK3 and its downstream effector, MLKL), and markers of pyroptosis (cleaved caspase 1 and its downstream substrates gasdermin D (GSDMD) and Interleukin-1 β (IL-1 β)) were not detected during replicative crisis or upon expression of ZBP1(S) in growing cells. Notably, RIPK3 levels were very low when compared to the necroptosis prone cell line HT-29 (Extended Data Figure 6b). On the other hand, LC3 lipidation and p62 degradation were observed (Extended Data Figure 6d,h), consistent with an autophagy-dependent cell death pathway (Nassour et al., *Nature* **565**, 659–663 (2019)). Notably, RIPK3 levels were very low when compared to necroptosis prone cell line HT-29.

Furthermore, we used siRNAs to deplete the essential autophagy related protein ATG7, as well as type I interferon (IFN) receptors (IFNAR1 and IFNAR2), ZBP1, and RIPK3 and measured the frequency of cell death during crisis using the Incucyte Live-Cell Analysis system. The frequency of cell death was significantly reduced upon suppression of ZBP1, IFNAR, and ATG7. On the other hand, no major effect was observed upon depletion of RIPK3 (Extended Data Figure 6i-j).

Thus, our data support a model where, in primary human cells, ZBP1(S) does not activate intrinsic PANoptosis through RIPK3, but rather drives a cell death signaling pathway that is dependent of IFN and autophagy.

2. What are the predicted molecular weights of ZBP1(L) and ZBP1(S)? Please show at least one example of a full western blot probed with anti-ZBP1 antibody, including size markers, to confirm that ZBP1(S) runs at the predicted molecular weight. This could be included in Fig S4.

Response: As differences between theoretically predicted and SDS PAGE-displayed molecular weights of proteins are often seen, we performed a set of experiments to address this point:

- 1) *Pre-crisis IMR90 E6E7 were transduced with two different shRNA lentiviral vectors:
-shZBP1#1 targeting a sequence within the Z α 1 domain and this resulted in a depletion of ZBP1(L) without affecting ZBP1(S) levels (Extended Data Figure 4d).
-shZBP1#2 targeting a sequence present in both ZBP1(L) and ZBP1(S), and this resulted in a depletion of both isoforms (Extended Data Figure 4d). The uncropped gels are shown.*

The following data have only been included for the referee in this response letter:

- 2) *We expressed a doxycycline (dox)-inducible ZBP1(L) and ZBP1(S) in growing IMR90 E6E7 (with low levels of endogenous ZBP1). Uncropped membranes were probed with ZBP1 antibody to observe the exact MW of ZBP1(S) and ZBP1(L). ZBP1(L) appears to run slightly above 50kDa while ZBP1(S) below 50kDa (Referee 1 – Figure 1a). Growing IMR90 E6E7 were treated with IFN- β and both bands corresponding to ZBP1(S) and ZBP1(L) were induced (Referee 1 – Figure 1a). The uncropped gels are shown.*

- 3) Pre-crisis IMR90 E6E7 were transfected with a pool of siRNA targeting ZBP1 and both bands corresponding to ZBP1(S) and ZBP1(L) were reduced. ZBP1 signal reduction was also confirmed by immunofluorescence (Referee 1 – Figure 1b-c). The uncropped gels are shown.

a

b

c

Referee1 - Figure1: a, Left panel: Immunoblotting of growing (PD40) IMR90 E6E7 expressing doxycycline-inducible ZBP1(L) or ZBP1(S). Protein extracts were collected 48 hours after doxycycline treatment (1 µg/mL). GAPDH loading control. Right panel: Immunoblotting of growing (PD40) IMR90 E6E7 treated or not with the recombinant human IFN-β (500U/ml). Protein extracts were collected 72 hours after IFN-β treatment. GAPDH loading control. b, Representative microscopy images of pre-crisis IMR90 E6E7 (PD100) transfected with either siZBP1(pool), siMAVS(pool), or non-targeting control siRNA (pool). Cells were immunostained with antibodies against either endogenous ZBP1 or MAVS 72 hours after siRNA transfection. Scale bar 10 µm. c, Immunoblotting of pre-crisis IMR90 E6E7 (PD100) transfected with siRNA as in b. Protein extracts were collected 72 hours after siRNA transfection. GAPDH loading control. Abbreviations: ZBP1(S): short isoform; ZBP1(L): long isoform; CTR: control; Dox: doxycycline; PD: population doubling.

3. What is the effect of ZBP1(L) in the experiments in Fig 1c,d?
4. Does ZBP1(L) show a different cellular localisation pattern than ZBP1(S) during crisis?

Response: We have addressed the referees' questions below. However, we would like to point out that current study focuses on ZBP1(S), the isoform that is most associated with replicative crisis. ZBP1(L) levels are weak and often difficult to detect during crisis. We therefore prefer not to include the data with ZBP1(L) in the main manuscript, but rather only in the response here:

To address the referee's questions, ZBP1 KO IMR90 E6E7 were reconstituted with either ZBP1(L) or ZBP1(S) and tested for their susceptibility to induce ISGs and undergo cell death (Referee 1 – Figure 2a-c). As expected, ZBP1(S) showed increased expression of ISGs concomitant with elevated cell death. Those reconstituted with ZBP1(L) also showed an increase of ISG levels and cell death frequency, though to a much lesser extent than ZBP1(S) (Referee 1 – Figure 2a-c).

Moreover, ZBP1(L) and ZBP1(S) displayed different subcellular localizations, consistent with previous reports showing that Za1 and Za2 domains play roles in determining the subcellular localization of ZBP1¹⁰ (Referee 1 – Figure 2d). While ZBP1(L) showed a nuclear and cytoplasmic diffuse staining, ZBP1(S) accumulated in large cytoplasmic granules and formed filaments. The ability of ZBP1 to form filaments on a subset of mitochondria could explain why the amplitude of the immune response is significantly higher when ZBP1(S) is induced than ZBP1(L). Further research is however required to clarify this point.

5. Would it be possible to devise an RNAi strategy that selectively depletes ZBP1(L) or ZBP1(S)? The former might be achievable with an siRNA in the Za domain sequence and the latter with an siRNA spanning the splice junction unique to ZBP1(S).

Response: As requested by the reviewer, we have now developed an shRNA-based strategy to deplete specifically ZBP1(L) without affecting ZBP1(S) (Extended Data Figure 4d). Since 100% of ZBP1(S) is contained in ZBP1(L), we were not able to specifically deplete ZBP1(S). These data were used to validate the specificity of the antibody (see point 2) and are now included in the manuscript.

6. How is ZBP1 alternative splicing regulated?

Response: The ZBP1 variants are an effect of alternative splicing and the use of alternative transcriptional start and stop sites. As a result, the ZBP1 mRNAs are quite heterogeneous and exon 2, coding for the first Za1 domain, appears to be spliced out in ~50% of ZBP1 mRNA^{10,11}. We have not addressed the exact regulation of the isoform generation in replicative crisis and hope the referee agrees with us that this exceeds the immediate scope of this manuscript.

7. To proof causality of TERRA RNAs in triggering ZBP1, can such RNAs be targeted by RNAi?

Response: To demonstrate a causal role of TERRA in ZBP1-dependent immune response, we used two individual siRNAs allowing us to partially suppress TERRA in cells co-expressing WT ZBP1(S) and VP16-TRF1 (Extended Data Figure 12e-g). These two siRNAs were found previously to partially deplete TERRA RNA levels up to ~40% as measured by RNA dot blot and real-time PCR¹². Partial TERRA depletion was sufficient to delay and attenuate STAT1 activation when compared to control cells (transfected with non-targeting siRNA). These results further validate the immunostimulatory properties of TERRA-derived RNA species and their potential in inducing a ZBP1-dependent immune response (Extended Data Figure 12e-g).

Referee1 – Figure2: a, Immunoblotting of pre-crisis (PD97) ZBP1 KO MR90 E6E7 (clone 67) reconstituted with either WT ZBP1(L)-Flag or WT ZBP1(S)-Flag. Protein extracts were collected 14 days after transduction. GAPDH loading control. **b**, Left: Representative IncuCyte images of IMR90 E6E7 as in **a** after 24 hours of incubation. Dead cells are labeled with Cytotox green and nuclei with NucLight red. Right: Scatter plots with bars showing IncuCyte analysis of cell death of pre-crisis IMR90 E6E7 as in **a** after 24 hours of incubation. Empty vector expressing cells are used as controls. Bars represent mean \pm s.d. from technical replicates. One-way ANOVA, ns: not significant, * $p < 0.05$, ** $p < 0.01$, *** $p < 0.001$. **c**, Scatter plots showing RT-qPCR analysis of ISGs in IMR90 E6E7 as in **a**. RNA extracts were collected 14 days after transduction. Expression levels were normalized to control cells with an empty vector. Bars represent mean \pm s.d. from technical replicates. n : number of technical replicates. One-way ANOVA, ns: not significant, * $p < 0.05$, ** $p < 0.01$, *** $p < 0.001$. **d**, Representative microscopy images of IMR90 E6E7 as in **a**. Cells were immunostained with antibody against Flag 14 days after transduction. Scale bar 10 μ m. Abbreviations: ZBP1(S): short isoform; ZBP1(L): long isoform; KO: knockout, PD: population doubling.

9. Since co-localisation does not always translate to direct interaction, can IP experiments be performed to show that ZBP1(S) interacts with MAVS?

Response: To examine the interaction between ZBP1(S) and MAVS, we performed two independent co-immunoprecipitation (co-IP) experiments:

1-IP ZBP1(S)-flag from growing and crisis IMR90E6E7 and search for MAVS signal by WB.

2-IP ZBP1(S)-flag from growing IMR90E6E7 with dysfunctional telomeres devoid of TRF2 and search for MAVS signal by WB.

In both settings, we were unable to observe MAVS signals in the ZBP1-flag immunoprecipitates.

As the referee is of course aware, co-IP methods tend to be more suitable for stable protein-protein interactions as the protein complex does not disassemble over time, while weak or transient interactions requires alternative methods with a crosslinking step to freeze the interaction during the pull-down. It is also important to mention that the fraction of ZBP1 filaments at mitochondria (colocalizing with MAVS) is relatively low when compared to the cytosolic fraction (not colocalizing with MAVS), making the detection by WB not sensitive enough to reveal MAVS signals. Therefore, the absence of MAVS co-precipitate signals does not necessarily reflect a lack of physical interaction with ZBP1.

However, since we were not able to experimentally demonstrate a direct ZBP1-MAVS interaction, we instead focused on the mechanism of the interaction for inflammation. We were able to demonstrate the dependency of ZBP1-mediated immune signaling on MAVS (now in Figure d-g and Extended Data Figure 16). To achieve this, ZBP1(S) was fused to the outer mitochondrial membrane targeting sequences of FIS1 in growing cells with functional telomeres and low TERRA levels. Under these circumstances, ZBP1(S) colocalized perfectly with MAVS (also on the outer mitochondrial membrane) and triggered an innate immune response marked by elevated ISG transcript levels. Suppression of MAVS prevented ISG induction in response to forced ZBP1 mitochondrial localization, indicating that ZBP1-driven innate immune signaling requires MAVS.

10. It is surprising that the ZBP1-MAVS signalling axis proposed here has been missed thus far by the many other groups working on ZBP1. Is it ZBP1(S) specifically that activates MAVS, and not ZBP1(L)? Is this why this has been overlooked?

Response: We can only speculate why our discovery has been missed previously: Almost all previous studies were done using mouse cells and models. However, both ZBP1 and TERRA are quite differently regulated in mice and humans. Mice do not express the ZBP1(S) allele that is relevant for crisis^{13,14}, mouse telomeres do not undergo replication associated shortening and thus crisis does not serve as proliferative barrier in mice, and TERRA transcription is not driven through telomere dysfunction, but through the pseudoautosomal PAR locus¹⁵. Additionally, our data indicate that ZBP1(L) does not form filaments at mitochondria and most likely does not activate MAVS efficiently. We therefore suggest that we have discovered a pathway that evolved specifically as tumor suppressor in humans along with replicative telomere shortening. Lastly, we discovered this role of ZBP1 via our unique screen for factors required for cell death in crisis, which to our knowledge is the first of its kind.

11. More work in MAVS-deficient cells is needed to test whether MAVS drives IFN induction (e.g. measured by RT qPCRs) and cell death during crisis.

Response: We agree completely and thus used CRISPR to delete MAVS or STING in pre-crisis IMR90E6E7 and performed growth curve and whole transcriptome analysis (now in Extended Data Figure 16a-c). CRISPR-mediated deletion of MAVS demonstrated its essential role in cell

death during crisis, thereby validating our CRISPR screen data. Control cells (sgGFP) entered crisis around PD102 when cell death was frequent and replicative capacity reduced. However, pooled MAVS and STING knock-out cells continued to proliferate and bypassed the crisis plateau, along with a remarkable reduction of cell death and maintenance of growth potential. Accordingly, these cells showed a reduced expression of ISGs and impaired IFN activity (Extended Data Figure 16a-c). These results demonstrate clearly that cell death and immune responses during crisis are also dependent on MAVS.

12. Fig S4d/lines 206-207. To exclude RIG-I/MDA5 redundancy, the authors should analyse ZBP1 induction in cells depleted simultaneously of both sensors.

Response: We repeated the siRNA screen and depleted RIG-I and MDA5 simultaneously. In contrast to MAVS suppression, co-depletion of RIG-I and MDA5 did not result in a significant reduction in ZBP1 mRNA levels in crisis (Extended Data Figure 4e).

Minor points

13. Line 35. Delete 'formidable'

14. Line 80. Delete 'entirely'

15. Line 89. Delete 'striking'

16. Line 133 please rephrase "barely detectable" – from the blots shown, especially in Extended Fig 4C, the ZBP1(L) protein band is clearly visible

Response: We deleted (formidable, entirely, striking) and rephrased barely detectable.

17. Since the manuscript will also be of interest to those outside the field of telomere research, it would be helpful to briefly elaborate on some key aspects of the experimental models. For example, in lines 84-86, why HPV E6E7 and SV40-LT were introduced into the cell lines, and lines 245-246, the rationale or hypothesis regarding the use of mutant TRF1 and V16-TRF1.

Response: We provided additional details on the experimental models.

18. Fig 1a. Please also show STING, encoded by the TMEM173 gene (nb 29 in Table S1)

Response: We highlighted STING in Fig1a.

19. Lines 273-4 (Cellular imaging of such molecules, however, often reveals aggregated structures rather than long filaments...). Please reference this statement.

Response: We referenced this statement.

20. Rename 'Zbeta' to 'Zalpha2' to avoid confusion with the non-functional Zbeta of Adar1-p150.

Response: We renamed 'Zbeta' to 'Zalpha2' to avoid confusion.

References:

1. Muendlein, H. I. et al. ZBP1 promotes LPS-induced cell death and IL-1 β release via RHIM-mediated interactions with RIPK1. *Nat. Commun.* 12, 86 (2021).
2. Kesavardhana, S. et al. ZBP1/DAI ubiquitination and sensing of influenza vRNPs activate programmed cell death. *J. Exp. Med.* 214, 2217–2229 (2017).

3. Kuriakose, T. et al. ZBP1/DAI is an innate sensor of influenza virus triggering the NLRP3 inflammasome and programmed cell death pathways. *Sci. Immunol.* 1, aag2045 (2016).
4. Upton, J. W., Kaiser, W. J. & Mocarski, E. S. DAI/ZBP1/DLM-1 Complexes with RIP3 to Mediate Virus-Induced Programmed Necrosis that Is Targeted by Murine Cytomegalovirus vIRA. *Cell Host Microbe* 11, 290–297 (2012).
5. Ingram, J. P. et al. ZBP1/DAI Drives RIPK3-Mediated Cell Death Induced by IFNs in the Absence of RIPK1. *J. Immunol.* 203, 1348–1355 (2019).
6. Zhang, T. et al. Influenza Virus Z-RNAs Induce ZBP1-Mediated Necroptosis. *Cell* 180, 1115–1129.e13 (2020).
7. Nogusa, S. et al. RIPK3 Activates Parallel Pathways of MLKL-Driven Necroptosis and FADD-Mediated Apoptosis to Protect against Influenza A Virus. *Cell Host Microbe* 20, 13–24 (2016).
8. Thapa, R. J. et al. DAI Senses Influenza A Virus Genomic RNA and Activates RIPK3-Dependent Cell Death. *Cell Host Microbe* 20, 674–681 (2016).
9. Maelfait, J. et al. Sensing of viral and endogenous RNA by ZBP1/DAI induces necroptosis. *EMBO J.* 36, 2529–2543 (2017).
10. Deigendesch, N., Koch-Nolte, F. & Rothenburg, S. ZBP1 subcellular localization and association with stress granules is controlled by its Z-DNA binding domains. *Nucleic Acids Res.* 34, 5007–5020 (2006).
11. Rothenburg, S., Schwartz, T., Koch-Nolte, F. & Haag, F. Complex regulation of the human gene for the Z-DNA binding protein DLM-1. *Nucleic Acids Res.* 30, 993–1000 (2002).
12. Deng, Z., Norseen, J., Wiedmer, A., Riethman, H. & Lieberman, P. M. TERRA RNA Binding to TRF2 Facilitates Heterochromatin Formation and ORC Recruitment at Telomeres. *Mol. Cell* 35, 403–413 (2009).
13. Devos, M. et al. Sensing of endogenous nucleic acids by ZBP1 induces keratinocyte necroptosis and skin inflammation. *J. Exp. Med.* 217, e20191913 (2020).
14. Jiao, H. et al. Z-nucleic-acid sensing triggers ZBP1-dependent necroptosis and inflammation. *Nature* 580, 391–395 (2020).
15. Viceconte, N. et al. PAR-TERRA is the main contributor to telomeric repeat-containing RNA transcripts in normal and cancer mouse cells. *RNA N. Y.* N 27, 106–121 (2021).

Referee#2 (Remarks to the Author):

In this paper, Karlseder and colleagues describe fundamentally new insights into how short telomeres induce cell death during cell crisis. It is demonstrated that during cell crisis, telomeres increase expression of the long noncoding RNA TERRA. TERRA associates with the RNA sensor ZBP1 (Z-DNA binding protein 1) whose expression is also induced by the cGAS-STING DNA sensing machinery. TERRA-activated ZBP1 associates with MAVS at mitochondrial membranes forming filaments, inducing innate immune signaling resulting in cell death by autophagy. Classically, it was assumed that chromosome end-to-end fusions and chromosome rupture and mis-segregation events would cause cell death during crisis. This paper, however discovers a signaling pathway which is elicited at critically short telomeres by TERRA, which via ZBP1 induces programmed cell death by autophagy. The described discoveries are paradigm-shifting providing unprecedented insights into how cell crisis suppresses tumorigenesis. The paper is of utmost importance for the understanding of cancer development. The experiments are very convincing and beautifully executed. The writing is also excellent. I am thrilled by this masterpiece.

Response: We thank the referee for describing our work as paradigm-shifting and unprecedented, as well as for constructive criticism to help us improve the manuscript further. We are humbled by the reference of our work as a 'masterpiece', which we consider a once-in-a-lifetime accolade! We have now addressed the referee's concerns point-by-point and included these changes in the main figures, extended data figures, and this response as supplementary figures.

Minor comments:

-On line 45, when MAVS is first mentioned, it might be helpful for the general readership to briefly provide some background information on MAVS function.

Response: As requested, we added background information on MAVS function.

-Figure 1a: I did not understand why there are 3 dots for IFNAR, 4 dots for ZBP1 and only 1 dot for IFNB1. The experiment was done twice. Perhaps information could be provided in the Figure legend.

Response: We thank the reviewer for pointing this out and we have now clarified this in the figure legend and method section. The dots represent individual gRNAs and the X-Y axis represent the fold change of gRNA counts in the two technical replicates (X axis being one replicate, and Y axis another replicate). The library we used contains 4 different gRNAs/gene with different knockout efficiencies, explaining why not all the 4 gRNAs of each hit are enriched in both replicates of the screen. In the new graph, we are showing the gRNAs with a log₂ fold change > 2 in both replicates. Here, for example, we only detect two gRNAs against ZBP1, while the other remaining two gRNAs do not appear as significantly enriched (most likely because the lower knockout efficiency).

-Figures 3a and 3b show technical replicates for which statistical analysis is done. Perhaps the biological replicates could also be shown as extended data, if not in the main Figures.

Response: We generate graphs for the two other biological RT-qPCR replicates and included them in the response for the referee here (Referee 2 – Figure 1). The raw data with the three biological replicates will be included as source data in the manuscript.

Referee2 – Figure1: a-d, Scatter plot with bars showing RT-qPCR analysis with 5 pairs of subtelomere-specific primers designed to measure TERRA transcribed from individual chromosome ends. RNA extracts were collected from growing IMR90 E6E7, pre-crisis IMR90 E6E7, and growing IMR90 E6E7 expressing non-targeting control shRNA (sh-scramble) or shRNA against TRF2. RNA extracts were collected at day 4 post-shRNA transduction. Expression levels were normalized to growing cells (a-b) and sh-scramble cells (c-d). The PDs are indicated. Bars represent mean \pm s.d. from technical replicates. Biological replicates are provided in Source data. n: number of technical replicates. One-way ANOVA, ns: not significant, * $p < 0.05$, ** $p < 0.01$, *** $p < 0.001$. Abbreviations: PD: population doubling.

-Figure 3f: The dot sizes could be reduced in order to distinguish the 4 conditions in the Figure.

Response: We updated the graph and reduced the dot sizes.

-Extended data Figure 5b: please define or spell out LE and SE.

Response: We defined LE and SE.

-Line 180: delete “they”.

Response: We deleted “they”.

-Line 217: Refer here to Figures 3a and b

Response: We referred to Figures 3a and b.

-Line 218: As no telomere marker is included, I would write: “presumably reflecting TERRA association with telomeres”.

-Extended data Figure 8c: I’m not too convinced by the FISH analysis of TERRA. In the protocol, the cells were pre-extracted and cytoplasmic proteins may have been removed. Indeed, the outline of cells is not visible and therefore the localization in the cytoplasm seems speculative. Ref. 39 also reports on extracellular but not cytoplasmic TERRA, if I understand correctly. Thus, I wonder if the claim that TERRA associates with ZBP1 in the cytoplasm is not premature.

Response: We thank the reviewer for pointing this out and agree with the concerns. The pre-extraction step used in the FISH protocol might result in the leakage of nuclear TERRA into the cytoplasm. In addition, no marker has been used to outline the cells. We therefore decided to not include the FISH experiment in the revised manuscript as the interaction between TERRA and ZBP1 is shown in two other independent experiments (ZBP1-RIP followed by Illumina sequencing and ZBP1-RIP followed by TERRA dot blot).

-Extended data Figure 14: spell out MN (or define in the legend).

Response: We defined MN.

Referee #3 (Remarks to the Author):

Extensive telomere shortening in cells with defective cell cycle checkpoint leads to cell death, a process known as replicative crisis. Nassour et al performed a CRISPR screen to identify genes required for cell death during replicative crisis. They identified ZBP1 as well as components of the cGAS-STING pathway known to activate an innate immune response to DNA. Focusing on ZBP1, they found that a short form of ZBP1, ZBP1(S) was induced during replicative crisis in a manner dependent on cGAS and type-I interferon signaling. Deletion of ZBP1 inhibits cell death as well as induction of interferon-stimulated genes during replicative crisis. Data was presented to suggest that ZBP1 binds to telomere repeat RNA (TERRA), and that deletion of a Z-RNA binding domain (Z-beta) impaired ZBP1's function, suggesting that binding of ZBP1 to TERRA mediates downstream signaling. ZBP1 forms filaments when cells enter crisis and some of these filaments appeared to co-localize with MAVS filaments, which are on the mitochondrial membrane and known to activate the type-I interferon pathway. The authors proposed that during replicative senescence, cytosolic DNA (perhaps in micronuclei) activates cGAS, which induces ZBP1. ZBP1 then associates with TERRA and form filaments on the mitochondrial surface in a process that depends on MAVS. The ZBP1 filaments activate MAVS, leading to IRF3 activation and induction of interferons, which somehow mediate cell death.

While the finding that ZBP1 is important for replicative crisis is interesting, the mechanism proposed in this paper is not convincing. The authors proposed that ZBP1 activates MAVS, which then mediates cell death through activation of IRF3 and type-I interferons (Extended Data Figure 14). However, MAVS activation of IRF3 and IFNs does not necessarily lead to cell death. If IRF3 and IFNs cause cell death, why is activation of cGAS-STING not sufficient to lead to cell death (Extended Data Figure 14)? There is good evidence that RIPK3 is important for ZBP1-induced cell death; the authors need to look into the simpler model that ZBP1 recruits RIPK3 to cause cell death.

Response: We thank the referee for the constructive criticism, our responses to which have greatly improved the manuscript. We agree with the referee's concern (shared by other reviewers) about whether the ZBP1(S)-mediated cell death pathway relies on the serine/threonine-protein kinase 3 (RIPK3) and PANoptosis, as it has been shown previously in mouse cells and models. Indeed, active ZBP1 (full-length) interacts with RIPK3 to form cell death signaling scaffolds known as PANoptosome and engage an NLRP3 inflammasome-dependent pyroptosis, CASP8-mediated apoptosis, and MLKL-driven necroptosis. These studies, however, were performed primarily on cell lines from rodents that constitutively express RIPK3 and therefore exist in an PANoptosis-prone state susceptible of cell death through this pathway. Moreover, the potential contribution of ZBP1 in cell death in human cells is limited to the use of the necroptosis-sensitive human HT-29 cells, a cell line that highly expresses RIPK3.

*We have now performed additional experiments to characterize the type of cell death driven by ZBP1(S) in human fibroblasts and found no detectable activation of either apoptosis, necroptosis, or pyroptosis along with undetectable or low levels of RIPK3. We observe however an increase in the autophagic flux, marked by LC3 lipidation and p62 degradation. These findings are consistent with an autophagy-dependent cell death program (Nassour et al., Nature **565**, 659–663 (2019)) (Extended Data Figure 6a-h). Furthermore, we used siRNAs to deplete the essential autophagy related protein ATG7, as well as type I interferon (IFN) receptors (IFNAR1 and IFNAR2), ZBP1, and RIPK3 and measured the frequency of cell death during crisis using the Incucyte Live-Cell Analysis system. The frequency of cell death was significantly reduced upon suppression of ZBP1, IFNAR, and ATG7. On the other hand, depleting RIPK3 did not prevent or attenuate cell death during crisis (Extended Data Figure 6i-j). Thus, our data support a model*

where, in primary human cells, ZBP1(S) does not activate intrinsic PANoptosis through RIPK3, but rather drives a cell death signaling pathway that is dependent of IFN and autophagy.

The cytotoxic potential of IFNs has been reported in numerous studies, and many cytokines, including IFNs, potentiate cell death in a context-dependent manner¹⁻³. We showed previously that the DNA-sensing cGAS-STING pathway plays a crucial role in autophagy-mediated cell death during crisis (Nassour et al., *Nature* **565**, 659–663 (2019). Importantly, here, we propose that cell death in crisis requires the combined actions of both cGAS-STING and ZBP1-MAVS signaling, that together contribute to robust IFN response beyond a critical threshold and/or to enact expression of a unique set of ISGs and autophagy-related cell death mediators for this purpose. Furthermore, this process is independent of apoptosis, necroptosis, or pyroptosis.

Other points:

1) Does transfection of TERRA into ZBP1-expressing cells lead to MAVS-dependent, and RIG-I/MDA5-independent induction of IFNs?

Response: As the referee has suggested, we have now induced TERRA by expressing TRF1-VP16 in cells lacking MAVS, RIG-I, MDA5, or both RIG-I and MDA5. While MAVS depletion prevented TERRA-induced IFN response, depletion of RIG-I and MDA5 had no effect of IFN β and ISG expression levels. These data are now included in Extended Data Figure 15.

2) If ZBP1 is upstream of MAVS, why is ZBP1 filament formation dependent on MAVS? Does ZBP1 filament formation precede MAVS filaments or is this the other way around? If the latter, what causes MAVS filament formation? The temporal order of ZBP1 and MAVS filament formation needs to be dissected in order to evaluate the proposed mechanism.

Response: We agree with the referee on the importance of the ZBP1-MAVS relation. Focusing on the mechanism of the interaction for inflammation, we have now demonstrated the dependency of the ZBP1-stimulated innate immune signaling on MAVS (now in Figure 4d-g and Extended Data Figure 16). To achieve this, ZBP1(S) was fused to the outer mitochondrial membrane targeting sequences of FIS1 in growing cells with functional telomeres. Under these circumstances, ZBP1(S) colocalized perfectly with MAVS (also on the outer mitochondrial membrane) and triggered an innate immune response marked by elevated ISG transcript levels. Suppression of MAVS prevented ISG transcription in response to forced ZBP1 mitochondrial localization, indicating that ZBP1-driven immune response necessitates MAVS.

We also know that the assembly of ZBP1 filaments at mitochondria requires two sequential steps:

- 1) Recruitment of ZBP1 to mitochondria via a direct or indirect interaction with MAVS located at the MOM. Indeed, cells lacking MAVS failed to form ZBP1 filaments at mitochondria ((Referee 3 – Figure 1).
- 2) ZBP1 oligomerization through RHIM1-RHIM1 interactions is required for filament formation at the MOM. Mutant ZBP1 with disrupted RHIM1 domain failed to oligomerize into filaments (Figure 4a).

Our study provides evidence that ZBP1 filaments assembly at mitochondria is required to activate the MAVS-TBK1-IRF3 innate immune signaling pathway. MAVS can form helical fibrils that self-propagate like prions to promote signaling complex assembly, and these helical filaments are visualized by cryo-electron microscopy imaging⁴. We agree that it is interesting to investigate the mechanism underlying MAVS filament formation by ZBP1, but we feel nailing down the

precise dynamics of this process exceeds the immediate scope of the manuscript, which focuses on the novel telomere-to-mitochondria signaling axis.

Referee3 – Figure1: a, Representative confocal microscopy images of growing and crisis IMR90 E6E7 and WI38 SV40LT co-immunostained for endogenous ZBP1 and MAVS. The PDs are indicated. Scale bar 10 μ m. **b**, Box and whisker plots showing the number of ZBP1 filaments in pre-crisis (PD94) IMR90 E6E7 expressing either empty vector or WT ZBP1(S)-Flag. Cells were transfected with three individual siMAVS (#1, #2, and #3) or non-targeting control siRNA. siMAVS#1 is no efficient in depleting MAVS and therefore serves as a control. Center line: median; box limits: 1st and 3rd quartiles; whiskers: 10th and 90th percentiles. n: number of cells analyzed. One-way ANOVA, ns: not significant, * $p < 0.05$, ** $p < 0.01$, *** $p < 0.001$. Three independent experiments were performed. Abbreviations: ZBP1(S): short isoform; CTR: control; PD: population doubling.

3) What features of TERRA that make it capable of activating MAVS in a manner dependent on ZBP1 but not RIG-I or MDA5?

Response: RIG-I primarily recognizes short double-stranded RNA with 5' triphosphate groups, 5'ppp single-stranded RNA and circular RNAs, while MDA5 primarily recognizes long double-stranded RNA (>1 kb) independent of 5'-end phosphorylation status^{20–22}. On the other hand, ZBP1 binds preferentially to Z-DNA^{9,10} and Z-RNA^{11–13} molecules with an unusual left-handed double helix structure. TERRA is a long non-coding single-stranded RNA molecule made up of UUAGGG repeats. The G-rich TERRA can form G-quadruplex (G4) RNA structures, which contain stacked Hoogsteen-bonded G-quartet motifs^{14–16}. Since this higher order arrangement of G4 have been observed in long TERRA sequences and the backbone progression of G4 RNA can adopt left-handed configurations, we reasoned that these non-canonical structures within TERRA might serve as specific ligands for ZBP1.

To test this, cells expressing ZBP1(S) were transfected with either TERRA-WT oligonucleotide [sequence (UUAGGG)₄] capable of folding into a G4 structure or TERRA-MUT with two mutated Gs in each repeat [sequence (UUACCG)₄] unable to form the G4 structure^{17,18} (Referee 3 – Figure 2). TERRA-WT triggered a ZBP1(S)-dependent immune response marked by a significant IFN β and ISG induction. On the other hand, ZBP1(S) expressing cells were insensitive to the

presence of TERRA-MUT and did not activate an immune response (Referee 3 – Figure 1a-b). These results suggest that ZBP1 could indeed recognize G4 features of TERRA and that TERRA G4 RNA interaction with ZBP1 Z α 2 domain is required for immune activation. The rigorous validation of this idea requires extensive biophysical studies and detailed analysis of TERRA structure. While we are working on these approaches, we hope the reviewer agrees that solving the precise nucleic acid structure required for ZBP1 binding/activation is a complex problem and therefore outside the immediate scope of this manuscript.

4) Is there a way to deplete TERRA in cells and show that it is essential for ZBP1 activation?

Response: To demonstrate a causal role of TERRA in ZBP1-dependent immune response, we used two individual siRNAs allowing us to partially suppress TERRA in cells co-expressing WT ZBP1(S) and VP16-TRF1 (Extended Data Figure 12f-g). These two siRNAs were found previously to partially deplete TERRA RNA levels up to ~40% as measured by RNA dot blot and real-time PCR¹⁹. TERRA depletion was sufficient to delay and attenuate STAT1 activation when compared to control cells (transfected with non-targeting siRNA). These results further validate the immunostimulatory properties of TERRA-derived RNA species and their potential in inducing ZBP1-dependent immune response (Extended Data Figure 12e-g).

5) Is there direct biochemical evidence that TERRA causes ZBP1 filament formation?

Response: ZBP1(S)-Flag fRIP and immunofluorescence experiments indicate that (i) ZBP1 interacts with TERRA and that this interaction requires the functional nucleic acid binding domain Z α 2 (Figure 3) (ii) ZBP1 filament formation requires a functional Z α 2 domain (Figure 4). We are in the process of analyzing structural changes of ZBP1 in response to TERRA binding, but these are complicated experiments that will take years to complete. We therefore hope the referee agrees with us that this (similar to that above in point 4) is outside of the immediate scope of this manuscript.

6) ZBP1 knockout mice are available. Do these mice exhibit a phenotype consistent with defective replicative crisis? The mouse telomeres are different from those of humans, but efforts to address the role of ZBP1 in an in vivo model would help to support (or refute) the authors' model.

Response: Studies using mouse models suggested that ZBP1 senses endogenous Z-form nucleic acids and triggers RIPK3-dependent necroptosis and inflammation^{20,21}. Both ZBP1 and TERRA are quite differently regulated in mice versus humans. The function of ZBP1(S) has not been addressed in mouse studies, since mouse cells only express the full-length version of ZBP1 as well as another short form that contains the Z α 1 and Z α 2 domains, but lacks the C-terminal part containing the RHIMs^{21,22}. Furthermore, mouse telomeres do not undergo replication associated shortening and thus crisis does not serve as natural proliferative barrier in mice, and TERRA transcription is not driven through telomere dysfunction, but through the pseudoautosomal PAR locus²³. We therefore suggest that we have discovered a pathway that evolved specifically as tumor suppressor in humans, together with replicative telomere shortening, and therefore ZBP1 mice do not exhibit comparable phenotypes. Developing 'humanized' mouse models for both telomere metabolism and ZBP1 splicing to address this is years away and we hope the reviewer agrees well beyond the scope of the current study.

TERRA-WT: UUAGGGUUAGGGUUAGGGUUAGGG
 TERRA-MUT: UUA**CC**GUUA**CC**GUUA**CC**GUUA**CC**G

a

b

Referee3 – Figure2: a-b, Scatter plot with bars showing RT-qPCR analysis ISGs of growing IMR90 E6E7 (PD45) expressing WT ZBP1(S)-Flag transfected with either TERRA-WT oligonucleotide [sequence (UUAGGG)4] or TERRA-MUT with two mutated Gs in each repeat [sequence (UUACCG)4]. RNA extracts were collected at day 4 post-transfection. Expression levels were normalized to non-transfected cells (NT). Bars represent mean \pm s.d. from technical replicates. n: number of technical replicates. One-way ANOVA, ns: not significant, * $p < 0.05$, ** $p < 0.01$, *** $p < 0.001$. Abbreviations: PDs: population doublings. Abbreviations: ZBP1(S): short isoform; NT: non-transfected, WT: wild-type; MUT: mutant; PD: population doubling.

References:

1. Sarhan, J. et al. Constitutive interferon signaling maintains critical threshold of MLKL expression to license necroptosis. *Cell Death Differ.* 26, 332–347 (2019).
2. Karki, R. et al. ADAR1 restricts ZBP1-mediated immune response and PANoptosis to promote tumorigenesis. *Cell Rep.* 37, 109858 (2021).
3. Karki, R. et al. Synergism of TNF- α and IFN- γ Triggers Inflammatory Cell Death, Tissue Damage, and Mortality in SARS-CoV-2 Infection and Cytokine Shock Syndromes. *Cell* 184, 149–168.e17 (2021).
4. Hou, F. et al. MAVS forms functional prion-like aggregates to activate and propagate antiviral innate immune response. *Cell* 146, 448–461 (2011).
5. Hornung, V. et al. 5'-Triphosphate RNA Is the Ligand for RIG-I. *Science* 314, 994–997 (2006).
6. Kato, H. et al. Length-dependent recognition of double-stranded ribonucleic acids by retinoic acid-inducible gene-1 and melanoma differentiation-associated gene 5. *J. Exp. Med.* 205, 1601–1610 (2008).
7. Pichlmair, A. et al. RIG-I-Mediated Antiviral Responses to Single-Stranded RNA Bearing 5'-Phosphates. *Science* 314, 997–1001 (2006).
8. Schlee, M. et al. Recognition of 5' Triphosphate by RIG-I Helicase Requires Short Blunt Double-Stranded RNA as Contained in Panhandle of Negative-Strand Virus. *Immunity* 31, 25–34 (2009).
9. Ha, S. C. et al. The crystal structure of the second Z-DNA binding domain of human DAI (ZBP1) in complex with Z-DNA reveals an unusual binding mode to Z-DNA. *Proc. Natl. Acad. Sci. U. S. A.* 105, 20671–20676 (2008).
10. Schwartz, T., Behlke, J., Lowenhaupt, K., Heinemann, U. & Rich, A. Structure of the DLM-1-Z-DNA complex reveals a conserved family of Z-DNA-binding proteins. *Nat. Struct. Biol.* 8, 761–765 (2001).
11. Thapa, R. J. et al. DAI Senses Influenza A Virus Genomic RNA and Activates RIPK3-Dependent Cell Death. *Cell Host Microbe* 20, 674–681 (2016).
12. Maelfait, J. et al. Sensing of viral and endogenous RNA by ZBP1/DAI induces necroptosis. *EMBO J.* 36, 2529–2543 (2017).
13. Placido, D., Brown, B. A., Lowenhaupt, K., Rich, A. & Athanasiadis, A. A left-handed RNA double helix bound by the Z alpha domain of the RNA-editing enzyme ADAR1. *Struct. Lond. Engl.* 1993 15, 395–404 (2007).
14. Martadinata, H. & Phan, A. T. Structure of human telomeric RNA (TERRA): stacking of two G-quadruplex blocks in K(+) solution. *Biochemistry* 52, 2176–2183 (2013).
15. Collie, G. W. et al. Electrospray mass spectrometry of telomeric RNA (TERRA) reveals the formation of stable multimeric G-quadruplex structures. *J. Am. Chem. Soc.* 132, 9328–9334 (2010).
16. Agarwala, P., Pandey, S. & Maiti, S. The tale of RNA G-quadruplex. *Org. Biomol. Chem.* 13, 5570–5585 (2015).
17. Biffi, G., Tannahill, D. & Balasubramanian, S. An intramolecular G-quadruplex structure is required for binding of telomeric repeat-containing RNA to the telomeric protein TRF2. *J. Am. Chem. Soc.* 134, 11974–11976 (2012).
18. Ghisays, F. et al. RTEL1 influences the abundance and localization of TERRA RNA. *Nat. Commun.* 12, 3016 (2021).
19. Deng, Z., Norseen, J., Wiedmer, A., Riethman, H. & Lieberman, P. M. TERRA RNA Binding to TRF2 Facilitates Heterochromatin Formation and ORC Recruitment at Telomeres. *Mol. Cell* 35, 403–413 (2009).
20. Ingram, J. P. et al. ZBP1/DAI Drives RIPK3-Mediated Cell Death Induced by IFNs in the Absence of RIPK1. *J. Immunol.* 203, 1348–1355 (2019).

21. Devos, M. et al. Sensing of endogenous nucleic acids by ZBP1 induces keratinocyte necroptosis and skin inflammation. *J. Exp. Med.* 217, e20191913 (2020).
22. Devos, M. et al. Sensing of endogenous nucleic acids by ZBP1 induces keratinocyte necroptosis and skin inflammation. *J. Exp. Med.* 217, e20191913 (2020).
23. Viceconte, N. et al. PAR-TERRA is the main contributor to telomeric repeat-containing RNA transcripts in normal and cancer mouse cells. *RNA N. Y.* N 27, 106–121 (2021).

Referee #4 (Remarks to the Author):

Nassour et al., propose that ZBP1 acts as a sensor for telomere damage-dependent replicative crisis in human lung fibroblasts that have been immortalized with either papillomavirus or polyomavirus oncogenes that overcome cellular senescence. This crisis is known to depend on cGAS-STING (a cytosolic DNA sensor) but here dependence is extended to telomeric repeat-containing RNA (TERRA), where “TERRA-ZBP1 interactions are sufficient to launch a detrimental MAVS-dependent innate immune signaling response”. The work represents observational novelty, particularly because it places focus on the short form of ZBP1. As presented, the work is very much underdeveloped and does not demonstrate awareness of existing understanding of ZBP1-dependent death pathways or its recognized partners, particularly RIPK3 and RIPK1. By focusing on oligomerization of ZBP1 “into long filaments that interact and colocalize with MAVS on the outer mitochondrial membrane of a subset of mitochondria” without considering more direct consequences, the reader is left with the distinct impression that not all the important features have been investigated or disclosed. The potential activation of ZBP1 by cGAS-STING signal transduction (the latter already suspected of contributing to an autophagy-dependent cell death program) leaves unresolved the impact of ZBP1 signal transduction via RIPK3, in particular. Basically, the manuscript follows from possibility to possibility without disclosing concrete and compelling mechanisms that fit with what is already known. ZBP1 is best known for triggering RIPK3-MLKL-dependent necroptosis and RIPK3-RIPK1-FADD-CASP8-dependent apoptosis (as well as nondeath consequences of component scaffold interactions that leads to activation of cytokines and potentially pyroptosis).

Response: We thank the referee for recognizing the novelty of our study, and appreciate the concerns raised over our limited treatment of the known cell death connections to ZBP1 in the original submission. Since we had previously shown that cell death in crisis was autophagy-dependent (Nassour et al., Nature 565, 659–663 (2019))²¹ and our preliminary data indicated that RIPK3-dependent cell death was not involved, we did not emphasize this in the initial submission. We realize now this is important to demonstrate, so, as requested, we have now addressed this in the revised version. However, before we detail the new data in this regard, we would like to clarify the first statement above made by the reviewer, so that we are starting from a common understanding of the cell model systems used. The reviewer states that: “human lung fibroblasts that have been immortalized with either papillomavirus or polyomavirus oncogenes that overcome cellular senescence”. It is actually incorrect that E6E7 and SV40LT are capable of immortalizing cells, as these viral oncoproteins only allow cells to bypass senescence through the disruption of p53/Rb. Without active telomere maintenance pathways these cells will still enter replicative crisis, which is the subject of this study. Immortalized cell lines, by definition, exhibit an unlimited replicative potential and therefore do not experience replicative crisis. In short, the cells used in this study are not immortalized.

As the reviewer states, ZBP1 is capable of recruiting RIPK3 and activating parallel cell death pathways of apoptosis/necroptosis/pyroptosis during viral infections with herpesviruses (MCMV, HSV-1), the poxvirus VACV, and the orthomyxoviruses IAV and IBV^{1–5}. In these settings, ZBP1 clusters RIPK3, which initiates cell death signaling through FADD-caspase8 (for apoptosis), MLKL (for necroptosis), and NLRP3 (for pyroptosis) downstream of RIPK3. These studies, however, were performed primarily on mouse fibroblasts, airway epithelial cells, and macrophages expressing both ZBP1(full-length) and RIPK3. Moreover, the potential contribution of ZBP1 in cell death in human cells is limited to the use of the necroptosis-sensitive human HT-29 cells, a cell line that highly expresses RIPK3. Therefore, we addressed this directly in the human cells used in this study. We performed a series of experiments to characterize the type of ZBP1(S)-dependent cell death in human fibroblasts and found no significant activation of the pyroptotic,

*apoptotic and necroptotic pathways upon expression of ZBP1(S) (Extended Data Figure 6a-c) or in crisis cells (Extended Data Figure 6e-g). Indeed, markers of apoptosis (cleaved caspase 3, cleaved caspase 7, and cleaved PARP1), markers of necroptosis (phosphorylation of RIPK3 and its downstream effector, MLKL), and markers of pyroptosis (cleaved caspase 1 and its downstream substrates gasdermin D (GSDMD) and Interleukin-1 β (IL-1 β)) were not detected during replicative crisis or upon expression of ZBP1(S) in growing cells. Notably, RIPK3 expression levels were low/undetectable when compared to the necroptosis-sensitive cell line HT-29 (Extended Data Figure 6b). On the other hand, LC3 lipidation and p62 degradation were observed (Extended Data Figure 6d,h), consistent with an autophagy-dependent cell death pathway (Nassour et al., Nature **565**, 659–663 (2019))²¹.*

Furthermore, we used siRNAs to deplete the essential autophagy related protein ATG7, as well as type I interferon (IFN) receptors (IFNAR1 and IFNAR2), ZBP1, and RIPK3 and measured the frequency of cell death during crisis using the Incucyte Live-Cell Analysis system. The frequency of cell death was significantly reduced upon suppression of ZBP1, IFNAR, and ATG7. On the other hand, depleting RIPK3 did not prevent or attenuate cell death during crisis (Extended Data Figure 6i-j). Thus, our data support a model where, in primary human cells, ZBP1(S) does not activate intrinsic PANoptosis through RIPK3, but rather drives a cell death signaling pathway that is dependent of IFN and autophagy.

Of course, MAVS is a component of the RIG-I-like signaling pathway. Thus, the reader faces data supporting several known innate immune signaling pathways all apparently collaborating in the execution of cells entering crisis. The work overlooks the question of whether this pattern represents known collaboration of NF-kappaB and IRF3, but show that IFN itself does contribute; however, this manuscript fails to unambiguously address its stated goal of demonstrating how “dysfunctional telomeres engage nucleic acid (NA) sensing machineries and activate immune signaling pathways required for crisis-associated cell death”.

*Response: While the role of NF- κ B and IRF3 per se were not a focus of this study, we do feel like we have made significant connections between various innate immune sensors in a novel pathway of events that culminates in cell death during crisis. First, we refer the referee to Nassour et al., Nature **565**, 659–663 (2019)²¹, where we demonstrate at length that cytoplasmic DNA species resulting from telomere dysfunction activate cGAS/STING, both of which are required for cell death in crisis. This is the initiating event in the new cascade we now document here. That is, cGAS-Sting upregulates ZBP1(S) and it then binds the TERRA RNA signal (a second signal from dysfunctional telomeres), allowing it to oligomerize, localize to mitochondria, and activate a second MAVS-dependent wave of IFN signaling. We agreed with the reviewer that more data on the MAVS dependence was needed to solidify the proposed pathway. Thus, in the revised manuscript, we include two additional experiments demonstrating the dependency of MAVS, but not RIG-I and MDA5, in the ZBP1-driven IFN pathway. 1) Suppression of MAVS, but not RIG-I and MDA5, resulted in a significant reduction of ZBP1 mRNA levels during crisis (Extended Data Figure 4e). 2) Suppression of MAVS, but not RIG-I and MDA5, prevented ZBP1-driven IFN response (Extended Data Figure 15).*

Since being studied for senescence, human WI38 fibroblasts (and IMR90 fibroblasts) have been work horses of this field. When immortalized by HPV E6/E7 or SV40 LT antigen (or when immortalized by overexpression of telomerase itself), these cells proliferate beyond the so-called “Hayflick limit” but nevertheless reach a point of replicative crisis that impacts most cells in a culture. This crisis is influenced by cGAS-STING signaling, has characteristics of a type I interferon (IFN) response and results in autophagy-type cell death. Here, IMR90 fibroblasts

immortalized with E6/E7 and WI38 cells immortalized with SV40 LT are found to show a increased IFN signature pattern as they reach a passage level just prior to crisis, suggesting that an inflammatory profile driven by the events preceding crisis contribute to crisis. This correlation drives the experimental design, but remains confounding throughout.

Response: As stated above, the referee is working on an incorrect premise here. The cells we are working with are not immortalized. Expression of E6/E7 or SV40LT does not immortalize cells, nor do telomerase-immortalized cells undergo crisis (since they have unlimited proliferation potential). We also argue that the inflammatory profile we demonstrate is far from just correlative, since we can switch it on and off at will by ZBP1(S) and TERRA expression as described in the following experiments:

1-siRNA-based suppression of ZBP1 abolished crisis-associated ISG signature (Figure 1)

2- CRISPR-based deletion of ZBP1 prevented IFN response and allowed crisis survival (Extended Figure 3).

3-ZBP1 overexpression in pre-crisis cells enhanced IFN response (Figure 2).

4- Telomere dysfunction activated a ZBP1-dependent IFN response (Figure 2).

5- TERRA induction activated a ZBP1-dependent IFN response and cell death (Figure 3).

ZBP1 has shown the ability to rescue defects in some of the most unexpected settings in mice as well as in human cells and organoids, potentially informing the question at hand with replicative senescence. Starting with observations on the combined role of TNF together with type I IFN in the death of RIPK1-deficient cells and mice, shown to be rescued by the combined elimination of RIPK3 and caspase-8 (Kaiser et al., PNAS, 2014), continuing with observations in RelA-deficient cells and mice (Thapa et al., Mol Cell Biol, 2011, and Xu et al., J Immunol, 2018), and the clear demonstration that ZBP1 is an important sensor that drives the activation of RIPK3 in the absence of RIPK1 or RIPK1 RHIM function (Newton et al., Nature, 2016; Lin et al., Nature 2016; Ingram et al., J Immunol 2019) the likely contribution of interferons in stressful settings seems predictable, particularly in combination with other inflammatory cytokines such as TNF. Observations first by Mandal et al. (Immunity, 2018) on endotoxic shock (a model of sepsis), but followed by Yang et al. (Cell Mol Immunol, 2020) and Karki et al. (Cell, 2021) with systemic inflammatory response syndrome (SIRS) revealed the combined impact of TNF and either type I or type II IFN signaling in driving lethal insult. Other allied work, such as Frank et al. (Cell Death Diff, 2019) showed that cell cycle disrupting cancer drugs known to induce genotoxic stress in cancer cells act in concert with type I IFN, triggering ZBP1-RIPK3-MLKL necroptosis. Also, genome instability-dependent death due to disruption of Setdb1 in mice or human organoids have been rescued by elimination of RIPK3, MLKL or ZBP1 (as well as by pharmaceutical inhibition of RIPK3 kinase activity) as shown Wang et al. (Nature 2020). The contribution of type I IFN seemed key to activation of ZBP1 such that authors commented, “the induction of cell death in organoids requires not only the upregulation of ZBP1 but also an unknown factor or factors to initiate ZBP1-mediated necroptosis”. In that work they go on to show derepressed endogenous RNA from “large endogenous retroviruses (ERVs)” in both the mouse and human settings as “the other factor”, observations relevant here. Recently published observations of Z-RNA as the ZBP1 ligand in both the influenza (Zhang et al., Cell 2021 – Ref 13) and vaccinia virus (Koehler et al Cell Host Microbe 2021) infection examples raises further concerns as to whether the experimental design here has overlooked some obvious contributors and outcomes from ZBP1-dependent pathways that play out independent of MAVS or other mitochondrial cell death signaling components.

Response: We thank the referee for the comprehensive summary of the ZBP1 literature. As mentioned above, we have now performed a series of experiments to characterize the type of ZBP1(S)-dependent cell death in human fibroblasts and found no significant activation of the pyroptotic, apoptotic and necroptotic pathways upon expression of ZBP1(S) (Extended Data

Figure 6a-d) or in crisis cells (Extended Data Figure 6e-h). Indeed, markers of apoptosis (cleaved caspase 3, cleaved caspase 7, and cleaved PARP1), markers of necroptosis (phosphorylation of RIPK3 and its downstream effector, MLKL), and markers of pyroptosis (cleaved caspase 1 and its downstream substrates gasdermin D (GSDMD) and Interleukin-1 β (IL-1 β)) were not detected during replicative crisis or upon expression of ZBP1(S) in growing cells. On the other hand, LC3 lipidation and p62 degradation were observed, consistent with an autophagy-dependent cell death pathway as we showed previously (Extended Data Figure 6a-h). Furthermore, we used siRNAs to deplete the essential autophagy related protein ATG7, as well as type I interferon (IFN) receptors (IFNAR1 and IFNAR2), ZBP1, and RIPK3 and measured the frequency of cell death during crisis using the Incucyte Live-Cell Analysis system. The frequency of cell death was significantly reduced upon suppression of ZBP1, IFNAR, and ATG7. On the other hand, depleting RIPK3 did not prevent or attenuate cell death during crisis (Extended Data Figure 6i-j). Thus, our data support a model where, in primary human cells, ZBP1(S) does not activate intrinsic PANoptosis through RIPK3, but rather drives an intrinsic cell death signaling pathway that is dependent of IFN and autophagy.

Although data presented in Figure 1 and Extended Data 1 provides evidence consistent with the authors hypothesis, the data are mostly correlative. The connection to actual crisis (where a majority of cell die) needs more complete single cell documentation.

Response: The results shown in Figure 1, Extended Data 1 and later in the manuscript show a direct role of ZBP1 in cell death and immune response during crisis and not a simple correlation. 1- Genome-wide CRISPR screening showed that ZBP1 is required for cell death during crisis (Figure 1).

2-Live-cell imaging experiments using ZBP1 KO clones reconstituted with ZBP1(S) showed increased frequencies of cell death and induction of ISGs (Figure 1).

3-siRNA-mediated depletion of ZBP1, but not RIPK3, reduced the frequency of cell death during crisis as measured by live-cell imaging experiments (Extended Data Figure 6i-j).

3-We can switch it on and off at will by ZBP1 and TERRA expression (Figure 3), and see response to the point above.

From the studies that have made use of human fibroblasts to evaluate ZBP1 contribution to triggering cell death over the past decade or so, two features stand out: (i) as a Z-nucleic acid sensor, ZBP1 is both induced by type I IFN and induces type I IFN and (ii) ZBP1 levels, together with levels of its partner RIPK3, vary dramatically with passage and from cell strain to cell strain in culture, as well as upon immortalization. To repeat a feature already mentioned, once activated, ZBP1 recruitment of RIPK3 has been implicated most consistently in the death of cells from all insults evaluated. Here, ZBP1 is purported to play a sensor role leading to crisis, but the absence of ZBP1 is shown to have a dramatic impact on the basal level of IFN signaling in these cells and that may be as important in the observations as its contribution as a sensor.

Response: We searched the literature, and we were unable to find studies mentioned by the reviewer supporting the statement: "from the studies that have made use of human fibroblasts to evaluate ZBP1 contribution to triggering cell death over the past decade...". No studies that we are aware of have been done on ZBP1-driven cell death pathways in human fibroblasts. The role of ZBP1 in mediating cell death (i.e. PANoptosis) comes from mostly studies on murine cell lines or the necroptosis-sensitive human HT-29 cells, which all highly express RIPK3. Thus, we feel our study in non-immortalized human cells breaks new ground in this area, and certainly is unique regarding the role of the ZBP1(S) isoform and discovering for the first time a role for TERRA RNA as second messenger of telomere stress.

Authors find ZBP1-KO cells enjoy a modest number of additional passages without performing the obvious flip side experiment. The reader is not convinced that ZBP1 represents a crucial player with only this level of observation, given “reduced expression and secretion of IFN- β ”, a lower tonic level of IFN signaling seems likely in the absence of ZBP1, meaning any of multiple ISGs could be limiting.

Response: We did indeed do the ‘flip side’ experiment in the original submission and these data remain in the revised version:

1-Expression of ZBP1(S) in precrisis cells stimulated IFN response and cell death (Figure 1).

2-Expression of ZBP1(S) together with TERRA in young cells induces crisis and ISG signaling (Figure 3, Extended Figure 12).

Furthermore, it is important to emphasize that ZBP1 KO cells bypassed crisis and continue to divide for an additional 7-10 population doublings (and not passages). A growth difference in 7-10 PD difference correspond to an increase in cell number by 100-1000 fold. Live-cell imaging showed a significant reduction in the percentage of dead cells (while 60% of control cells (sgLUC, sgGFP) were dead at 24 hours post-imaging, only 5-10% of sgZBP1 cells were dying). We hope the reviewer can appreciate that this is not a ‘modest’ effect.

If the authors’ viewpoint is correct, conditional expression of ZBP1 expression should be sufficient to trigger crisis at an earlier passage (an experiment that will go much further to demonstrate its critical role as a driver of crisis). If this reader’s interpretation is correct, that overexpression of ZBP1 may provide a basal level of protection. The cells that are entering crisis must be subjected to single cell evaluation to establish whether elevated ZBP1-GFP levels precede death in time-dependent assays within a given cell. The authors contention that “deletion of ZBP1 demonstrated its essential role in crisis” is premature. Indirect impacts on IFN induction as well as IFN signaling are likely confounders here that make it appear ZBP1 contributes when the underlying tonic inflammatory state may be the culprit. Specificity of any of the signaling pathways being implicated here remain all important.

Response: The Referee is correct, expression of ZBP1 (together with its substrate TERRA) should be sufficient to trigger crisis in young cells with functional telomeres (which do not express TERRA endogenously). As explained in the point above, an experiment showing that expression of ZBP1(S) together with TERRA in young cells induces crisis and ISG signaling (Figure 3, Extended Figure 12) was in the original submission and remains in the revised version.

(In 133) As the authors proceed to evaluate type I IFN activation with attention first towards TBK1 and later towards MAVS, contributions of ZBP1 and cGAS-STING must be better distinguished in an unambiguous manner as these data are presented. The number of sensors that can contribute to innate immune activation of type I IFN includes both as well as many more that are not included in this study. In particular, genotoxic stress has long been a topic of study as mentioned above. The conclusion that “cGAS/STING pathway is responsible for an initial transcriptional induction of ZBP1” requires more complete demonstration. Current data does not allow the reader to comprehend the contributions of these respective pathways or the independence of the signaling outcomes.

Response: This point is similar to another raised above about the relationship between the many innate immune receptors in the observed crisis IFN response and thus we are reiterating our response here. We argue that we have indeed made significant connections between various innate immune sensors in a novel pathway of events that culminates in cell death during crisis.

First, we refer the referee to Nassour et al., *Nature* **565**, 659–663 (2019)²¹, where we demonstrate convincingly that cytoplasmic DNA species resulting from telomere dysfunction activate cGAS/STING, both of which are required for cell death in crisis. This is the initiating event in the new cascade we now document here. That is, cGAS-STING upregulates ZBP1(S), which then binds the TERRA RNA signal (a second signal from dysfunctional telomeres), allowing it to oligomerize and localize to outer mitochondrial membrane, where it activates a second MAVS-dependent wave of IFN signaling. We agreed with the reviewer that more data on the MAVS dependence was needed to solidify the proposed pathway. Thus, in the revised manuscript, we include two additional experiments demonstrating the dependency of MAVS, but not RIG-I and MDA5, in the ZBP1-driven IFN pathway:

- 1) Suppression of MAVS, but not RIG-I and MDA5, resulted in a significant reduction of ZBP1 mRNA levels during crisis (Extended Data Figure 4e).
- 2) Suppression of MAVS, but not RIG-I and MDA5, prevented ZBP1-driven IFN response (Extended Data Figure 15). Finally, Extended Data Figure 4e demonstrates the loss of ZBP1 induction in crisis upon suppression of cGAS or STING, placing ZBP1 induction downstream of cGAS activation as we propose.

(In 142) Authors report that “individual suppression of cGAS, STING, or IFN receptors (IFNAR1 and IFNAR2) resulted in a significant reduction of ZBP1” but fail to evaluate the expected contributions of RHIM signal transduction via RIPK1 or RIPK3.

Response: As the referee requested, we have now performed additional experiments to evaluate the role of RIPK3 in ZBP1-mediated cell death (PANoptosis) during crisis.

1- As mentioned previously, we have now performed a series of experiments to characterize the type of ZBP1(S)-dependent cell death in human fibroblasts and found no significant activation of the pyroptotic, apoptotic and necroptotic pathways upon expression of ZBP1(S) (Extended Data Figure 6a-c) or in crisis cells (Extended Data Figure 6e-g). Indeed, markers of apoptosis (cleaved caspase 3, cleaved caspase 7, and cleaved PARP1), markers of necroptosis (phosphorylation of RIPK3 and its downstream effector, MLKL), and markers of pyroptosis (cleaved caspase 1 and its downstream substrates gasdermin D (GSDMD) and Interleukin-1 β (IL-1 β)) were not detected during replicative crisis or upon expression of ZBP1(S) in growing cells. Notably, RIPK3 expression levels were low/undetectable when compared to the necroptosis-sensitive cell line HT-29 (Extended Data Figure 6b). On the other hand, LC3 lipidation and p62 degradation were observed (Extended Data Figure 6d,h), consistent with an autophagy-dependent cell death pathway (Nassour et al., *Nature* **565**, 659–663 (2019)²¹. Furthermore, we used siRNAs to deplete the essential autophagy related protein ATG7, as well as type I interferon (IFN) receptors (IFNAR1 and IFNAR2), ZBP1, and RIPK3 and measured the frequency of cell death during crisis using the Incucyte Live-Cell Analysis system. The frequency of cell death was significantly reduced upon suppression of ZBP1, IFNAR, and ATG7. On the other hand, depleting RIPK3 did not prevent or attenuate cell death during crisis (Extended Data Figure 6i-j). Thus, our data support a model where, in primary human cells, ZBP1(S) does not activate intrinsic PANoptosis through RIPK3, but rather drives a cell death signaling pathway that is dependent of IFN and autophagy.

2-Crisis IMR90 E6E7 (PD105) were transfected with siRNAs against ZBP1, IFNAR1, IFNAR2, ATG7, and RIPK3. The frequency of cell death was measured by the incorporation of the Cytotox dye using the IncuCyte live-cell imaging platform. While suppression of ZBP1 led to a significant reduction in the percentage of dead cells, depletion of RIPK3 had no detectable effect, indicating that cell death in crisis is dependent on ZBP1, interferon-signaling and autophagy, but independent on RIPK3 (Extended Data Figure 6i).

(In 157) The important ZBP1 Zalpha2 (called Zbeta here) domain here seems reminiscent of the sensing of RNA in MCMV, HSV, VACV and influenza infections, where the activating ligand is now known to be Z-RNA in both influenza and VACV (Zhang et al and Koehler et al.). Also, the most proximal RHIM has been implicated in these systems (starting with Upton et al., Cell Host Microbe 2012, where an MCMV-encoded suppressor of RHIM signaling was reported). In every system that has been studied since, the role of this RHIM has recruitment of RIPK3 and/or RIPK1. Here the ongoing IFN response together with autophagy contribute, but in what order and with what signaling consequences? Authors should make a more sincere effort to describe how hTERT immortalized fibroblasts proceed to crisis, either here or elsewhere because this is a common means of human fibroblast immortalization. Authors must investigate whether a necroptotic (RIPK3-MLKL) outcome or multiple death (RIPK3-MLKL and RIPK3-RIPK1-FADD-CASP8) pathways play out in the presence of viral oncogenes or hTERT to make this complete. Overlooking this enormous body of research while implicating ZBP1 as a sensor of impending crisis has likely left key early steps leading to crisis completely undisclosed. Something similar may be playing out in crisis if, and only if ZBP1 is actually acting as a sensor and not as a regulator of basal interferon levels. Finally, for this data series to be compelling, levels of all mutant forms of ZBP1 must be evaluated and shown on western blots.

Response: First, we must state again that hTERT immortalized fibroblasts have an unlimited growth potential and do not proceed to crisis. Thus, it is not possible to experimentally address the above criticism. Second, while one can argue there is always more characterization that can be done, we feel we have provided a wealth of data, including significant new data requested by this reviewer and the others. Our data has allowed us to make new major headway describing a novel pathway that requires a crisis-associated isoform of ZBP1 that together with its ligand TERRA executes a new form of cell death that prevents cell from progressing to malignancy.

The authors' focus on the short form of ZBP1 is based on its abundance in IFN-activated cells, which leaves much to be desired. Little has been done with this form in the past and so one reference is not sufficient to bring this forward as a candidate. Comparison of the two forms must be pursued. Although the short isoform is predicted to lack Zalpha1, nothing concrete has been reported. Thus, authors must be more complete in their evaluation of this aspect of ZBP1 here or leave it for a subsequent follow-up study.

Response: Little has been done with ZBP1(S) as almost all previous studies were done using mouse cells and models which do not express the ZBP1(S) isoform. ZBP1(S) has been reported recently in human myeloma cells⁶ and we have added this reference to the manuscript.

We have addressed the referees' questions below. However, we would like to point out that current study focuses on ZBP1(S), the isoform that is most associated with replicative crisis. ZBP1(L) levels are often almost impossible to detect during crisis. We therefore prefer not to include the data with ZBP1(L) in the main manuscript but rather only in the response here.

To address the referee's questions, ZBP1 KO IMR90 E6E7 were reconstituted with either ZBP1(L) or ZBP1(S) and tested for their susceptibility to induce ISGs and undergo cell death (Referee 4 – Figure 1). As expected, ZBP1(S) showed increased expression of ISGs concomitant with elevated cell death. Those reconstituted with ZBP1(L) also showed an increase of ISG levels and cell death frequency, though to a much lesser extent than ZBP1(S) (Referee 4 – Figure 1).

In 174) The fact that “depletion of ZBP1 dampened the TBK1-IRF3 signaling axis, reduced ISG expression, and attenuated autophagy” may again be an indirect consequence of

the impact ZBP1 has on basal/tonic levels rather than a direct role in the final steps of crisis. The observation that hTERT “resulted in telomere maintenance and continued growth past the crisis barrier” although these cells “did not express ZBP1, nor they did they activate IFN signaling” comes again to the chicken and egg question raised repeatedly above. For this section to be more compelling, conditional expression should be employed to bring on ZBP1 at various times in order to determine any direct contribution at different passage level as cells approach crisis. Control should include the Zalpha2 mutant reconstituted cells.

Response: We again mention that a key experiment in this regard was performed in the original submission and remains in the revised version. That is, ZBP1(S) together with conditional expression of TERRA in young cells induces a crisis-like phenotype marked by widespread cell death and robust ISG signaling (Figure 3, Extended Data Figure 12). We feel this is a compelling experiment to the referees point and feel that crisis experiments involving cell reconstituted with various mutant is a great future direction for the project, but not critical for the main conclusions reached in the current study.

Referee4 – Figure1: **a**, Immunoblotting of pre-crisis (PD97) ZBP1 KO MR90 E6E7 (clone 67) reconstituted with either WT ZBP1(L)-Flag or WT ZBP1(S)-Flag. Protein extracts were collected 14 days after transduction. GAPDH loading control. **b**, Left: Representative IncuCyte images of IMR90 E6E7 as in **a** after 24 hours of incubation. Dead cells are labeled with Cytotox green and nuclei with NucLight red. Right: Scatter plots with bars showing IncuCyte analysis of cell death of pre-crisis IMR90 E6E7 as in **a** after 24 hours of incubation. Empty vector expressing cells are used as controls. Bars represent mean \pm s.e.m. from technical replicates. One-way ANOVA, ns: not significant, * p <0.05, ** p <0.01, *** p <0.001. **c**, Scatter plots showing RT-qPCR analysis of ISGs in IMR90 E6E7 as in **a**. RNA extracts were collected 14 days after transduction. Expression levels were normalized to control cells with an empty vector. Bars represent mean \pm s.d. from technical replicates. n : number of technical replicates. One-way ANOVA, ns: not significant, * p <0.05, ** p <0.01, *** p <0.001. Abbreviations: ZBP1(S): short isoform; ZBP1(L): long isoform; KO: knockout, PD: population doubling.

(In 199) This section of the manuscript dealing with TERRA as a ligand for ZBP1 represents potentially the most interesting topic in the manuscript, particularly with the previous implications as a ligand for RNA sensors. This section must also be complete show compelling evidence that TERRA interaction with ZBP1 triggers activation through additional complementary methods. Such a strategy has already been accomplished in other settings recently published (Zhang et al., and Koehler et al.,) that more completely dissect the direct role of Z-RNA interactions with ZBP1. The experiments in the submitted manuscript seem to show that the RIG-I-like receptors are not involved with MAVS activation, which runs contrary to a large body of work, so must also be presented in a more compelling manner. The work must distinguish from nonspecific impacts such as those described by the Lieberman lab with their exosome work. CD data showing this RNA is Z-form will be needed above and beyond dot blots. Authors must more completely address interactions natural or synthetic TERRA binds recombinant Zalpha beyond the immunoprecipitations shown in the submitted manuscript for this all to reach an objective conclusion.

Response: The Lieberman group showed that TERRA can be found in extracellular fractions in mouse tumor and embryonic brain tissue, as well as in human tissue culture cell lines. This cell-free form of TERRA (cfTERRA) is enriched in extracellular exosomes and can modulate the inflammatory response in the tissue microenvironment. Our study, however, focuses on the recognition of TERRA by ZBP1(S), a cytosolic innate immune sensor. In Figure 3, we show that sensing of TERRA by WT ZBP1(S) (harboring a functional Za2 binding domain) induces a crisis-like phenotype associated with cell death and IFN-response. On the other hand, cells with a mutant form of ZBP1 (with disrupted Za2) did not respond to TERRA induction and did not exhibit an immune response.

TERRA is a long non-coding single-stranded RNA molecule made up of UUAGGG repeats. The G-rich TERRA can form G-quadruplex (G4) RNA structures, which contain stacked Hoogsteen-bonded G-quartet motifs⁸⁻¹⁰. Since this higher order arrangement of G4 have been observed in long TERRA sequences and the backbone progression of G4 RNA can adopt left-handed configurations, we reasoned that these non-canonical structures within TERRA might serve as specific ligands for ZBP1. To test this, cells expressing ZBP1(S) were transfected with either TERRA-WT oligonucleotide [sequence (UUAGGG)₄] capable of folding into a G4 structure or TERRA-MUT with two mutated Gs in each repeat [sequence (UUACCG)₄] unable to form the G4 structure^{11,12} (Referee 4 – Figure 2). TERRA-WT triggered an ZBP1(S)-dependent immune response marked by a significant IFN β and ISG induction. On the other hand, ZBP1(S) expressing cells were insensitive to the presence of TERRA-MUT and did not activate an immune response (Referee 4 – Figure 2). These results suggest that ZBP1 could indeed recognize G4 features of TERRA and that TERRA G4 RNA interaction with ZBP1 Za2 domain is required for immune activation. The validation of this idea requires extensive biophysical studies and detailed analysis of TERRA structure. While we are working on these approaches, we feel they are outside the immediate scope of this manuscript.

(In 270) The authors' suggestion that TERRA-activated ZBP1 SHORT drives "homotypic RHIM1 domain interactions" forming "aggregated structures rather than long filaments" that "then translocate to mitochondria to interact and activate MAVS" must include more compelling data. To begin, this process must be shown to be independent of other RHIM-containing adaptors (RIPK1, RIPK3 and TRIF), and to precede their activation when they are present. The physical nature of the oligomer must be dissected through biophysical assessment. The manuscript requires considerable cell free work with ZBP1, ZBP1 mutants and TERRA RNA to

get anywhere close to the conclusions currently being posited in this manuscript. At this point it is unclear if the oligomer is a consequence rather than a cause of crisis. The reader is left wondering “Is ZBP1 promote MAVS oligomerization” or “Is MAVS promoting ZBP1 fibril formation? (Or is this all a consequence of ZBP1 recruitment of RIPK3?).

Response: We agree with the referee on the importance of the ZBP1-MAVS relation. Focusing on the mechanism of the interaction for inflammation, we could demonstrate the dependency of the ZBP1 stimulated immune signaling on MAVS (now in Figure 4e-g and Extended Data Figure 16). To achieve this, ZBP1(S) was fused to the outer mitochondrial membrane targeting sequences of FIS1 in growing cells with functional telomeres. Under these circumstances, ZBP1(S) colocalized perfectly with MAVS (also on the mitochondrial outer membrane) and triggered an innate immune response marked by elevated ISG transcript levels. Suppression of MAVS prevented ISG transcription in response to forced ZBP1 mitochondrial localization, indicating that ZBP1-driven immune response necessitates MAVS.

Our study provides evidence that ZBP1 assembly into filaments at mitochondria is required to activate the immune pathway MAVS-TBK1-IRF3. MAVS can form helical fibrils that self-propagate like prions to promote signaling complex assembly, and these helical filaments are visualized by cryo-electron microscopy imaging¹³. We agree that it is interesting to investigate the mechanism underlying MAVS filament formation by ZBP1, but this and its precise relationship to TERRA binding requires extensive biochemical and biophysical studies that we feel are well beyond the scope of the current manuscript, which focuses on the telomere-mitochondria axis and already brings to light a novel nexus of cell biology, signaling and inflammation during crisis, an important anti-cancer barrier.

Finally, it is completely unclear whether the observations that are presented here for immortal WI38 and IMR90 are broadly relevant and generalizable. Many questions come to mind but are left unanswered.... Do normal (pre-senescent) fibroblasts have any of these characteristics as they reach senescence? Do fully transformed fibroblasts completely avoid this phenomenon? Are there natural settings where crisis occurs in a cancer cell progression that relate to these stages with diploid fibroblasts? This reader’s suspicion remains that this series of observations describes a particular correlative phenomenon in the two cell lines chosen. Nothing more.

Response: Our manuscript is focusing on replicative crisis, not on senescence. Crisis and senescence are two entirely different cellular responses, triggered by different stimuli, with very different outcomes. The inflammatory SASP response in senescence has been extensively characterized and does not lead to cell death. More importantly, we point the referee at the many manuscripts demonstrating the importance of crisis in vitro and in vivo¹⁴⁻²⁰, and at our extensive prior characterization of the crisis-autophagy phenotype in fibroblasts and epithelial cells of different origin²¹. Thus, we are confident in the sound basis for the well-established cell model systems used in this study and of the broader implications of the results we have obtained.

Most importantly, and also related to impact, autophagy remains a highly controversial mode of death regardless of the number of reports that implicate this pathway as a manner in which cells die. Given ZBP1 kills by far more established and direct programmed modes, the data must explore those conventional apoptotic and necroptotic mechanisms driven off of the same RHIM of ZBP1 as implicated here. Timing of events are likely key. Given that crisis occurs several passages beyond the point where experiments can be carried out, time course evaluation of single cells appears to be key to objective experimentation along with the need to recognize that ZBP1 is already known to activate RIPK3, MLKL, RIPK1, FADD and CASP8 signaling.

Response: There are now many recognized forms of cell death that operate in many different physiological contexts. While the reviewer's opinion is that autophagy as form a cell death is somewhat controversial, we see it as contextual. In fact, we again refer the reviewer to our previous paper that sets up the current study, which showed clearly that cell death during replicative crisis is dependent on autophagy without any involvement of apoptosis or any other caspase-mediated cell death pathways²¹. In addition, we performed a series of experiments to characterize the type of ZBP1(S)-dependent cell death in human fibroblasts and found no significant activation of the pyroptotic, apoptotic and necroptotic pathways upon expression of ZBP1(S) or in crisis cells. Our data support a model where in human cells, ZBP1(S) does not activate intrinsic PANoptosis through RIPK3, but rather drives a cell death signaling pathway that is dependent of IFN and autophagy (please see reply to reviewer's comment on Ln142).

TERRA-WT: UUAGGGUUAGGGUUAGGGUUAGGG
 TERRA-MUT: UUA**CC**GUUA**CC**GUUA**CC**GUUA**CC**G

a

b

Referee4 – Figure2: a-b, Scatter plot with bars showing RT-qPCR analysis ISGs of growing IMR90 E6E7 (PD45) expressing WT ZBP1(S)-Flag transfected with either TERRA-WT oligonucleotide [sequence (UUAGGG)4] or TERRA-MUT with two mutated Gs in each repeat [sequence (UUACCG)4]. RNA extracts were collected at day 4 post-transfection. Expression levels were normalized to non-transfected cells (NT). Bars represent mean \pm s.d. from technical replicates. Biological replicates are provided in Source data. n: number of technical replicates. One-way ANOVA, ns: not significant, * $p < 0.05$, ** $p < 0.01$, *** $p < 0.001$. Two independent experiments were performed. Abbreviations: PDs: population doublings. Abbreviations: ZBP1(S): short isoform; NT: non-transfected, WT: wild type; MUT: mutant; PD: population doubling.

References:

1. Kuriakose, T. et al. ZBP1/DAI is an innate sensor of influenza virus triggering the NLRP3 inflammasome and programmed cell death pathways. *Sci. Immunol.* **1**, aag2045 (2016).
2. Upton, J. W., Kaiser, W. J. & Mocarski, E. S. DAI/ZBP1/DLM-1 Complexes with RIP3 to Mediate Virus-Induced Programmed Necrosis that Is Targeted by Murine Cytomegalovirus vIRA. *Cell Host Microbe* **11**, 290–297 (2012).
3. Thapa, R. J. et al. DAI Senses Influenza A Virus Genomic RNA and Activates RIPK3-Dependent Cell Death. *Cell Host Microbe* **20**, 674–681 (2016).
4. Guo, H. et al. Species-independent contribution of ZBP1/DAI/DLM-1-triggered necroptosis in host defense against HSV1. *Cell Death Dis.* **9**, 1–11 (2018).
5. Pham, T. H., Kwon, K. M., Kim, Y.-E., Kim, K. K. & Ahn, J.-H. DNA Sensing-Independent Inhibition of Herpes Simplex Virus 1 Replication by DAI/ZBP1. *J. Virol.* **87**, 3076–3086 (2013).
6. Ponnusamy, K. et al. The innate sensor ZBP1-IRF3 axis regulates cell proliferation in multiple myeloma. *Haematologica* **107**, 721–732 (2021).
7. Deigendesch, N., Koch-Nolte, F. & Rothenburg, S. ZBP1 subcellular localization and association with stress granules is controlled by its Z-DNA binding domains. *Nucleic Acids Res.* **34**, 5007–5020 (2006).
8. Martadinata, H. & Phan, A. T. Structure of human telomeric RNA (TERRA): stacking of two G-quadruplex blocks in K(+) solution. *Biochemistry* **52**, 2176–2183 (2013).
9. Collie, G. W. et al. Electrospray mass spectrometry of telomeric RNA (TERRA) reveals the formation of stable multimeric G-quadruplex structures. *J. Am. Chem. Soc.* **132**, 9328–9334 (2010).
10. Agarwala, P., Pandey, S. & Maiti, S. The tale of RNA G-quadruplex. *Org. Biomol. Chem.* **13**, 5570–5585 (2015).
11. Biffi, G., Tannahill, D. & Balasubramanian, S. An intramolecular G-quadruplex structure is required for binding of telomeric repeat-containing RNA to the telomeric protein TRF2. *J. Am. Chem. Soc.* **134**, 11974–11976 (2012).
12. Ghisays, F. et al. RTEL1 influences the abundance and localization of TERRA RNA. *Nat. Commun.* **12**, 3016 (2021).
13. Hou, F. et al. MAVS forms functional prion-like aggregates to activate and propagate antiviral innate immune response. *Cell* **146**, 448–461 (2011).
14. Cleal, K., Jones, R. E., Grimstead, J. W., Hendrickson, E. A. & Baird, D. M. Chromothripsis during telomere crisis is independent of NHEJ, and consistent with a replicative origin. *Genome Res.* **29**, 737–749 (2019).
15. Cleal, K. & Baird, D. M. Catastrophic Endgames: Emerging Mechanisms of Telomere-Driven Genomic Instability. *Trends Genet. TIG* **36**, 347–359 (2020).
16. Liddiard, K., Grimstead, J. W., Cleal, K., Evans, A. & Baird, D. M. Tracking telomere fusions through crisis reveals conflict between DNA transcription and the DNA damage response. *NAR Cancer* **3**, zcaa044 (2021).
17. Arnoult, N., Van Beneden, A. & Decottignies, A. Telomere length regulates TERRA levels through increased trimethylation of telomeric H3K9 and HP1 α . *Nat. Struct. Mol. Biol.* **19**, 948–956 (2012).
18. Jones, R. E., Grimstead, J. W., Sedani, A., Baird, D. & Upadhyaya, M. Telomere erosion in NF1 tumorigenesis. *Oncotarget* **8**, 40132–40139 (2017).
19. Pepper, C., Baird, D. & Fegan, C. Telomere analysis to predict chronic lymphocytic leukemia outcome: a STELA test to change clinical practice? *Expert Rev. Hematol.* **7**, 701–703 (2014).
20. Norris, K. et al. High-throughput STELA provides a rapid test for the diagnosis of telomere biology disorders. *Hum. Genet.* **140**, 945–955 (2021).
21. Nassour, J. et al. Autophagic cell death restricts chromosomal instability during replicative crisis. *Nature* **565**, 659–663 (2019).

Reviewer Reports on the First Revision:

Referees' comments:

Referee #1 (Remarks to the Author):

The authors have adequately addressed most of the specific points I raised during the initial round of review. I congratulate them on the revised manuscript that contains a lot of high-quality data, which are important and timely.

Although significant new data have been added, the absence of biochemical data on the proposed ZBP1(S)-MAVS signalling axis leaves this aspect of the manuscript weak. The possibility (as mentioned in the initial review) that the role of MAVS is largely to prime cGAS/STING expression remains open in my view, given that the mitochondrial tethering of ZBP1 in pre-crisis cells (Fig 4d-g) is a somewhat artificial system. At a minimum, the authors should acknowledge this possibility in the manuscript and Extended Data Figure 16f.

Minor points

1. I suggest to include the data shown in Referee1 – Figure1 and Referee2 – Figure2 as additional Extended Data Figures. These are high-quality data of interest to specialists in the area.
2. Given the importance of Extended Data Figure 4e, it is surprising this experiment was done only once. Do the authors have repeat experiments? I am wondering whether this figure could be included in the main manuscript.
3. In Extended Data Figure 5d, please change Zbeta to Zalpha2 in the x-axis labelling.

Referee #2 (Remarks to the Author):

The authors have addressed all my critique points very well. I recommend publication of this paper in Nature.

Referee #3 (Remarks to the Author):

This revision has addressed some of my concerns, but I still have reservation about the authors' main model. The main conclusion of this paper is that a short isoform of ZBP1 binds to the TERRA RNA and mediates replicative crisis by activating MAVS and autophagy, thereby leading to cell death. However, considering that autophagic cell death is still controversial, and that there is no evidence that MAVS promotes autophagy, it is still not clear how ZBP1 causes cell death. This could have been straightforward, because ZBP1 is known to activate RIPK3, leading to necrosis and other forms of cell death. However, the authors went to great lengths in the revised manuscript to show that ZBP1 does not engage RIPK3 to cause cell death in human cells. This caused the authors to propose the MAVS-interferon-autophagy-cell death model, which seems convoluted. Now the question becomes: does MAVS activation and/or interferon induction lead to cell death by inducing autophagy? I have not

seen strong evidence from this paper or from the literature that supports this model. As the conclusion is based on data from two cell lines, this also raises the question of whether the model is generally applicable to other cells and is relevant in vivo.

Referee #4 (Remarks to the Author):

Nassour et al. have revised their manuscript “ZBP1-mediated telomere-to-mitochondria signaling prevents cancer initiation” which seeks to extend the understanding of crisis in ageing human fibroblasts carrying either HPV E6/E7 or SV40 large T (SVLT) to allow cells to sidestep an earlier demise by senescence due to the oncogene-mediated neutralization of pRB and p53. Thus, crisis here is defined as “a redundant tumor-suppressor mechanism for replicative senescence that eliminates aberrant cells through autophagy-dependent cell death” and the “checkpoint-deficient” immortalization offered by these oncogenes is cut short by telomere shortening due to crisis that may be averted by overexpression of hTERT. The study implicates two additional distinct interferon activation players (ZBP1 and MAVS) beyond the cGAS-STING-mediated autophagy already been implicated in Ref #1. What comes across clearly to the reader is that an elevated interferon signaling environment accompanies crisis and that reducing that signaling allows a few more cell divisions. Immortalization has long been viewed as a component of oncogenic transformation, best established using rodents, hence the connection to cancer.

The experimental design (developed in Ref #1) is employed to show modest extension of cumulative population doubling when type I interferon signaling has been subdued by either removal of pathogen sensing/signaling or type I interferon receptor components. All have the same impact of extending by 7 to 10 doublings (~5% increment). The assay applies to either IMR90-E6/E7 or WI38-SVLT doubling and is shown graphically, but the manuscript, like the prior publication (Ref #1) does not include an independent quantitative correlate of population doubling. Thus, the data at best suggest that the same range of machinery that controls sensing of pathogen nucleic acids to induce interferon signaling and cell death become associated with the process of crisis. Reduction of interferon signaling extends doubling in human fibroblasts where pRB and p53 have been neutralized by viral oncogenes. With these observations, and the potential to draw connections to cancer, it is important to recall crucial reports that showed human fibroblasts may be transformed to a tumorigenic state by SVLT plus hTERT (Hahn et al., *Nature*, 1999) but not by E6/E7 plus hTERT (Morales et al., *Nature Genetics*, 1999). Thus, the lumping together of these into a single category seems problematic when cancer is on the table. Many additional aspects of this work remain questionable beyond any connection to cancer. Fibroblasts have been widely used to study replicative senescence since the mid-20th century, but less so for the crisis feature that applies to cells that have escaped senescence due to neutralization of pRB and p53. The study does not really address whether interferon signaling drives cancer, normal ageing or the combination of both. It is worth noting that the observation that interferon activation contributes to crisis as studied here was made clearly in Ref #1 and need not be remade at this time.

Looking at the details of the study now that authors have revised the manuscript to address reviewers' previous concerns; however, there remains a striking level of speculation and conceptual posturing around the multiple components that may be implicated in the interferon response. Each

of the pathways implicated here may either be operating in a novel collaboration that is imagined by the authors to engage cGAS-STING, MAVS and ZBP1, or the pathways may all be occurring simultaneously or sequentially over the extended time frame that crisis involves. They may even be occurring well before crisis. Because the authors have used the same assay previously to implicate autophagy machinery and exclude apoptosis (despite many publications promoting apoptosis) the question remains “How does interferon signaling drive crisis” (Nassour et al., 2019). Although authors extend the life of cells beyond crisis 7 to 10 population doublings by disrupting ZBP1 recognition of telomeric repeat-containing nucleic acids as well as my manipulating MAVS (as well as through “cGAS/STING signaling [triggering] a non-canonical form of autophagy capable of executing cell death”), more needs to be done to determine whether these are all in the same pathway. To the reader, it seems likely that this pattern represents consequences of primary drivers already well established to be in the secretome (TNF, IL6, IL1 and other inflammatory cytokines besides type I interferon, possibly in combination) released during senescence/crisis.

Senescence and crisis are considered “mutually exclusive antiproliferative barriers” to cancer progression modeled using human fibroblasts engineered to bypass senescence. In crisis, these cells show “genome instability that often manifests as chromosome end-to-end fusions, nucleoplasmic bridges (NPBs), and micronuclei (MN)” even though full transformation is rare. By exploring crisis in cells that skip the senescence step, authors seek define “how dysfunctional telomeres engage nucleic acid (NA) sensing machineries and activate immune signaling pathways required for crisis-associated cell death”. Using IMR90-E6/E7 or WI38-SVLT immortalized cell lines authors detect distinctive RNA changes associated with crisis. Because of the substantial differences between HPV E6/E7, the reader expects that there should be differences between the transcriptome driven by these different oncogenes, but here they focus on a common inflammatory signature during crisis that develops over a period of several days to a week.

The work seeks then to evaluate “telomere-dependent pathways leading to immune activation” by mutating ZBP1, a known ISG that has been studied as a pathogen sensor recognizing Z-RNA (Zhang et al., Cell, 2020; Koehler et al., Host Cell Microbe, 2021). Because the time frame is prolonged, ZBP1-dependent death and cytokine activation pathways here may play out in parallel with cGAS-STING, for example, although it remains possible that these pathways are also somehow interdependent. The authors have addressed many questions raised by reviewers regarding ZBP1-interacting partner RIPK3, and have excluded contributions from necroptosis; however, the authors have completely overlooked the likely contributions of RIPK1 interactions here, particularly to the death-independent signaling long known to be controlled by ZBP1. RIPK1 has grown in importance controlling cell death-dependent and cell-death independent signaling in human cells as well as in mouse settings. RIPK1 is recognized as a master regulator of cell survival and death, as well as death independent inflammation that is controlled by inflammatory cytokines. This was dramatically revealed when the perinatal lethality of RIPK1-deficient mice was shown to be rescued by elimination of combined TNF and type I interferon pathways that are mediated via caspase-8 and RIPK3, with influence from ZBP1 over how the whole picture unfolds (see reviews by Newton et al., such as in Science, 2021). While the data presented addresses “the importance of the crosstalk between NA-sensing pathways in potentiating the immune response during crisis”, the overall dissection of the signaling involved in the ZBP1-MAVS connection leaves concerns that have not been fully addressed. Important confounders here are the slow nature of crisis-dependent death and the likelihood that there are

consequences comingled with drivers of crisis.

The observations in Figure 1 appear straightforward but not fully fleshed out. ZBP1 is identified and shown to influence crisis by CRISPR mutagenesis, but there is no mention of either autophagy players or cGAS-STING being detected in the screen and no indication of whether these lay in the same pathway. If, as the authors project “cGAS/STING pathway is responsible for an initial transcriptional induction of ZBP1 that primes cells for a rapid response to aberrant accumulation of cytosolic NA species”, the combined elimination of both pathways would have the same phenotype as ZBP1-deficiency; however, if parallel, the combined mutants would have a greater impact here. In fact, this leaves the reader unclear on how this new observation fits with earlier work, noting again the subjective assay system and slowly developing endpoints that rely on a subjective assessment by the observer. Authors report that pre-crisis, E6-E7 immortalized IMR90 fibroblasts “reconstituted with ZBP1 containing point mutations in key conserved residues involved in NA binding (N141A, Y145A) 32 or RHIM1-based interaction (I206A, Q207A, I208A, G209A) 33,34 failed to induce ISGs and remained insensitive to telomere-driven cell death”. This description brings to mind diverse studies on ZBP1 as a sensor of RNA in a number of virus infection settings (herpesviruses, influenza viruses and poxvirus vaccinia) as well as non-infectious settings where the sensor responds to endogenous RNA. In some cases, an NF- κ B or IFN response is measured and in others cell death is the measured outcome. That being said, “autophagy-dependent cell death” has not come up in the other systems with any regularity and so this connection is highly suspect here. If it is to be implicated here this point must be made with additional clear data.

Authors show hTERT expression prolongs population doubling and reduces activation of ISGs, amongst which is ZBP1 (to be expected from past reports). Authors suggest that the Za1-deficient form of, ZBP1(S) “potentiated an IFN response when expressed in pre-crisis cells with short telomeres”. This dual cGAS-STING followed by ZBP1-MAVS IFN response amplification is clearly also tied into the action of interferon itself, leaving questions about complexity and order of events as well as any role of autophagy machinery. That being said, these data actually mirror observations performed to look at NF- κ B and type I interferon activation by ZBP1 from studies that started with Takaoka et al., 2007, and Wang et al., 2008, and were extended by Kaiser et al., 2008 (Ref 33) and Rebsamen et al., 2009, (Ref 34) who demonstrated the importance of RIPK1 RHIM recruitment in signaling. Importantly, and in contrast to the perspective that the authors give in their text, this RHIM signaling work was performed in human cells and not just in mouse cells. Activation of type I IFNs by ZBP1(S) shown here in human cells are consistent with an increased sensitivity this ZBP1 isoform to recruitment into granules (Deigendesch et al., 2006; Pham et al., 2006) which has been shown to occur in a RIPK1 RHIM-dependent pathway (Kaiser et al., 2008). In these earlier studies, signaling was not tied to MAVS or to mitochondria, increasing the need for complete elucidation of any connection that is claimed here. The authors demonstration that “individual suppression of cGAS, STING, or IFN receptors (IFNAR1 and IFNAR2) resulted in a significant reduction of ZBP1” does not address the steps involved. Combined suppression will be necessary to determine whether these lie in the same pathway as suggested by the authors or whether each is following its distinct and previously described pathway. It seems possible that in the time frame of crisis, a number of parallel, possibly cross-dependent pathways may be activated or, alternatively, that some underlying metabolic changes disrupts these signaling molecules independently. No matter, the picture must be clear. Each will need to be fully fleshed out before initiator can be identified and cross-talk can be

convincingly demonstrated. Authors should appreciate that studies looking into activation of type I IFN response by herpesviruses have touched upon the themes being evaluated here and in the same human fibroblast cell type. For example, human CMV was shown to depend on STING but be independent of MAVS (Defillipis 2010). In addition, Pham et al., 2013 (Figure 5) working on HSV defined a role for RHIM signaling and RIPK1 in the activation of IFN in human cells. Overall, it is an oversight that the authors have not evaluated the contributions of RIPK1 to the outcomes being assessed.

Do pre-crisis cells with disrupted ZBP1 show the same reduced type I interferon as they show during crisis? Prolonged culture in a low tonic interferon environment as is afforded by the disruption of ZBP1 alone might contribute to the continued proliferation 7 to 10 doublings beyond the controls and have nothing directly to do with the actual crisis itself.

This process plays out slowly allowing ample time for conventional signaling to play out and influence subsequent innate signaling steps. Need to define the MAVS complex and link it to ZBP1 signaling for this work to be considered unambiguous. Does knocking out RIPK1 extend cells similarly to MAVS, and if combined what is the impact prior to crisis as well as through crisis?

Author Rebuttals to First Revision:

Response to referees:

We thank Referees 1, 2 and 3 for their time, support and constructive criticism through two review cycles. As requested by the editor, below we propose approaches to address their few, but legitimate remaining concerns. We also appreciate any extra effort going forward to weigh in on the legitimacy of what has been a more-difficult situation with regard to a recalcitrant reviewer 4. It has been a challenge to address the unclear and, in many places, redundant and aggressive comments of review 4. We hope the other referees can independently evaluate our conscientious and extensive effort to constructively address the original concerns of referee 4. From our perspective, this has been an unnecessarily uphill battle, since the referee clearly states an open bias against our work or anything that speaks to alternative mechanisms to what they 'believe' should be true. For example, from the second-to-last paragraph of the original review: "Most importantly, and also related to impact, autophagy remains a highly controversial mode of death regardless of the number of reports that implicate this pathway as a manner in which cells die." This is just one example of broad and unreasonable statements made in both the original and second review by reviewer 4. Thus, we remain truly disheartened and frustrated by the consistently unconstructive, and at times offensive, nature of review 4's critique, as well as their blatant disregard of our previous published paper already showing autophagy-driven cell death in crisis in multiple cell types (Nassour et al., *Nature* **565**, 659–663 (2019)). If this were not enough, in the initial review, reviewer 4 goes as far as to insinuate that we lack scientific integrity by suggesting that we "pick and choose" our cell lines to conform to the data we want to see. This accusation is unwarranted, and we interpret it as an attempt to discredit us to prevent publication of the study. Finally, in the original review, as well as in the response to our revision, referee 4 strongly condemns us for not presenting data that are actually present and discussed in the manuscript (e.g., the ability to induce an ISG driven crisis response in young cells upon expression of ZBP1 and TERRA; the identification of cGAS and STING as hits in the survival screen; the dependence of ZBP1(S) expression on cGAS/STING; the absence of RIPK3-mediated PANoptosis (Pyroptosis, Apoptosis, and Necroptosis) in ZBP1-driven cell death in human cells). We have thoroughly and successfully completed most the experiments requested by the reviewer 4 in the first round of revision. The results achieved further exemplify and support our original conclusions. After reading the second revision, it is disappointing to see these results and efforts completely unaddressed, overlooked, and unjustly labeled imaginative. We interpret this to mean that reviewer 4 is either not carefully reading/reviewing the manuscript, is trying to misrepresent what we have done to the other reviewers/editor, or is otherwise attempting to undermine the review process to prevent publication.

Despite the above situation, we made the strongest effort possible to address referee 4's concerns. In particular, as requested, we better addressed the requirement for MAVS and RIPK3-dependent PANoptosis in cell death during replicative crisis. It turns out that RIPK3 is not involved in the pathway presented, yet referee 4 ignores these data and continues to insist that RIPK3 must play role and questions our integrity. If a referee is not willing to accept/believe that the data we are presenting is valid and true, this puts us in a no-win situation and is counter to the normal peer-review process.

Like with any study, we do not contend that we have unraveled every conceivable detail of the signaling pathway between telomeres, mitochondria, inflammation, autophagy, cell death and cancer formation. However, our data is extensive, robust and legitimate and as a whole illuminates a very novel pathway, through which telomere metabolism intersects with DNA- and RNA-sensing innate immune pathways that leads to ZBP1-dependent MAVS activation at the MOM, thereby eliminating cells with dysfunctional telomeres through inflammation. We hope

referees 1, 2 and 3 share our excitement about bringing such diverse scientific fields together and providing a study that is highly novel and impactful on many levels.

Referee #1 (Remarks to the Author):

The authors have adequately addressed most of the specific points I raised during the initial round of review. I congratulate them on the revised manuscript that contains a lot of high-quality data, which are important and timely.

Although significant new data have been added, the absence of biochemical data on the proposed ZBP1(S)-MAVS signalling axis leaves this aspect of the manuscript weak. The possibility (as mentioned in the initial review) that the role of MAVS is largely to prime cGAS/STING expression remains open in my view, given that the mitochondrial tethering of ZBP1 in pre-crisis cells (Fig 4d-g) is a somewhat artificial system. At a minimum, the authors should acknowledge this possibility in the manuscript and Extended Data Figure 16f.

Minor points

1. I suggest to include the data shown in Referee1 – Figure1 and Referee2 – Figure2 as additional Extended Data Figures. These are high-quality data of interest to specialists in the area.
2. Given the importance of Extended Data Figure 4e, it is surprising this experiment was done only once. Do the authors have repeat experiments? I am wondering whether this figure could be included in the main manuscript.
3. In Extended Data Figure 5d, please change Zbeta to Zalpha2 in the x-axis labelling.

Response: First, we thank the reviewer for the positive comments about our responsiveness and the timeliness and importance of the work. While we are happy to better discuss the possibility that MAVS primes cGAS-STING signaling, this is not the simplest explanation of our data. cGAS and STING are not ISGs, and their expression levels do not vary upon interferon signaling, as we show in the manuscript (e.g. MAVS suppression by CRISPR had no impact on STING protein levels – Extended Data Figure 16b). Furthermore, we show throughout the manuscript that the ZBP1 signaling cascade can be uncoupled from cGAS-STING signaling entirely, simply by expressing ZBP1(S) and Terra, yet the signal remains MAVS-dependent. It is therefore very difficult for us to reason that MAVS signaling primes cGAS-STING expression. However, to directly test this as requested, we will examine cGAS and STING mRNA and protein levels in the background of MAVS suppression. In addition, we will examine cGAS activity by measuring cGAMP levels by ELISA in WT and MAVS KO cells. If these arguments and new data do not fully convince the reviewer, we are happy to acknowledge the MAVS-priming possibility in the final revised manuscript as requested. The additional minor points can easily be accommodated, and we would like to point out that Extended Figure 4e shows three independent replicates of the expression analysis within a single experiment. This experiment (excluding siRIG-I+siMDA5) was already performed three independent times in the first version of the manuscript. We will however include two additional biological replicates in the revised manuscript.

Referee #3 (Remarks to the Author):

This revision has addressed some of my concerns, but I still have reservation about the authors' main model. The main conclusion of this paper is that a short isoform of ZBP1 binds to the TERRA RNA and mediates replicative crisis by activating MAVS and autophagy, thereby leading to cell death. However, considering that autophagic cell death is still controversial, and that there is no evidence that MAVS promotes autophagy, it is still not clear how ZBP1 causes cell death. This could have been straightforward, because ZBP1 is known to activate RIPK3, leading to necrosis

and other forms of cell death. However, the authors went to great lengths in the revised manuscript to show that ZBP1 does not engage RIPK3 to cause cell death in human cells. This caused the authors to propose the MAVS-interferon-autophagy-cell death model, which seems convoluted. Now the question becomes: does MAVS activation and/or interferon induction lead to cell death by inducing autophagy? I have not seen strong evidence from this paper or from the literature that supports this model. As the conclusion is based on data from two cell lines, this also raises the question of whether the model is generally applicable to other cells and is relevant in vivo.

Response: We would like to emphasize that this manuscript focuses on the signaling cascade between short telomeres, mitochondria, and inflammation, where we uncover an entirely novel human-specific pathway. Also, we would like to point out that we already showed convincingly that autophagic cell death is the mechanism operating during replicative crisis in Nassour et al, Nature 565, 659-663, 2019 in fibroblasts and in epithelial cells (i.e. this pathway operates in multiple, relevant cells types). At this point, there are many forms of cell death that operate in very context-specific conditions, so that autophagy-mediated cell death is operating under the special conditions of crisis should not be difficult to accept in principle. Regardless, this manuscript does not focus on the autophagy axis, but rather the very novel signaling pathway between telomeres and mitochondria (as already stated). Finally, it is important to point out that cGAS-STING signaling activates autophagy and, in fact, it is proposed that this is the primordial function of the pathway (Gui et al. Nature 2019). Thus, we argue our model where cGAS-STING and MAVS-Zbp1 collaborate to induce an interferon and autophagy-dependent form of cell death is the simplest synthesized explanation of our current data and our prior work (Nassour et al.). The precise intersection of these two lethal signaling events in cell death we agree is a priority subject of future work for our ongoing collaboration.

However, to experimentally address the referee's concern, we will activate MAVS by ZBP1 in cells lacking IFN receptors (IFNAR1 and IFNAR2) and measure both autophagy and cell death frequencies. The results obtained from this experiment will address whether cell death by autophagy is mediated directly by MAVS or indirectly through a second wave of interferon production. In addition, we will confirm ZBP1 upregulation, mitochondrial localization, and inflammation activation in epithelial cells in replicative crisis to address conservation of the pathway in another relevant cell type.

Referee #4 (Remarks to the Author):

Nassour et al. have revised their manuscript "ZBP1-mediated telomere-to-mitochondria signaling prevents cancer initiation" which seeks to extend the understanding of crisis in ageing human fibroblasts carrying either HPV E6/E7 or SV40 large T (SVLT) to allow cells to sidestep an earlier demise by senescence due to the oncogene-mediated neutralization of pRB and p53. Thus, crisis here is defined as "a redundant tumor-suppressor mechanism for replicative senescence that eliminates aberrant cells through autophagy-dependent cell death" and the "checkpoint-deficient" immortalization offered by these oncogenes is cut short by telomere shortening due to crisis that may be averted by overexpression of hTERT. The study implicates two additional distinct interferon activation players (ZBP1 and MAVS) beyond the cGAS-STING-mediated autophagy already been implicated in Ref #1. What comes across clearly to the reader is that an elevated interferon signaling environment accompanies crisis and that reducing that signaling allows a few more cell divisions. Immortalization has long been viewed as a component of oncogenic transformation, best established using rodents, hence the connection to cancer.

We would like to point out that it is not us that defined replicative crisis, but that it has been a well-accepted term for the Mortality (M2) plateau for decades.

The experimental design (developed in Ref #1) is employed to show modest extension of cumulative population doubling when type I interferon signaling has been subdued by either removal of pathogen sensing/signaling or type I interferon receptor components. All have the same impact of extending by 7 to 10 doublings (~5% increment). The assay applies to either IMR90-E6/E7 or WI38-SVLT doubling and is shown graphically, but the manuscript, like the prior publication (Ref #1) does not include an independent quantitative correlate of population doubling.

It is important to emphasize again that ZBP1 KO cells bypassed crisis and continue to divide for an additional 7-10 population doublings. A growth difference in 7-10 PD difference correspond to an increase in cell number by 100-1000 fold.

Cell numbers and population doublings were determined through direct counting, and are not a correlation, but a direct relation. In addition to the growth curve analysis, we have performed automated real-time assessments by IncuCyte live-cell imager to provide a direct measurement of the percentage of dead cells. Live-cell imaging showed a significant reduction in the percentage of dead cells (while 60% of control cells (sgLUC, sgGFP) were dead at 24 hours post-imaging, only 5-10% of sgZBP1 cells were dying). All of these data were included in the first version of the manuscript (extended data figure 3) and we hope the reviewer can appreciate that this is not a 'modest' effect.

Thus, the data at best suggest that the same range of machinery that controls sensing of pathogen nucleic acids to induce interferon signaling and cell death become associated with the process of crisis. Reduction of interferon signaling extends doubling in human fibroblasts where pRB and p53 have been neutralized by viral oncogenes. With these observations, and the potential to draw connections to cancer, it is important to recall crucial reports that showed human fibroblasts may be transformed to a tumorigenic state by SVLT plus hTERT (Hahn et al., Nature, 1999) but not by E6/E7 plus hTERT (Morales et al., Nature Genetics, 1999). Thus, the lumping together of these into a single category seems problematic when cancer is on the table. Many additional aspects of this work remain questionable beyond any connection to cancer. Fibroblasts have been widely used to study replicative senescence since the mid-20th century, but less so for the crisis feature that applies to cells that have escaped senescence due to neutralization of pRB and p53. The study does not really address whether interferon signaling drives cancer, normal ageing or the combination of both. It is worth noting that the observation that interferon activation contributes to crisis as studied here was made clearly in Ref #1 and need not be remade at this time.

We would like to point out that the Nature paper by Bill Hahn that the referee points out did in fact use the whole early region of SV40, not just the LT antigen. Consequently, not only p53 and RB were targeted, but many other pathways.

Our study was not designed to directly address the question whether interferon signaling drives cancer, and our manuscript does not make that claim. What we show here is a novel pathway, where telomeres intersect with mitochondria via DNA and RNA sensing pathways to cause cell death in crisis, which is a known tumor avoidance mechanism. Also, the referee's suggestion that interferon activation contributes to crisis has already been addressed in Nassour et al., Nature, 2019, is incorrect, as that publication, while implicating cGAS/STING in the recognition of products from chromosome breakage, focuses on the autophagy aspects and not the innate immune

response. We previously showed that cell death during replicative crisis is mediated by autophagy and that this pathway is activated by the breakage of fused chromosome, accumulation of cytosolic DNA species detected by cGAS/STING. We did not study the innate immune response and did not include a single experiment assessing the involvement of IFN signaling in cell death during crisis.

Looking at the details of the study now that authors have revised the manuscript to address reviewers' previous concerns; however, there remains a striking level of speculation and conceptual posturing around the multiple components that may be implicated in the interferon response. Each of the pathways implicated here may either be operating in a novel collaboration that is imagined by the authors to engage cGAS-STING, MAVS and ZBP1, or the pathways may all be occurring simultaneously or sequentially over the extended time frame that crisis involves. They may even be occurring well before crisis. Because the authors have used the same assay previously to implicate autophagy machinery and exclude apoptosis (despite many publications promoting apoptosis) the question remains "How does interferon signaling drive crisis" (Nassour et al., 2019). Although authors extend the life of cells beyond crisis 7 to 10 population doublings by disrupting ZBP1 recognition of telomeric repeat-containing nucleic acids as well as my manipulating MAVS (as well as through "cGAS/STING signaling [triggering] a non-canonical form of autophagy capable of executing cell death"), more needs to be done to determine whether these are all in the same pathway. To the reader, it seems likely that this pattern represents consequences of primary drivers already well established to be in the secretome (TNF, IL6, IL1 and other inflammatory cytokines besides type I interferon, possibly in combination) released during senescence/crisis.

We disagree and argue we have directly addressed the order of events and inter-dependence of the pathways in question throughout the manuscript. We have shown that replicative crisis requires the co-activation of the cGAS and ZBP1-mediated immune pathways. When cGAS, STING, ZBP1, or MAVS are suppressed cells evade cell death and bypass crisis despite their critically short and dysfunctional telomeres. We also demonstrated that the cGAS-STING and ZBP1-MAVS pathways are interconnected and that ZBP1 (mRNA and protein) levels are induced by the cGAS-STING-IFN signaling pathway. In the absence of cGAS-STING, ZBP1 is not induced, and therefore the ZBP1-MAVS-IFN pathway cannot be activated. All the conclusions in the manuscript are well supported by series of controlled experiments repeated 2-3 independent times and shown in the main and extended figures. It is disappointing to see these data disregarded and labeled as imaginative.

Also, we state again that this manuscript focuses on crisis, not senescence and that these processes should not be lumped together as the reviewer states. The SASP response in senescence is well-established and drives cell cycle arrest, not cell death, and is not relevant here.

Senescence and crisis are considered "mutually exclusive antiproliferative barriers" to cancer progression modeled using human fibroblasts engineered to bypass senescence. In crisis, these cells show "genome instability that often manifests as chromosome end-to-end fusions, nucleoplasmic bridges (NPBs), and micronuclei (MN)" even though full transformation is rare. By exploring crisis in cells that skip the senescence step, authors seek define "how dysfunctional telomeres engage nucleic acid (NA) sensing machineries and activate immune signaling pathways required for crisis-associated cell death". Using IMR90-E6/E7 or WI38-SVLT immortalized cell lines authors detect distinctive RNA changes associated with crisis. Because of

the substantial differences between HPV E6/E7, the reader expects that there should be differences between the transcriptome driven by these different oncogenes, but here they focus on a common inflammatory signature during crisis that develops over a period of several days to a week.

We would like to point out again that the crisis-autophagy connection is not simply a consequence of HPVE6/E7 expression, as it has also been shown in fibroblasts and epithelial cells where senescence has been bypassed through dominant negative p53 and cdk4 expression, as well as in cells that spontaneously escape senescence (Nassour et al, Nature, 2019). Thus, the E6/E7 model is simply a convenient and systematic way to study crisis and is not idiosyncratic.

The work seeks then to evaluate “telomere-dependent pathways leading to immune activation” by mutating ZBP1, a known ISG that has been studied as a pathogen sensor recognizing Z-RNA (Zhang et al., Cell, 2020; Koehler et al., Host Cell Microbe, 2021). Because the time frame is prolonged, ZBP1-dependent death and cytokine activation pathways here may play out in parallel with cGAS-STING, for example, although it remains possible that these pathways are also somehow interdependent. The authors have addressed many questions raised by reviewers regarding ZBP1-interacting partner RIPK3, and have excluded contributions from necroptosis; however, the authors have completely overlooked the likely contributions of RIPK1 interactions here, particularly to the death-independent signaling long known to be controlled by ZBP1. RIPK1 has grown in importance controlling cell death-dependent and cell-death independent signaling in human cells as well as in mouse settings. RIPK1 is recognized as a master regulator of cell survival and death, as well as death independent inflammation that is controlled by inflammatory cytokines. This was dramatically revealed when the perinatal lethality of RIPK1-deficient mice was shown to be rescued by elimination of combined TNF and type I interferon pathways that are mediated via caspase-8 and RIPK3, with influence from ZBP1 over how the whole picture unfolds (see reviews by Newton et al., such as in Science, 2021). While the data presented addresses “the importance of the crosstalk between NA-sensing pathways in potentiating the immune response during crisis”, the overall dissection of the signaling involved in the ZBP1-MAVS connection leaves concerns that have not been fully addressed. Important confounders here are the slow nature of crisis-dependent death and the likelihood that there are consequences comingled with drivers of crisis.

This is a misrepresentation of the facts. We focused on RIPK3, as the referee had requested to address RIPK3 in particular in their initial review. This was justified, since RIPK3 is one of the master regulators of ZBP1 driven necroptosis. Targeting RIPK3 also eliminates RIPK1 function. In the revised version, we directly addressed the role of RIPK3 as requested, but found it is not involved (see Extended Data Figure 6). Furthermore, RIPK3 and RIPK1 are not expressed in IMR90 fibroblasts. It is therefore incorrect to state ‘the authors have completely overlooked the likely contributions of RIPK1 interactions here, particularly to the death-independent signaling long known to be controlled by ZBP1.’

The reviewer repeatedly refers the “slow nature” of the crisis program as if this is really a confounding factor. We fail to see the point here. Crisis is what it is and takes the time it requires. This really has no bearing on defining phenotypes and signaling pathways involved in our opinion.

The observations in Figure 1 appear straightforward but not fully fleshed out. ZBP1 is identified and shown to influence crisis by CRISPR mutagenesis, but there is no mention of either autophagy players or cGAS-STING being detected in the screen and no indication of whether

these lay in the same pathway. If, as the authors project “cGAS/STING pathway is responsible for an initial transcriptional induction of ZBP1 that primes cells for a rapid response to aberrant accumulation of cytosolic NA species”, the combined elimination of both pathways would have the same phenotype as ZBP1-deficiency; however, if parallel, the combined mutants would have a greater impact here. In fact, this leaves the reader unclear on how this new observation fits with earlier work, noting again the subjective assay system and slowly developing endpoints that rely on a subjective assessment by the observer.

We show in Figure 1a that cGAS, STING and IFN receptors were detected in the screen. Again, dependence of the cGAS/STING and ZBP1/MAVS pathways are directly addressed throughout the manuscript. We demonstrate that ZBP1 is not expressed in young cells (Figure 2c; Extended Data Figure 4c), that forced expression of ZBP1 in young cells has no effect (Figure 2d), that expression of ZBP1 in crisis is dependent on cGAS and STING (Extended Data Figure 4f), that MAVS is required for the signaling cascade (Figures 4d,g; Extended data Figures 15b, c; Extended data Figures 16b,c). It is not clear why this reviewer resorts to statements like “subjective assay system” and “subjective assessment” other than as a deliberate attempt to infer that we are not objectively gathering and interpreting our data. Again, this is an inappropriate reviewer tactic for a just peer review system.

Authors report that pre-crisis, E6-E7 immortalized IMR90 fibroblasts “reconstituted with ZBP1 containing point mutations in key conserved residues involved in NA binding (N141A, Y145A) 32 or RHIM1-based interaction (I206A, Q207A, I208A, G209A) 33,34 failed to induce ISGs and remained insensitive to telomere-driven cell death”. This description brings to mind diverse studies on ZBP1 as a sensor of RNA in a number of virus infection settings (herpesviruses, influenza viruses and poxvirus vaccinia) as well as non-infectious settings where the sensor responds to endogenous RNA. In some cases, an NF- κ B or IFN response is measured and in others cell death is the measured outcome. That being said, “autophagy-dependent cell death” has not come up in the other systems with any regularity and so this connection is highly suspect here. If it is to be implicated here this point must be made with additional clear data.

The notion that a discovery should be discounted because it has not been reported previously in other systems is, for lack of a better word, ridiculous. Indeed, the whole point of science is to discover and understand things that were not observed before. Thus to conclude that our data are “highly suspect” because autophagy has not been reported in other manuscripts addressing ZBP1 during viral infections, etc., is illogical and again seems to hint that we are somehow not ethical or objective. This is unacceptable behavior by a reviewer in peer review in our opinion.

Authors show hTERT expression prolongs population doubling and reduces activation of ISGs, amongst which is ZBP1 (to be expected from past reports). Authors suggest that the Za1-deficient form of, ZBP1(S) “potentiated an IFN response when expressed in pre-crisis cells with short telomeres”. This dual cGAS-STING followed by ZBP1-MAVS IFN response amplification is clearly also tied into the action of interferon itself, leaving questions about complexity and order of events as well as any role of autophagy machinery. That being said, these data actually mirror observations performed to look at NF- κ B and type I interferon activation by ZBP1 from studies that started with Takaoka et al., 2007, and Wang et al., 2008, and were extended by Kaiser et al., 2008 (Ref 33) and Rebsamen et al., 2009, (Ref 34) who demonstrated the importance of RIPK1 RHIM recruitment in signaling. Importantly, and in contrast to the perspective that the authors give in their text, this RHIM signaling work was performed in human cells and not just in mouse cells. Activation of type I IFNs by ZBP1(S) shown here in human cells are consistent with an increased

sensitivity this ZBP1 isoform to recruitment into granules (Deigendesch et al., 2006; Pham et al., 2006) which has been shown to occur in a RIPK1 RHIM-dependent pathway (Kaiser et al., 2008). In these earlier studies, signaling was not tied to MAVS or to mitochondria, increasing the need for complete elucidation of any connection that is claimed here. The authors demonstration that “individual suppression of cGAS, STING, or IFN receptors (IFNAR1 and IFNAR2) resulted in a significant reduction of ZBP1” does not address the steps involved. Combined suppression will be necessary to determine whether these lie in the same pathway as suggested by the authors or whether each is following its distinct and previously described pathway. It seems possible that in the time frame of crisis, a number of parallel, possibly cross-dependent pathways may be activated or, alternatively, that some underlying metabolic changes disrupts these signaling molecules independently. No matter, the picture must be clear. Each will need to be fully fleshed out before initiator can be identified and cross-talk can be convincingly demonstrated. Authors should appreciate that studies looking into activation of type I IFN response by herpesviruses have touched upon the themes being evaluated here and in the same human fibroblast cell type. For example, human CMV was shown to depend on STING but be independent of MAVS (Defillipis 2010). In addition, Pham et al., 2013 (Figure 5) working on HSV defined a role for RHIM signaling and RIPK1 in the activation of IFN in human cells. Overall, it is an oversight that the authors have not evaluated the contributions of RIPK1 to the outcomes being assessed.

We can only repeat that the dependence of the cGAS/STING and ZBP1/MAVS pathways are directly addressed throughout the manuscript. We demonstrate that ZBP1 is not expressed in young cells (Figure 2c; Extended Data Figure 4c), that forced expression of ZBP1 in young cells has no effect (Figure 2d), that expression of ZBP1 in crisis is dependent on cGAS and STING (Extended Data Figure 4f), that MAVS is required for the signaling cascade (Figures 4d,g; Extended data Figures 15b, c; Extended data Figures 16b,c).

And again, the RIPK3/RIPK1 pathway as an effector has been excluded, as is shown in Extended data Figure 6.

Do pre-crisis cells with disrupted ZBP1 show the same reduced type I interferon as they show during crisis? Prolonged culture in a low tonic interferon environment as is afforded by the disruption of ZBP1 alone might contribute to the continued proliferation 7 to 10 doublings beyond the controls and have nothing directly to do with the actual crisis itself.

Young cells neither express ZBP1 (Figure 2c; Extended Data Figure 4c), nor does forced expression of ZBP1 in young cells have an effect (Figure 2d). Furthermore, we demonstrate in the reverse experiment that expression of ZBP1 (S) together with TERRA induces crisis in young cells (Figure 3e, f), demonstrating a direct link to crisis.

This process plays out slowly allowing ample time for conventional signaling to play out and influence subsequent innate signaling steps. Need to define the MAVS complex and link it to ZBP1 signaling for this work to be considered unambiguous. Does knocking out RIPK1 extend cells similarly to MAVS, and if combined what is the impact prior to crisis as well as through crisis?

We have demonstrated that MAVS is required for the signaling cascade (Figures 4d,g; Extended data Figures 15b, c; Extended data Figures 16b,c). We have excluded a role for RIPK3 in the suggested pathway (Extended data Figure 6) and RIPK3/RIPK1 did not emerge in the screen for life span extension, which is not surprising, given that they are not expressed in IMR90 cells.

As stated above in another context, the reviewer repeatedly refers to the idea that the crisis program “plays out slowly” as if this is really a confounding factor. We fail to see the point here. Crisis is what it is and takes the time it requires. This really has no bearing on defining phenotypes and signaling pathways involved in our opinion.

Reviewer Reports on the Second Revision:

Referees' comments:

Referee #1 (Remarks to the Author):

Referee 1 thanks the authors for their reply to the comments raised during the second round of review.

It is not correct that cGAS and STING are not ISGs. For example, MB21D1 (the gene encoding cGAS) is one of the ISGs included in a widely used collection of ISGs (Schoggins et al., Nature 2011). However, it is possible that cGAS is not induced by IFNs in the specific cells the authors use. In any case, the authors propose to measure cGAS and STING mRNA levels in MAVS-depleted cells. This is a good idea and straightforward to do. The authors also want to measure cGAMP levels by ELISA. The reviewer's lab uses cGAMP ELISA and it would be surprising if baseline/tonic cGAMP levels can be detected. However, this may be possible in crisis cells. Again, this is easy enough to try.

Referee 1 agrees with the authors that minor points 1-3 can be easily addressed, in particular if the siRIG-I + siMDA5 experiment in Extended data Figure 4e is repeated two more times.

In sum, this reviewer remains very supportive of publication in Nature and looks forward to seeing the data from the small number of additional experiments.

Referee #3 (Remarks to the Author):

I offer my assessment of the strengths and weaknesses of the paper by Nassour et al.

Strengths:

- 1) Finding that ZBP1 is important for replicative crisis
- 2) Identifying that TERRA RNA binds to ZBP1 and probably mediates ZBP1 activation during replicative crisis
- 3) Showing that ZBP1 forms filaments on the mitochondrial surface during replicative crisis
- 4) Showing that ZBP1 induces ISGs through MAVS
- 5) The finding that ZBP1 activates MAVS in response to telomere dysfunction in a manner independent of RIG-I and MDA5 is novel and interesting if proven true.

Weaknesses:

- 1) It is not clear how ZBP1 causes cell death, if this does not involve RIPK1 or RIPK3. If the authors insist on an autophagy-dependent mechanism, there is no evidence that ZBP1, MAVS or interferons causes autophagy-induced cell death. This is a very important issue, because the paper is about the role of ZBP1 in replicative crisis-induced cell death. Most data presented in the paper were on expression of interferon-stimulated genes, which do not equate cell death.

2) The study is limited to two oncogene-immortalized human fibroblast cell lines. It's not clear whether this can apply to other cells and "cancer initiation" in vivo. The data are not sufficient to support the title: "ZBP1-mediated telomere-to-mitochondria signaling prevents cancer initiation".

Author Rebuttals to Second Revision:

Response to referees:

We thank Referees 1, 2 and 3 for their time, support, and constructive criticism through two review cycles. Below, is our point-by-point response to each reviewer. New data have been included in the main figures and extended data figures, as indicated. In addition, due to space limitations, some data are only included in this response letter for the referees to consider.

Referee #1 (Remarks to the Author):

The authors have adequately addressed most of the specific points I raised during the initial round of review. I congratulate them on the revised manuscript that contains a lot of high-quality data, which are important and timely.

Although significant new data have been added, the absence of biochemical data on the proposed ZBP1(S)-MAVS signaling axis leaves this aspect of the manuscript weak. The possibility (as mentioned in the initial review) that the role of MAVS is largely to prime cGAS/STING expression remains open in my view, given that the mitochondrial tethering of ZBP1 in pre-crisis cells (Fig 4d-g) is a somewhat artificial system. At a minimum, the authors should acknowledge this possibility in the manuscript and Extended Data Figure 16f.

Response: We have now performed a set of experiments to address the role of MAVS in priming cGAS-STING signaling and the data is presented here in Referee 1 - Figure 1:

1-cGAS and STING were not induced during crisis and their protein and mRNA levels did not vary significantly between growing and crisis cells.

2-Depletion of MAVS in pre-crisis cells did not result in a reduction in cGAS and STING protein levels.

3- Depletion of MAVS in pre-crisis cells did not result in a reduction in cGAS activity as measured by the production of 2'3'-cGAMP.

Taken together, these results indicate that MAVS has no detectable priming effects on the cGAS-STING signaling pathway.

Referee 1 - Figure 1. **a**, Scatter plot with bars showing TPM values of cGAS and STING transcripts in growing and crisis IMR90^{E6E7} obtained from the RNA-sequencing data presented in Extended Data Figure 1. One experiment was performed in three technical replicates. **b**, Immunoblotting of growing and crisis IMR90^{E6E7}. GAPDH loading control. Two independent experiments were performed. **c**, Immunoblotting of pre-crisis IMR90^{E6E7} transfected with siMAVS or non-targeting control siRNA at day 6 post-transfection. GAPDH loading control. Two independent experiments were performed. **d**, Scatter plot with bars showing 2'3'-cGAMP levels in cell lysates obtained from pre-crisis cells as in **c**. The low levels of cGAMP were detected by Cayman Chemical 2'3'-cGAMP

ELISA kit (Interchim, Cat#:501700) according to manufacturer's instruction. Two independent experiments were performed.

Minor points

1. I suggest to include the data shown in Referee1 – Figure1 and Referee2 – Figure2 as additional Extended Data Figures. These are high-quality data of interest to specialists in the area.

Response: *Due to space limitations we prefer to keep these data in the 'response to referees' section, which we support to be available online for readers. We thank the referee for understanding.*

2. Given the importance of Extended Data Figure 4e, it is surprising this experiment was done only once. Do the authors have repeat experiments? I am wondering whether this figure could be included in the main manuscript.

Response: *We have now performed two additional biological replicates for the experiment in Extended Data Figure 4e and included all the graphs in the manuscript.*

3. In Extended Data Figure 5d, please change Zbeta to Zalpha2 in the x-axis labelling.

Response: *Extended Data Figure 5d has been corrected.*

Referee #3 (Remarks to the Author):

This revision has addressed some of my concerns, but I still have reservation about the authors' main model. The main conclusion of this paper is that a short isoform of ZBP1 binds to the TERRA RNA and mediates replicative crisis by activating MAVS and autophagy, thereby leading to cell death. However, considering that autophagic cell death is still controversial, and that there is no evidence that MAVS promotes autophagy, it is still not clear how ZBP1 causes cell death. This could have been straightforward, because ZBP1 is known to activate RIPK3, leading to necrosis and other forms of cell death. However, the authors went to great lengths in the revised manuscript to show that ZBP1 does not engage RIPK3 to cause cell death in human cells. This caused the authors to propose the MAVS-interferon-autophagy-cell death model, which seems convoluted. Now the question becomes: does MAVS activation and/or interferon induction lead to cell death by inducing autophagy? I have not seen strong evidence from this paper or from the literature that supports this model. As the conclusion is based on data from two cell lines, this also raises the question of whether the model is generally applicable to other cells and is relevant in vivo.

Response: *As requested by the reviewer, we have now performed a series of experiments to address the autophagy-dependent cell death pathway driven by ZBP1(S)-MAVS signaling and the new results were included in the manuscript as Extended Data Figures.*

- siRNA-mediated suppression of RIPK3/NLRP3/PYCARD/Caspase-1-mediated pyroptosis, RIPK1/FADD/caspase-8-mediated apoptosis, or RIPK3/MLKL-mediated necroptosis did not

protect cells from ZBP1(S)-induced cell death (Extended Data Figure 6e-f). In contrast, elimination of components required for IFN signaling (IFN receptors and ISGF3 complex containing STAT1, STAT2 and IRF9) or the autophagy machinery (ATG5, ATG7, and ATG12, all required for autophagosome biogenesis) attenuated the frequency of cell death induced by ZBP1(S) (Extended Data Figure 6e-f).

- ZBP1(S) was fused to the mitochondrial targeting sequence of FIS1 to direct it specifically to the MOM in growing cells with functional telomeres (Extended Data Figure 16a). Under these circumstances, ZBP1(S) colocalized with MAVS, triggered an IFN response, stimulated autophagy, and induced cell death (Figure 4d-g; Extended Data Figure 16a-e). Suppression of MAVS attenuated the IFN response, prevented autophagy, and reduced the frequency of cell death (Figure 4d-g; Extended Data Figure 16a-e). Comparable attenuation was observed upon disruption of IFNAR signaling pathway through the depletion of type I interferon (IFN) receptor 2. (Extended Data Figure 16c-e). Indeed, siRNA-mediated suppression of IFNAR2 reduced autophagy and cell death frequencies to the same extend as MAVS suppression. These experiments were performed using human fibroblasts (IMR90E6E7) as well as mammary epithelial cells (HMECs) that have escaped from senescence through a spontaneous silencing of p16 (Nassour et al., Nature 565, 659–663), indicating that this pathway is conserved across different cell types (Extended Data Figure 16c-e).

Taken together, these findings reveal a novel mechanism of action by which the ZBP1-MAVS axis drives cell death in a manner that dependent on signaling activity of IFNs and their cognate IFNAR receptor and autophagy. Our data support previous studies describing a novel function of type I IFN as an inducer of autophagy in multiple cell lines (Hana Schmeisser et al. Autophagy 9:5, 683–696; Malene Ambjørn et al. Autophagy, 9:3, 287-302).

Referee 3 - Figure 1. Our data support a model where autophagy-dependent cell death during crisis implicates a simultaneous activation of the cGAS-STING and ZBP1-MAVS innate immune pathways, resulting in a persistent production of type I IFNs and a chronic autocrine/paracrine IFN signaling. The precise signaling mechanisms through which these IFNs induce autophagy and how autophagy executes cell death are well beyond the scope of the current manuscript and constitute a priority subject of future research.

Reviewer Reports on the Third Revision:

Referees' comments:

Referee #1 (Remarks to the Author):

The authors have now excluded that MAVS simply primes cGAS/STING expression in crisis cells, which is consistent with the proposed new ZBP1(S)-MAVS signalling pathway. However, I remain somewhat skeptical of this aspect of the manuscript due to the lack of biochemical data.

Nonetheless, I would like to reiterate that the main discovery of the role of ZBP1 in sensing dysfunctional telomeres and driving crisis are well supported and timely. As such, I recommend publication in Nature.

Referee #3 (Remarks to the Author):

Nassour et al have provided strong evidence that ZBP1 is important for replicative crisis caused by telomere dysfunction. They further showed that ZBP1 binds to the TERRA RNA and forms filaments that interact with and activate MAVS on the mitochondrial outer membrane. MAVS then induces type-I interferons (IFNs). The data supporting this ZBP1/TERRA – MAVS – IFN pathway is solid overall. The authors further propose that IFNs induce autophagy, which in turn cause cell death observed in replicative crisis. This IFN > autophagy > cell death model is less convincing (the right panel of Referee 3 – Figure 1). After all, IFNs have been used in humans for treating hepatitis and multiple sclerosis and there are no reports of severe cell death caused by IFNs. The authors seem to acknowledge this lack of understanding by stating in the rebuttal letter that “the precise signaling mechanisms through which these IFNs induce autophagy and how autophagy executes cell death are well beyond the scope of the current manuscript ...”. Since this is the third round of review, it’s up to the editors to decide whether the finding that ZBP1/TERRA is important for replicative crisis caused by telomere dysfunction warrants its publication in Nature. Perhaps so.

Author Rebuttals to Third Revision:

Response to referees:

We thank Referees 1 and 3 for their time and constructive criticism throughout three review cycles. We truly appreciate the support, which has made this a much better manuscript.

Referee #1 (Remarks to the Author):

The authors have now excluded that MAVS simply primes cGAS/STING expression in crisis cells, which is consistent with the proposed new ZBP1(S)-MAVS signalling pathway. However, I remain somewhat skeptical of this aspect of the manuscript due to the lack of biochemical data. Nonetheless, I would like to reiterate that the main discovery of the role of ZBP1 in sensing dysfunctional telomers and driving crisis are well supported and timely. As such, I recommend publication in Nature.

Response: Considering that deletion of MAVS did not affect the cGAS/STING response, we find it difficult to design biochemical experiments that demonstrate the absence of an interaction.

We very much thank the referee for supporting our main discovery of the telomere to mitochondria communication via ZBP1-TERRA.

Referee #3 (Remarks to the Author):

Nassour et al have provided strong evidence that ZBP1 is important for replicative crisis caused by telomere dysfunction. They further showed that ZBP1 binds to the TERRA RNA and forms filaments that interact with and activate MAVS on the mitochondrial outer membrane. MAVS then induces type-I interferons (IFNs). The data supporting this ZBP1/TERRA – MAVS – IFN pathway is solid overall. The authors further propose that IFNs induce autophagy, which in turn cause cell death observed in replicative crisis. This IFN > autophagy > cell death model is less convincing (the right panel of Referee 3 – Figure 1). After all, IFNs have been used in humans for treating hepatitis and multiple sclerosis and there are no reports of severe cell death caused by IFNs. The authors seem to acknowledge this lack of understanding by stating in the rebuttal letter that “the precise signaling mechanisms through which these IFNs induce autophagy and how autophagy executes cell death are well beyond the scope of the current manuscript ...”. Since this is the third round of review, it’s up to the editors to decide whether the finding that ZBP1/TERRA is important for replicative crisis caused by telomere dysfunction warrants its publication in Nature. Perhaps so.

Response: We appreciate the Referee’s comment regarding IFN treatment of several diseases, where no cell death is observed. We would like to point out that under such circumstances it is highly unlikely that telomeres are dysfunctional. And hence, TERRA are not transcribed. As a consequence, even if ZBP1 is elevated due to IFN treatment, it remains inactive, since it can not associate with TERRA.

As suggested by the Referee, we have emphasized in the second to last sentence of the manuscript that the molecular pathway between ZBP1-TERRA-MAVS and cell death by autophagy remains to be explored.

We deeply appreciate the Referee's support of our discovery of linking telomere dysfunction in crisis with ZBP1-TERRA fueled inflammation.